# The Role of Different Immunocompetent Cell Populations in the Pathogenesis of Head and Neck Cancer—Regulatory Mechanisms of Pro- and Anti-Cancer Activity and Their Impact on Immunotherapy

**DOI:** 10.3390/cancers15061642

**Published:** 2023-03-07

**Authors:** Katarzyna Starska-Kowarska

**Affiliations:** 1Department of Physiology, Pathophysiology and Clinical Immunology, Department of Clinical Physiology, Medical University of Lodz, Żeligowskiego 7/9, 90-752 Lodz, Poland; katarzyna.starska@umed.lodz.pl; Tel.: +48-604-541-412; 2Department of Otorhinolaryngology, EnelMed Center Expert, Drewnowska 58, 91-001 Lodz, Poland

**Keywords:** adoptive cell therapy, cancer invasion, cancer pathogenesis, cellular signalling, head and neck cancer (HNC), human papilloma virus (HPV), immune cells, immunotherapy, immune checkpoints, pro- and anti-cancer activity, squamous cell carcinoma

## Abstract

**Simple Summary:**

According to the latest GLOBOCAN data, head and neck squamous cell carcinoma (HNSCC) represents the sixth most prevalent human malignancy. Recent studies indicate that various immune cell populations may determine the pathogenesis of HNSCCs. The aim of this review was to provide a comprehensive overview of the role of the immune response in HNSCC tumorigenesis, molecular signatures and the mechanisms regulating pro- and anti-cancer activity; it also examines their impact on the current status and future prospects of immunotherapeutic strategies for overcoming immune escape of HNSCC. The study corpus encompasses a wide range of recent molecular, observational and intervention studies on the role of immune signalling pathways and interaction between neoplastic cells and immune cells in human HNSCCs. Rapid advances in the field of immuno-oncology and the constantly growing body of knowledge concerning immunosuppressive mechanisms have allowed effective and personalized immunotherapy to be used as a first-line therapeutic procedure or an essential component of a combination therapy for primary, relapsed and metastatic HNSCC. A greater understanding of the immune response in cancers may also contribute to the further identification of new potential immunological biomarkers necessary for greater clinical benefit in HNSCC patients.

**Abstract:**

Head and neck squamous cell carcinoma (HNSCC) is one of the most aggressive and heterogeneous groups of human neoplasms. HNSCC is characterized by high morbidity, accounting for 3% of all cancers, and high mortality with ~1.5% of all cancer deaths. It was the most common cancer worldwide in 2020, according to the latest GLOBOCAN data, representing the seventh most prevalent human malignancy. Despite great advances in surgical techniques and the application of modern combinations and cytotoxic therapies, HNSCC remains a leading cause of death worldwide with a low overall survival rate not exceeding 40–60% of the patient population. The most common causes of death in patients are its frequent nodal metastases and local neoplastic recurrences, as well as the relatively low response to treatment and severe drug resistance. Much evidence suggests that the tumour microenvironment (TME), tumour infiltrating lymphocytes (TILs) and circulating various subpopulations of immunocompetent cells, such regulatory T cells (CD4^+^CD25^+^Foxp3^+^T_regs_), cytotoxic CD3^+^CD8^+^ T cells (CTLs) and CD3^+^CD4^+^ T helper type 1/2/9/17 (Th_1_/Th_2_/Th_9_/Th_17_) lymphocytes, T follicular helper cells (T_fh_) and CD56^dim^/CD16^bright^ activated natural killer cells (NK), carcinoma-associated fibroblasts (CAFs), myeloid-derived suppressor cells (MDSCs), tumour-associated neutrophils (N1/N2 TANs), as well as tumour-associated macrophages (M1/M2 phenotype TAMs) can affect initiation, progression and spread of HNSCC and determine the response to immunotherapy. Rapid advances in the field of immuno-oncology and the constantly growing knowledge of the immunosuppressive mechanisms and effects of tumour cancer have allowed for the use of effective and personalized immunotherapy as a first-line therapeutic procedure or an essential component of a combination therapy for primary, relapsed and metastatic HNSCC. This review presents the latest reports and molecular studies regarding the anti-tumour role of selected subpopulations of immunocompetent cells in the pathogenesis of HNSCC, including HPV*^+ve^* (HPV^+^) and HPV*^−ve^* (HPV^−^) tumours. The article focuses on the crucial regulatory mechanisms of pro- and anti-tumour activity, key genetic or epigenetic changes that favour tumour immune escape, and the strategies that the tumour employs to avoid recognition by immunocompetent cells, as well as resistance mechanisms to T and NK cell-based immunotherapy in HNSCC. The present review also provides an overview of the pre- and clinical early trials (I/II phase) and phase-III clinical trials published in this arena, which highlight the unprecedented effectiveness and limitations of immunotherapy in HNSCC, and the emerging issues facing the field of HNSCC immuno-oncology.

## 1. Introduction

Squamous cell carcinoma of the head and neck (HNSCC) is the most common histological type of a heterogeneous group of malignant neoplasms originating from the mucosa of the upper respiratory system and the gastrointestinal tract [1,2]. A typical feature of all HNSCC tumours, regardless of origin, is their considerable diversity in terms of morphological and molecular changes, and thus their clinical phenotype. The diverse biological features of HNSCC largely determine the clinical course of the neoplastic disease and individual parameters, i.e., factors promoting the development of lesions and rates of morbidity and mortality, tumour progression and advancement, treatment outcomes and long-term prognosis, as well as local and nodal recurrence rates, and patient survival [3]. According to the latest data from Global Cancer Statistics (GLOBOCAN), HNSCC is the seventh most common cancer in the world, affecting over 660,000 people. The HNSCC diagnosis rate increased to almost 900,000 cases/year in 2020, which accounts for ~4.5% of malignant neoplasms in humans. It is predicted that by 2030, the incidence of HNSCC will increase by another 30% to almost 1,000,000 new cases annually (GLOBOCAN; gco.iarc.fr/today, accessed on 30 January 2023). Currently, the increasing incidence of oropharyngeal squamous cell carcinoma (OPSCC) is believed to have the greatest influence on annual head and neck cancer incidence rates [4,5]. Unfortunately, most HNSCC patients are diagnosed with an advanced stage of the malignant disease, i.e., WHO classification stages III and IV; as such, despite significant advances in diagnostic and surgical techniques and the use of combined treatment, global overall survival or 5-year tumour-free survival rates do not exceed 40–60% in HNSCC patients. Importantly, survival rates significantly differ depending on the tumour HPV status. When the prognostic significance of HPV is taken into account, tumour HPV positivity has been found to be associated with a significant improvement in overall survival, especially in cases associated with high-risk HPV oropharyngeal carcinoma (OPSCC), which is much more commonly diagnosed in the earlier stages. Patients with OPSCC demonstrate 75–80% 5-year overall survival in HPV-positive tumours, regardless of the mode of treatment, as opposed to 45–50% in HPV-negative tumours [6,7,8,9,10].

Although HNSCCs are derived from the same squamous epithelium, they represent a very biologically and clinically diverse group of malignant tumours with various risk factors, molecular pathogeneses, responses to treatment, and prognoses [11,12,13,14,15,16]. The traditionally most important clinical factors for HNSCC development are tobacco-derived carcinogens and excessive alcohol consumption. In the literature, the smoke-related head and neck cancer subtype is often referred to as non-HPV-associated disease or HPV*^−ve^* (HPV^−^) tumours. Research clearly indicates that tobacco-smoking patients are subject to a significantly higher risk of developing HNSCC, one that can be 5 to 25-times higher than the non-smoking population. While alcohol promotes epithelial atrophy and lipid decomposition of cell membranes and promotes the facilitated absorption of tobacco-derived carcinogens, it is also known to be a co-carcinogen which multiplies the risk of HNSCC in a dose-response manner among people who smoke tobacco simultaneously. Moreover, the major metabolite of ethanol metabolism, acetaldehyde, is also highly mutagenic. The coexistence of these two risk factors increases the risk of developing HNSCC by up to 35 times [12,14,15]. Tobacco smoke contains ~7000 toxic substances, of which 60 constitute active carcinogens. Tobacco components, such as benzo[a]pyrene, a key representative of polycyclic aromatic hydrocarbons (PAH), as well as tobacco-specific nitrosamines (TSNAs) and N′-Nitrosonornicotine (NNN), may contribute to the promotion, progression, nodal metastasis and further dissemination of HNSCC by inducting an epithelial-mesenchymal transition (EMT)-like phenotype [12,14,15].

In 2015, The Cancer Genome Atlas Network (TCGA) conducted a massive genetic study based on 279 primary head and neck tumours derived from common HNSCC sites. The epithelial cells were found to undergo a number of changes, including the inactivation of cancer suppressor genes and the overexpression of proto-oncogenes, resulting in tumour cell proliferation and distant metastasis. The TCGA gene mutations were separated into four groups comprising genes regulating the cell-cycle (*CDKN2A* and *CCND1*), those regulating cell proliferation and survival (*TP53*, *HRAS*, *PIK3CA*, and *EGFR*), those responsible for cellular differentiation (*NOTCH1*), and the Wnt signalling pathway regulator gene *FAT1*, a protocadherin modulator of adhesion and invasion [3]. Moreover, HPV*^−ve^* (HPV^−^) cancers commonly present with a loss of chromosome 9p, responsible for the downregulation of p16 (CDKN2A) expression, and the duplication of chromosome 7p, promoting the epidermal growth factor receptor (EGFR) overexpression [17,18]. The TCGA further categorized genes into the following: well-described smoking-associated mutational signatures, altering protooncogenes, i.e., *c-MYC*, *c-KIT*, *EGFR*, *HER-2*, *RAS-KRAS*, *HRAS*, *NRAS*, *ERB-B*, *BRAF*, *BCL-2*, *STAT3* and tumour suppression genes called anti-oncogenes, i.e., *RB1*, *P53*, *INK4*, *PTEN*, *CDKN2A*, as well as genes promoting a pro-inflammatory tumour microenvironment [3,19]. Nicotine also increases the expression of mesenchymal marker proteins, such as fibronectin and vimentin, but inhibits the action of the epithelial marker proteins β-catenin and E-cadherin [15]. Furthermore, nicotine may interfere with drug efficacy through CYP-metabolism, glucuronidation and crucial protein binding. This association was found to be stronger among oropharyngeal and hypopharyngeal tumours than those of the oral cavity and larynx [8,9,10]. Smoking remains an undisputed parameter determining worse treatment outcomes, i.e., high prevalence of loco-regional recurrence and/or metastatic disease, shortening the 5-year overall survival time, and increasing the risk of secondary primary neoplasms (SPNs). In contrast, smoking cessation and alcohol abstinence significantly reduce the risk of developing of HNSCC. Other risk factors, such as chewing betel nut, unbalanced and poor diet, and poor oral health have also been established in recent decades as contributing factors [12,14,15]. Interestingly, HPV^+^ cancer had a more favourable prognosis and distinct clinical features, behaving as a completely different disease to HPV^−^ HNSCCs [12,16,20].

Over the last twenty years, epidemiologists have reported the occurrence of an increasingly common variant of HNSCC pathogenetically associated with the human papillomavirus (HPV) and known as HPV-associated/HPV-related tumours or HPV*^+ve^* cancers. These HNSCCs are less frequent, accounting for approximately 26% of all HNSCC cases worldwide. Tumours initiated independently of other common carcinogens, such as smoking and alcohol consumption are most often located in the oropharynx and tonsils (OPSCC), they present an early primary tumour stage and most commonly affect younger, male, Caucasian non-smokers of higher socioeconomic status in developed countries. Indeed, patients with HPV-associated tumours are typically diagnosed with large, cystic metastatic cervical lymph nodes; however, they are highly responsive to standard treatment approaches and have significantly better prognoses compared to HPV-negative patients [21,22]. However, it should be noted that co-factors, such as cigarette smoke, can also enhance the risk of such HPV-related cancers [11,12,13,14,15,16].

Importantly, this biologically distinct subgroup of HNSCC is associated with prior infection with oncogenic strains of subtypes HPV16 and HPV18, and rarely HPV33 and HPV52, and shows a better prognosis and a higher 5-year overall survival rate (70–80%), as well as improved disease-free survival. Moreover, despite presenting more frequently locally advanced disease and regional lymph node metastases at the time of diagnosis, OPSCC patients are more suitable for treatment de-escalation due to better responses to chemo- and radiotherapy and favourable therapeutic results [6,11,12,13,14,15,16].

The molecular pathogenesis of carcinogenesis in OPSCC tumours is related to the integration of HPV genomic DNA into the genetic material of host epithelial cells. This phenomenon results in the increased expression of HPV16/18 E5, E6 and E7 viral antigens, which act as oncoproteins/oncogenic drivers that promote tumour growth and malignant transformation, as well as the transcriptional regulatory protein E2. In HPV16 infected cells, the E5 oncoprotein plays a key role in cell-cycle regulation, cell growth and impairs several signal transduction pathways. Furthermore, pro-carcinogenic activities are also performed by HPV16/18 E5, including the stimulation of EGF-mediated cell proliferation, the inhibition of apoptosis induced by tumour necrosis factor ligand (TNFL) and CD95 ligand (CD95L) and the modulation of genes involved in cell adhesion and cell motility [23]. As a consequence, this leads to a loss of control of key intracellular signalling pathways responsible for cell cycle control due to the degradation of cell cycle regulatory proteins, i.e., tumour suppressor protein p53 by HPV16/18 E6 and pRB1 by HPV16/18 E7. This forces the cells to undergo uncontrolled cellular division, evading the preventive checkpoints and leading to the release of E2F transcription factors, which transcribe cyclin E, cyclin A and p16^INK4A^, an inhibitor of CDK4/6, resulting in the cells undergoing premature entry into the S-phase. HPV-infected cells show the co-expression of E6 and E7 oncoproteins, which establishes the perfect environment for sustained proliferative signalling [14,24,25]. Moreover, the HPV16/18 E6 protein interacts with the *c-MYC* oncogene, which forms the c-MYC/E6 complex, which stimulates the transcription of the human telomerase catalytic subunit (hTERT), contributing to the immortalization of neoplastic cells [14,15]. Deregulation of c-MYC leads to the disruption of Cdks, cyclins and E2F transcription factors, as MYC is capable of inducing cyclin/Cdk complexes with the help of Cdk activating kinase (CAK) and Cdc25 phosphatases. MYC is further found to reverse the Cdk inhibiting activity of p27^K*IP*1^ and p21^CIP1/WAF1^ [19,24]. The TCGA also highlighted a gene mutational spectrum for HPV-related tumours, including *PIK3CA*, *DDX3X*, *CYLD* and *FGFR* [26]. HPV-driven cancers also express amplifications in chromosome 3q, and the loss of chromosomes 16p, 16q, 14q, 13q and 11q [17,18]. Interestingly, HPV-positive tumours demonstrate stronger activation of several immune signalling pathways, as well as the higher expression of genes related to total tumour-infiltrating lymphocytes (TILs), CD8^+^ T cells, effector cytotoxic cells (CTLs), exhausted CD8^+^ cells and macrophages (TAMs) than HPV-negative types. Moreover, HPV*^+ve^* lesions demonstrate increased PD-L1 expression compared to HPV*^−ve^* neoplasms, and PD-L1^+^ tumour cells and macrophages are closer to PD-1^+^ cytotoxic T lymphocytes. Thus, it would be beneficial to incorporate immune checkpoint inhibitors (ICIs) in the loco-regional therapy strategies for patients with heavily infiltrated treatment-naïve HPV*^+ve^* HNSCCs and for combining ICIs with tumour-specific T cell response inducers or TAM modulators in cases of advanced regional head and neck cancer (R/M HNSCCs) [27,28,29,30,31]. Due to these fundamental differences between non-HPV-associated HNSCC and HPV-related OPSCC, the 2017 8th Edition AJCC/UICC (The American Joint Committee on Cancer AJCC/Union for International Cancer Control UICC) updated the traditional staging of HNSCCs based on the pathological tumour-node-metastasis system (pTNM) to incorporate additional information relevant to the HPV-positive disease [7,13,20,32].

During carcinogenesis, the immune cells undergo reprogramming, which is also accompanied by the remodelling of the tumour microenvironment (TME) to avoid anti-cancer immune mechanisms. Furthermore, the changes in the tumour milieu impede and even inhibit the humoral and cellular immune responses and change potentially malignant anti-immune behaviours, e.g., by creating tumour escape mechanisms from immune surveillance [33,34]. Moreover, a number of extensive carcinogenic interactions can be identified between immunocompetent cells and the TME compounds and neoplastic cells. It is well known that the immune microenvironment gradually becomes suppressive over the course of tumorigenesis, and favours immune escape [35,36,37,38]. It is also known that immune cells can transform into phenotypes that suppress the immune function (M2 phenotype TAMs, N2 TANs), produce anti-inflammatory mediators (NET, IL-10, TGF-β and IL-2) and recruit immunosuppressive cells (T_regs_, MDSCs, CAFs) that favour potentially malignant cells capable of evading immunity. Moreover, increased angiogenesis mediated by VEGF accelerates the process of carcinogenesis [33,39]. Recent studies clearly indicate that HNSCC patients are characterized by impaired dendritic cells (DCs) and natural killer (NK) cell function, increased production of inhibiting cytokines and immunosuppressive factors, loss of costimulatory molecules and MHC class I molecules, a decrease in the number of regulatory T cell subpopulations, a lower number of lymphocyte subsets and a poor response to antigen-presenting cells [14]. Therefore, in an attempt to identify appropriate targets for the effective immunotherapy of various cancers, many researchers have attempted to explain the dynamic mechanisms between immune cells and cancer cells occurring in the tumour niche. Unfortunately, a comprehensive review of the role of different subpopulations of immune system cells and TME changes during the carcinogenesis of HNSCCs, and the importance of interactions between these microenvironments is still lacking; in addition, few proven strategies exist for prophylaxis or effective personalized treatment aimed at enhancing the anti-cancer response to cancer antigens. Furthermore, despite advances in the development of new treatments over the past few decades, prognosis and overall survival for patients with HNSCC remain low and unsatisfactory, resulting in this being one of the deadliest human cancers. Therefore, attempting to establish effective mono-immunotherapy or combination therapies, as well as developing new therapeutic options, requires a thorough understanding of the interaction between the immune system and HNSCC [14,40,41].

The latest generating consensus statement on immunotherapy of squamous cell carcinoma of the head and neck (HNSCC) was formed by the Society for Immunotherapy of Cancer (SITC) and it can be found on the SITC website (available from: https://www.sitcancer.org/research/cancer-immunotherapy-guidelines, date last access 30 January 2023) [42]. The current guidelines for the use of immunotherapy in patients with HNSCC are also presented in the National Comprehensive Cancer Network^®^ (NCCN^®^) Clinical Practice Guidelines in Oncology (NCCN Guidelines^®^). Head and Neck Cancers; NCCN Evidence BlocksTM (Version 1.2023, 01/30/23) [Available from: www.nccn.org/patents for a current list of applicable patents. Head and Neck Cancers Version 1.2023—30 January 2023] [43]. The immune checkpoint inhibitor anti-programmed cell death protein 1 (PD-1)/PD-L1 was the first drug to demonstrate prognostic benefits for the treatment of R/M (recurrent/metastatic) HNSCC unresponsive to platinum-based treatment, and was recently granted approval for use by the Food and Drug Administration (FDA). Other immune checkpoint receptors, such as cytotoxic T-lymphocyte-associated protein 4 (CTLA-4), have also been gaining more attention for use in monotherapy or in conjunction with other immunotherapies or conventional cancer therapeutics [44,45,46,47]. Therefore, the expert committee at the SITC has developed jointly agreed recommendations on emerging immunotherapies, including appropriate patient population selection, therapy sequence, response monitoring, adverse events and biomarker research [42,43]. The key idea of the consensus was to contribute to a better understanding of the role of immunotherapy in HNSCC and to introduce a useful tool underpinning clinical therapeutic decisions; it was also intended to standardize the indications and benefits of immune monotherapy for patients, and the inclusion of immunotherapy as part of new combination therapy strategies for HNSCC. The consensus also indicates crucial clinical immunotherapy recommendations for the treatment of patients with HNSCC, highlighting the great importance of the U.S. Food and Drug Administration, FDA-approved therapies with the use of, inter alia, immune checkpoint inhibitor (ICI) targeting proteins, such as anti-programmed cell death protein (PD-1) and programmed death ligand 1 (PD-L1) or cytotoxic T-lymphocyte-associated protein 4 (CTLA-4), i.e., pembrolizumab (Keytruda, Merck) and nivolumab (Opdivo, Bristol-Myers Squibb); these can be used as second-line therapies in patients with relapsed/metastatic head and neck squamous cell cancer (R/M HNSCC) and who are refractory to platinum-based chemotherapy. The statement answers key clinical questions and presents treatment algorithms based on completed and/or continued randomized controlled clinical landmark trials, i.e., KEYNOTE-012 (clinicaltrials.gov identifier: NCT01848834), KEYNOTE-040 (NCT02252042)—pembrolizumab vs. standard single-agent systemic therapy for patients diagnosed with R/M HNSCC, CheckMate-141 (NCT02105636)—nivolumab vs. standard single-agent systemic therapy, KEYNOTE-048 (NCT02358031)—pembrolizumab monotherapy vs. EXTREME, pembrolizumab plus platinum-based chemotherapy (CT) vs. EXTREME, Active8 (NCT01836029), KESTREL (NCT02551159), CheckMate-651 (NCT02741570) and EAGLE (NCT02369874)—durvalumab with/without tremelimumab vs. standard single-agent systemic therapy, etc. [43,48].

Unfortunately, the general failure of treatment in head and neck cancers has prompted surgeons and biologists to seek new therapeutic solutions to develop modern combinations and cytotoxic therapies for HNSCC that would give patients a chance for longer survival and a better health-related quality of life (HRQoL). The current review provides a comprehensive summary of the role of selected subpopulations of immune cells involved in HNSCC pathogenesis, and highlights the most recent knowledge on the immune surveillance and immunoediting, as well as the biological mechanisms of tumour immune escape involved in HNSCC progression. It also summarises the evidence supporting the further development in immunotherapy, and proposes promising new combinations and personalized treatment modalities for head and neck squamous cell carcinoma.

### 1.1. Emerging Key Immune Cells in the Immune Evasion of Cancer—Tumour Infiltrating Lymphocytes (TILs) and Tumour-Associated Immunocompetent Cells in the TME

#### 1.1.1. Tumour Infiltrating Lymphocytes (TILs)

Immune cells play a dual role in the dynamic ecosystem of the cancer tumour microenvironment (TME). Responses orchestrated by tumour-specific immunocompetent cells infiltrating the TME may effectively trigger mechanisms that inhibit initiation and cancer progression. Unfortunately, more often, the tumour microenvironment becomes a unique and complex environment that promotes cancer development. Under the influence of active machineries of the immune-escape of a tumour from immune surveillance, immune cells may transform into immunologically ineffective cells with tolerogenic phenotypes that are capable of strongly sustaining carcinogenesis [49,50,51]. Recent studies in human cancers have confirmed that the cellular and non-cellular composition of the tumour immune microenvironment (TME), particularly the activity of cytotoxic T cells, can be an essential factor that can predict clinical outcomes [36,52]. In the TME, the successive steps of initiation, growth and invasion at the primary tumour site, up to extravasation and colonisation of tumour cells at the metastasis site, are governed by a complex network of signalling pathways and intercellular interactions between tumour stroma components, immune cells and neoplastic cells [52,53,54].

HNSCC microenvironments harbour multiple cell types that infiltrate the intracellular tumour matrix and interact with each other and with neoplastic cells. Tumour infiltrating lymphocytes (TILs) constitute a subpopulation of immunocompetent mono- and polymorphonuclear lymphocytic cells, the most important of which includes a subset of CD3^+^CD4^+^ T helper type 1 (Th_1_) lymphocytes, cytotoxic CD3^+^CD8^+^ T cells (CTLs) and regulatory T lymphocytes (CD4^+^CD25^+^Foxp3^+^T_regs_). In addition to T cells, variable proportions of CD56^dim^/CD16^bright^ activated natural killers (NK) cells, macrophages (M1/M2 phenotype macrophages, TAMs), neutrophils (N1/N2 TANs), dendritic cells (DCs), myeloid-derived suppressor cells (MDSCs), mast cells, eosinophils and basophils, and an insignificant number of B-line cells also infiltrate tumour tissue [49,50,51,55,56,57]. Growing evidence indicates that the immunocompetent cells in the TME may contribute to tumour initiation and progression. Interactions between neoplastic cells and the innate immune cells may result in an environment and local niche that favours tumour spread and metastasis [39,52,53,54].

Research indicates that CD4^+^ T helper cells, CD8^+^ cytotoxic T cells, NK cells, N1 neutrophils, M1 macrophages and DCs are protective against tumour growth, and that an effective interaction of these subsets is needed to protect the host against a developing tumour. Conversely, the T_regs_ subset of CD3^+^CD4^+^ lymphocytes, N2 neutrophils, M2 macrophages and MDSCs can promote carcinogenesis and facilitate tumour immune escape from immune destruction [49,50,55,56,57]. These interactions between immunocompetent cells, both innate (e.g., NK cells) and adaptive (e.g., CD8^+^ T cells) cells and the tumour itself are together referred to as immunoediting [49,51]. In recent years, many researchers have indicated that the evaluation of immune infiltration in the tumour microenvironment may have a relevant prognostic value in various cancers, including HNSCC. Moreover, in the era of immunotherapy, the immune infiltrate of tumours and the parameters of the innate and adaptive anti-cancer immune response is proposed to be a possible useful predictive marker of treatment efficacy [49,51,58].

Numerous studies on the role of tumour-infiltrating immunocompetent cells and the neoplasm microenvironment indicate the existence of two categories of tumour escape from immune surveillance, indicated on the basis of both cellular changes and the characteristic molecular features of the neoplastic lesions [59]. One major subset is associated with the activity of tumour infiltrating lymphocytes (TILs) and other inflammatory cells, the production of a wide panel of cytokines/chemokines and a type-I interferon signature, and concerns innate immune activation. This phenotype contributes to the conditions for immune escape of the tumour by the dominance of immune-suppressing pathways. Another group of neoplastic lesions are “null phenotype” tumours, characterized by the absence of immune-competent inflammatory cells; these are able to resist the immune system due to exclusion or ignorance, as well as by T cell anergy, leading to T cell-intrinsic dysfunction, as well as anergy-promoting tumour environment. Consequently, these two distinct tumour immunophenotypes appear to warrant considering immunologically diverse therapeutic interventions [59,60].

The main function of TILs in the stroma is killing cancer cells. In the tumour microenvironment, activated Th_1_ lymphocytes and other CD133^+^CD44^+^ cancer stem cells (CSCs) stimulate in vivo the effector cytotoxic CTLs by secreting the key cytokine IL-2 and interacting through costimulatory molecules, i.e., MHC class II molecule, CD27, a member of the TNF receptor superfamily, and CD134, called TNF receptor superfamily member 4 (TNFRSF4) or OX-40 receptor [61]. However, the role of T cells in TME is ambiguous. Apart from the presence of an active Th_1_ subpopulation, Th_2_ cells can also appear in the tumour microenvironment. This phenotypic dichotomy of T cell subpopulations in TILs is related to the different coordination of the immune response to neoplastic antigens through differential cytokine production, i.e., Th_1_ cells govern the pro-inflammatory phenotype and Th_2_ cells govern the immunosuppressive phenotype [39]. Mechanistically, cytokines secreted by specific subpopulations of T cells within the TME can manipulate the immune functions of immunocompetent cells against neoplastic antigens, culminating in the cycle of tumour immunity and subsequent stages of tumour immunoediting leading to growth and progression or, alternatively, enhancing innate immune responses directed against cancerous cells [39].

Importantly, TILs demonstrate different anti-tumour activity against tumour neoantigens in both HPV*^+ve^* and HPV*^−ve^* HNSCC. A study of RNA sequencing data from the HNSCC cohort of the Cancer Genome Atlas tumour bank found that both HPV^+^ and HPV^−^ HNSCC exhibit one of the strongest forms of immune infiltration in comparison with the most immunogenic types of cancer [62]. Hypothetically, the difference in the anti-tumour response between HPV^+^ and HPV^−^ tumours is probably due to a different immune response directed against the viral antigens (oncoproteins E5, E6 and E7). Hence, the prognostic significance of TILs according to the HPV status of the tumour remains a subject of controversy. More trials, preferably randomized, with a sufficient number of patients are required in order to elucidate the relationship between TILs and the presence of HPV.

It has been confirmed that the presence of TILs that affect the adaptive immune response to tumour antigens correlates directly with the phenotype of neoplasm aggressiveness, the local extent of neoplastic lesions, the presence of nodal metastases, and the rate of response to the applied therapies, as well as local and distant recurrences of different solid tumours, including HNSCC. In recent years, many clinical studies, systematic reviews and meta-analyses have been conducted to assess the prognostic value of the presence of various subpopulations of TILs in patients with HNSCC. The overall conclusion, based on a review of the literature, confirms that the presence of TILs in TME is frequently associated with better clinical outcomes and higher overall survival (OS), disease-free survival (DFS) and negative locoregional tumour control (LCR), as well as distant metastasis free survival (DMFS) in both after surgery and/or immunotherapy in a wide panel of human solid cancers [55,63,64]. The prognostic value of different TIL subsets with regard to variation in tumour primary subsites, pTNM classification and stage, as well as HPV^+^ and HPV^−^ HNSCC condition was also discussed in recent systematic reviews and meta-analyses [55,63,64].

An interesting multicentre meta-analysis from the previous year indicates that a reduction of the risk of cancer death among patients treated for HNSCC is significantly related to the presence of large CD4^+^ TILs (HR 0.77; 95% CI: 0.65–0.93) and CD8^+^ (HR 0.64; 95% CI: 0.47–0.88) subpopulations [63]. Importantly, the key parameter in this study was the location of the primary tumour, i.e., high values of CD4^+^ TILs were related to a better OS among oropharyngeal HNSCC (HR 0.52; 95% CI: 0.31–0.89) [63,65,66]. High levels of CD8^+^ TILs were associated with a better prognosis in hypopharyngeal cancers (HR 0.43; 95% CI: 0.30–0.63), and in both HPV-related tumours and non-HPV-associated tumours (HR 0.39; 95% CI: 0.16–0.93 and HR 0.40; 95% CI 0.21–0.76, respectively) [63,67,68,69]. Moreover, the assessment of pre-treatment CD8^+^ TIL density was also confirmed as a valuable predictor of distant metastasis after definitive treatment in patients with stage III/IV in hypopharyngeal HNSCC. Unfortunately, no significant association occurred for individuals with HNSCC of the oral cavity or larynx [63].

The presence of CD3^+^ TILs in both HPV*^+ve^* and in HPV*^−ve^* tumours was found to have a strong influence on clinical outcome, resulting in the significant prolongation of overall and disease-free survival and the negative locoregional tumour control in another cohort of HNSCC [64,65,70,71,72,73]. The pooled analysis confirmed that the CD3^+^ population of TILs has a favourable prognostic role for OS (HR 0.64; 95% CI: 0.47–0.85) and for DFS (HR 0.63; 95% CI: 0.49–0.82) in HPV^−^ cancers. In HPV^+^ patients, high CD3^+^ TILs also demonstrated a similar clinical benefit for overall survival. However, the most reliable prognostic immune biomarker was the amount of cytotoxic CD8^+^ TIL cells (CTLs). Since CTLs belong to the subset of CD3^+^ lymphocytes that directly target tumour cells, their prognostic value has been the source of a great deal of research in recent years. The findings confirmed that high levels of CD8^+^ TILs have a favourable outcome for all prognostic parameters in the HPV^−^ HNSCC cohort, i.e., for OS (HR 0.67; 95% CI: 0.58–0.79), DFS (HR 0.50; 95% CI: 0.37–0.68) and LRC (HR 0.82; 95% CI: 0.70–0.96) [64,70,71,72]. High CD8^+^ TILs also offer similar advantages in HPV^+ve^ individuals for both OS (HR 0.31; 95% CI: 0.17–0.59) and DFS (HR 0.82; 95% CI: 0.70–0.96) [64,70,71,72,73]. Interestingly, the role of helper CD4^+^ T cells in the tumour microenvironment is not as clear and obvious. Meta-analyses have confirmed that high CD4^+^ TIL quantities are related to both better OS (HR 0.76; 95% CI: 0.64–0.89) and LCR (HR 0.81; 95% CI: 0.68–0.96) in HPV^−^ patients [64,65,70,71]. Interestingly, several studies have also investigated the influence of CD4^+^ subset of TILs on prognosis in HPV^+^ patients; however, they have yielded contradictory results for both OS (HR 1.33; 95% CI: 0.63–2.81) and LRC (HR 1.56; 95% CI: 0.78–3.12) in the studied HNSCC group [65,73]. These contradictory results could be explained by the heterogeneity of the CD4^+^ T cell population and a wide range of functions by the Th cell subsets. The prognostic role of CD4^+^ TILs may remain unclear due to not taking into account the heterogeneity in the expression of specific antigens/markers for the CD4^+^ T helper cell subpopulations, ranging from a cytotoxic cell response stimulating Th_1_ lymphocytes, thorough CD4^+^ Th type 2 or 17 lymphocytes (Th_2_ or Th_17_), T follicular helper cells (T_fh_) and to the immune suppressing regulatory T lymphocytes (CD4^+^CD25^+^Foxp3^+^T_regs_) [64]. It should be noted, however, that all systematic reviews and meta-analyses conclude that future studies are needed to ensure more insight in the predictive value of TILs in HNSCC, and these should employ larger patient cohorts that are homogeneous with regard to tumour primary subsite, tumour pTNM and stage, and include therapy with clinically proven, modern techniques.

Regulatory T lymphocytes (CD4^+^CD25^+^Foxp3^+^T_regs_), together with other immunosuppressive cells, are also a topic of growing interest and are studied extensively in HNSCC [49,58]. Importantly, T_regs_ constitute a key CD3^+^CD4^+^ subset that inhibits an effective anti-cancer immune response by producing the immunosuppressive cytokines IL-10 and TGF-β and consuming IL-2. Consequently, T_regs_ express negative co-stimulatory checkpoint inhibitors, e.g., CTLA-4 and PD-1, which suppress the activity of effector T cells. The immune-escape phenomenon of tumour cells is also determined by MDSCs which inhibit lymphocyte activity by inducing CD4^+^CD25^+^Foxp3^+^T_regs_ cells in TME, producing TGF-β, depletion or sequestration of amino acids required for T cell function, or nitration of T cell receptors or chemokine receptors on tumour-specific T cells [49,58].

Furthermore, in TILs, CD133^+^CD44^+^ cancer stem cells (CSCs) have the ability to impair the function of cytotoxic CD8^+^ T cells (CTLs) in TME by altering the expression of their programmed-death ligand 1 (PD-L1) receptor. Recent studies show that the increase in PD-L1 expression in TILs, leading to CSC evasion and epithelial-mesenchymal transition (EMT), occurs following activation of the EMT/β-catenin/STT3/PD-L1 signalling pathway and various epigenetic changes, i.e., STT3-dependent PD-L1 N-glycosylation and subsequent receptor stabilization and upregulation [74]. It was also shown that aberrant PD-L1 expression on the CD133^+^CD44^+^ CSC cell population was associated with increased activation of the Notch3/mTOR signalling axis and HMGA1-dependent pathways, including PI3K/AKT and MEK/ERK pathways in cancer cells in vitro and in vivo [75,76]. Moreover, activation of the IL-6/JAK1 axis enhanced PD-L1 phosphorylation by Janus kinase 1 to promote cancer immune evasion [77]. This oncogenic signalling can initiate the glycosylation of co-inhibitory molecules to induce immunosuppression. Interestingly, IL-6 targeting by antibodies was found to induce synergistic T cell killing effects when combined with T cell immunoglobulin and mucin-domain containing-3, also known as HAVCR2 receptor (TIM-3/ HAVCR2) therapy, in animal cancer models [77].

In conclusion, being the first-line cells of the adaptive immune system that are recruited to the neoplastic site, tumour-infiltrating lymphocytes (TILs) are of increasing interest to oncologists and immunologists. Unfortunately, the small number of TILs in TME and their lower activity limit their ability to fight the growing tumour. Nevertheless, TIL titre is an interesting prognostic factor in cancer patients. Research is also being conducted on the use of TILs in primary anti-cancer therapies [78,79].

#### 1.1.2. Tumour-Associated Immunocompetent Cells in the Tumour Microenvironment (TME)

##### Antigen-Presenting Cells (APCs)—Dendritic Cells (DCs)

Dendritic cells (DCs) form a heterogeneous population of immunocompetent cells that are responsible for priming the adaptive immune response through the presentation of tumour antigens acting as professional APCs. Phenotypically, DCs can be divided into subtypes that differ in ontogenesis, localization and action via various specific receptors and an individual panel of produced cytokines [80]. DCs originate from CD34^+^ bone marrow precursor cells and migrate with the blood stream to peripheral tissues where they capture antigens. DCs then travel to lymphoid tissues where the captured antigens are processed and presented to T cells. In the tumour microenvironment of HNSCC, DCs differentiate into myeloid and lymphoid/plasmacytoid cells, which play specific roles in the tumour-associated immunity cycle, i.e., myeloid DCs are known to enhance anti-tumour or anti-inflammatory immune responses, whereas lymphoid and plasmacytoid subtypes support immune deterrence and/or tolerance [49,81,82]. Many factors derived from TME cells and milieu, including VEGF, non-classical HLA class I, death ligands (FasL and TRAIL), pro- and anti-inflammatory cytokines (TNF-α, IL-10, IL-1β, TGFβ, IL-8) and metabolites (e.g., IDO, ROS, RNS, NO), are directly responsible for ensuring proper DC anti-tumoral activity [83]. A typical feature of myeloid DCs, the dominant category of DC subpopulation, is the presence of MHC class II, costimulatory molecules (CD40, CD80 and CD86) and adhesion molecules (CD11a, CD15s, CD18, CD29, CD44, CD49d, CD50 and CD54), which determine the development and activation of TILs. DCs with a classical DC phenotype, i.e., CD11c^+^CD11b^+^CD8^−^, present tumour-associated epitopes to the naïve antigen-specific T cells and provide costimulatory signals in effector T cell activation. DCs can also activate NK cells and B cells [80,82,84,85]. The presence of mature, active DCs in TME is associated with increased recruitment and immune activity of effector cells. Unfortunately, numerous studies in vitro indicate that DC function is effectively suppressed in the neoplastic milieu of different human cancers, and tumour conditioned media strongly influence dendritic cell maturation or apoptosis [49,81,82]. For instance, pre-treatment of monocyte-derived dendritic cells with the tumour conditioned media containing high levels of the chemokines CCL2, CXCL1, CXCL5 and VEGF inhibited the upregulation of CD86, CD83, CD54 and HLA-DR, enhancing IL-10 and reducing IL-12 secretion, and causing progressive increased expression of PD-1 on cancer-infiltrating DCs followed by paralysis of DCs [86]. Suppression of effector DCs appeared to be mediated by T cell-associated PD-1. In contrast, the antibody blockade against the immune regulator PD-1 molecule on tumour-associated DCs resulted in the inhibition of NF-κB activation, and release of regulatory cytokines, such as IFN-α, IL-6, IL-8 and TNF-α, and the upregulation of co-stimulatory molecules [87]. Therefore, in pre- and clinical trials in different phases, it is proposed to use DC-based vaccines that are stimulated with neoantigens specific for each patient. Cancer immunotherapy using DC-only vaccines or combined with inhibitory checkpoint blockade are expected to result in restoring the function of DCs as APCs [57].

The opinions on the role of specific subpopulations of DCs in the course of cancer disease, including HNSCC and prognosis, are ambiguous [49,88,89,90,91]. However, most researchers indicate that the higher densities of DCs present in TME correlate with high counts of tumour-infiltrating lymphocytes (TILs) and contribute to a favourable prognosis with increased progression-free and overall survival and a good anti-tumour therapy response [49]. In addition, recent studies confirm that the DCs present in TME show higher nuclear expression of the transcription factor interferon regulatory factor 7 (IRF7), which is characteristic of an activated DC phenotype [88]. Other investigators have observed that an increased number of tumour-infiltrating DCs correlates with lymph node metastasis and an adverse outcome in primary oral squamous cell carcinoma [89].

Due to their role in the priming of the adaptive immune response to tumour antigens, DCs are an attractive target for immune oncology-based therapeutic approaches. However, targeting DC cells has proven difficult, and many research studies have been inconclusive or have failed to yield benefits in a clinical trial setting [92,93,94]. Strategies to target DCs in cancer disease include a few mechanisms and targetability, such as maturation and differentiation, antigen processing and presentation, cytokine production and migration, immunomodulation of the TME and co-stimulation. These potential therapeutic strategies include various therapeutic approaches, including those that counteract the impaired function of DCs by blocking pathways that diminish their activation, such as STAT3, IDO and PD-1/PD-L1 axis or CTLA-4 immune checkpoint blockade; these employ the molecules for DC activation, such as Toll-like receptor (TLR) agonists, stimulator of interferon genes (STING) agonists or CD40 agonists, or therapies combined with other strategies, such as immune checkpoint blockade or DC vaccination [92,95].

The TME can be modulated by STAT3, promoting the suppression of IFN-α/β, IL-12, TNF-α, CCL5, CXCL10 and the upregulation of IL-6, IL-10, TGFβ and VEGF. STAT3 blockade can counter DC immunosuppression by inducing their maturation and increasing CD3^+^CD4^+^ and CD3^+^CD8^+^ T cells in TILs while decreasing them in T regulatory cells [95]. Indoleamine-2,3-dioxygenase (IDO) expression has a controversial impact in the TME, with higher IDO levels correlated with an inhibition of tumour growth and higher overall survival in cancer patients [96], while other studies indicate that IDO is often related to cancer progression, formation of metastatic niches, downregulation of T cell activity, higher frequencies of liver and pulmonary metastasis, and reduced TIL and NK cells in human cancers [92]. Importantly, IDO inhibition, for instance through the PF-06840003 inhibitor, has been confirmed to counter the immunosuppressive and effector T cell exhaustion effect when used in combination with anti-PD-L1 [92,97]. Recent phase I-III clinical trials have demonstrated that immune checkpoint blockade therapy or radiotherapy with IDO1 inhibitors yields more promising clinical results when used in combination than used separately [98]. However, until now, no phase III studies blocking these mechanisms have confirmed any significantly improved patient outcome [92,98]. Furthermore, few studies unequivocally confirm that TLR stimulation induces DCs maturation, inhibits their phagocytic abilities and upregulates costimulatory molecule expression, as well as promotes Th_1_ polarization and production of cytokines IL-12 and IFN-γ. For example, it was proven that treatment with the TLR3 agonist, polyI:C induced therapeutic activity in preclinical models of human cancers [92,99].

Currently, several preclinical and phase I/II studies also indicate that the administration of CpG oligodinucleotides (ODN), a TLR9 agonist, activates tumour-draining lymph node DCs and enhances the production of IFN-α/β cytokines, resulting in T cell activation. Moreover, the combined administration of TLR7 and TLR9 agonists with PD-1 blockade in HNSCCs suppresses tumour growth and progression related to TAMs and CD8^+^ T cell activation and increases the ability of DCs to gain a mature phenotype and migration [100].

Interestingly, STING (stimulator of interferon genes) is known to enhance the production of IFN-α/β in response to cytosolic DNA (cDNA) [101]. In cancer cells, IFN-α/β can cause decreased immunity, as IFN-β is related to the increase in PD-L1 and PD-L2 expression [92,102]. STING is necessary for anti-cancer immunity, including the promotion of DC maturation and their activation [92,103]. Moreover, it has been proven that STING signalling promotes STING-mediated interferon response in non-tumour cells to activate CD56^dim^/CD16^bright^ NK-dependent cytotoxicity, induces apoptosis of malignant B cells and remodels the TME by antagonizing myeloid-derived suppressor cell (MDSC) expansion in nasopharyngeal carcinoma, and enhances the presentation of TAAs by DCs and cross presentation to CD8^+^ T cells [103,104,105,106]. Recent clinical trials in metastatic HNSCCs and other solid tumours indicate the benefits of using the cyclic dinucleotides (CDNs) in combination with immune checkpoint inhibitor (ICI) therapy or as adjuvants in chemotherapy [106,107].

The CD40 molecule, the cell surface co-stimulatory member of the TNF superfamily on DCs, is associated with maturation, activation of a specific phenotype necessary for effector T cell activation, as well as DC survival and an anti-tumour DC40-dependent response. Thus, the lack of CD40 and IL-23 is related to DC immune tolerance [108,109]. Preclinical trials using in vivo administration of agonistic anti-CD40 antibodies resulted in effector T cell activation and tumour regression [110,111,112,113]. Moreover, recent research confirms that combined RT and CD40 activation therapy leads to an increased response to ICIs, causing remodelling in the TME via upregulation of MHC class I, costimulatory molecules, activation of DCs and infiltration of CD8^+^ T cells [114,115]. This phenomenon occurs by reducing the expression of the PD-1 molecule on effector T cells and downregulating CTLA4 and PD-1 on regulatory T cells (CD4^+^CD25^+^Foxp3^+^T_regs_), thus reversing resistance to anti-PD1 therapy [114,115]. However, it should be emphasized that the use of monotherapy with an anti-CD40 agonist resulted in no, or minimal clinical response in cancer patients; as such, it can be concluded that therapy with CD40 agonists should only be used in combination [109,116].

##### Regulatory T Cells (CD4^+^CD25^+^Foxp3+T_regs_)

Regulatory T cells (T_regs_) are a unique immunosuppressive subset of CD3^+^CD4^+^ T helper lymphocytes present in the tumour microenvironment (TME); they express specific biomarkers, such as CD4, CD25 and Foxp3 (forkhead box protein 3) and are thought to be derived from the same lineage as naïve CD4^+^ cells. Recruitment of CD4^+^CD25^+^Foxp3^+^T_reg_ subtype cells from circulation (T_regs_ account for 5–10% of CD4^+^ T cells in the peripheral circulation) to the neoplastic extracellular milieu is mediated through the activity of chemokines and related receptors, i.e., CCL28-CCR10 and CXCL12-CXCR4. T regulatory immunocompetent cells can make up as much as 20–30% of the total CD3^+^CD4^+^ population around the TME, thus causing large numbers of tumour-infiltrating lymphocytes (TILs) [117,118,119]. Interestingly, T_regs_ found in the blood, or recruited to non-cancerous inflamed tissue, lack two co-expressed proteins on the cell membrane of T_regs_ that infiltrate tumour tissues as a TIL subpopulation, viz. IL-1 receptor type 1 (IL-1R-1) and inducible T cell co-stimulatory molecule (ICOS), and these regulatory cells have different gene expression profiles.

It has recently been found that these intra-tumoral IL1R1^+^ T_reg_ cells respond to antigens and are clonally expanded with a superior suppressive function compared with IL1R1-negative T_regs_ [120]. T_regs_ cells exert their immunosuppressive function through their ability to suppress or downregulate the induction and proliferation of effector T cells, such as CD3^+^CD4^+^ and CD3^+^CD8^+^ T cells, B cells, CD56^dim^/CD16^bright^ NK cells and macrophages (M1/M2 phenotype TAMs/MФ) and dendritic cells (DCs); they are also able to participate in T_reg_-dependent peripheral immunotolerance primarily by producing key cytokines and immune-suppressing molecules, such as interleukin IL-10, IL-35 and transforming growth factor-β (TGF-β). T_regs_ can also suppress the immune system by cytolysis via granzyme-A/B-dependent and perforin-dependent killing of target cells, suppress effector T cells by metabolic disruption via depletion/consuming of IL-2, and suppress the immune system by modulation of dendritic-cell (DC) maturation or function [121,122]. Importantly, the suppressive effect of the CD4^+^CD25^+^Foxp3^+^T_reg_ subpopulation is also associated with the high expression of the CTLA-4 (cytotoxic T-lymphocyte-associated protein 4) molecule, which binds to co-stimulatory proteins CD80(B7-1)/CD86(B7-2) on antigen-presenting cells (APCs), leading to the inhibition of CD4^+^ T helper cells and CD8^+^ T cytotoxic effector lymphocytes (CTLs). Interestingly, intra-tumoral CD4^+^CD25^+^Foxp3^+^T_regs_ exhibit a more immunosuppressive effect than circulating regulatory T cells, as evidenced by the increased expression of immune checkpoint molecules [123]. Recent studies support the importance of the identified CD4^+^CTLA-4^+^CD25^high^T_reg_ subpopulation with high levels of T cell immunoglobulin and mucin domain-3 (TIM-3), programmed cell death 1 (PD-1), lymphocyte activation gene-3 (LAG-3), glucocorticoid-induced tumour necrosis factor receptor family-related protein (GITR) and OX40 (CD134) [124].

TIM-3^+^ T_reg_ are functionally and phenotypically distinct in HNSCC TIL, and are highly effective at inhibiting T cell proliferation despite high PD-1 expression. Increased IFN-γ expression induced by anti–PD-1 immunotherapy may be beneficial by reversing TIM-3^+^ T_reg_ suppression. From the clinical viewpoint, a TIM-3^+^ T_reg_ phenotype is related to more immunosuppressive activity compared to T_regs_ with a low level of TIM-3 expression. A significant reduction of the CD4^+^CD25^+^Foxp3^+^T_reg_ subpopulation was noted in HNSCC patients who underwent immunotherapy with anti-PD-1 monoclonal antibodies [123,125]. Moreover, it was demonstrated that T_regs_ can also dampen the response to radiation therapy in HNSCC individuals by creating an immune-inhibitory microenvironment. It has been proposed that a combination of inhibitors of signal transducer and activator of transcription 3 (STAT3) with radiation may improve tumour growth delay and reduce T_regs_ activity, and increase myeloid-derived suppressor cell (MDSC) stimulation, M2 macrophages and enhanced effector T cells and M1 macrophages [126].

While several studies have demonstrated that CD4^+^CD25^+^Foxp3^+^T_reg_ may contribute to the progression of HNSCC and have a negative prognostic impact in HNSCC patients [127,128,129,130], several others presented contradictory results on the prognostic value of CD4^+^CD25^+^Foxp3^+^T_reg_ TILs. In non-HPV-associated HNSCC tumours, a better OS was noted in patients with high Foxp3^+^ TILs (HR 0.80; 95% CI: 0.70–0.92) and similar dependences were seen for DFS (HR 0.77; 95% CI: 0.57–1.02) and LRC (HR 0.92; 95% CI: 0.81–1.04) [70,71]. Unfortunately, only one study reported a slightly and non-significantly better OS for HPV-positive patients with high Foxp3^+^ TILs for the HNSCC patient group [131]. A possible explanation for this unexpected phenomenon is that being regulatory cells, CD4^+^Foxp3^+^T_regs_ may inhibit the active inflammatory process, the production and action of growth factors and pro-inflammatory cytokines directly related to promoting tumour growth; they may also significantly support the mechanisms related to the activity of CTLs and induce a high CD8^+^/Foxp3^+^ ratio in HNSCC [132,133,134]. Other important factors include the high heterogeneity of the T_reg_ lymphocyte subpopulation, the expression of surface immune-biomarkers, i.e., CD25^+^, but not Foxp3, a marker not specific for activated regulator cells, and the absence of the CD127 molecule [132].

A recent study proposed that T_regs_ could be also divided into functional subsets based not only on the expression of Foxp3, but also the CD45RA antigen. The research indicated that a high frequency of CD45RA^−^Foxp3^high^ T_reg_ correlated with a poor prognosis in HNSCC patients, and a low frequency was confirmed in individuals before treatment who later showed a better clinical outcome, even in cases with an advanced stage of cancer. Interestingly, CD45RA-Foxp3^high^ T_reg_ counts decreased after intensive treatment, although their counts returned in the early stages of recurrent cases, even before clinical symptoms [129]. Recent studies also support the role of intra-tumoral T_reg_ cells preferentially expressing the neuropilin-1 (NRP1) molecule; human T_regs_ adopt a transient activation state driven by continuous T cell receptor (TCR) signalling through the mitogen-activated protein kinase (MAPK) pathway and IL-2 exposure. It was reported that the prevalence of NRP1^+^ T_regs_ is associated with reduced progression-free survival in head and neck cancer [135,136]; however, conversely, some studies indicate that high levels of Foxp3^+^ T_reg_ infiltration in HNSCC were associated with longer recurrence free survival (RFS) and overall survival (OS), which was associated with the presence of high levels of CD4^+^CD25^+^Foxp3^+^T_reg_ in TILs. This indicates an ongoing strong anti-tumour immune response contributing to the inhibition of cancer growth [132,137].

##### Tumour-Associated Macrophages (TAMs/MФ)

Tumour-associated macrophages (TAMs), i.e., mature macrophages, constitute a critical immunosuppressive population of TME cells that facilitate tumour progression, angiogenesis and suppression of the anti-tumour immune response by supporting effective tumour immunity, cell growth and proliferation, local invasion, nodal/distant neoplastic spread. TAMs show a high expression of PD-1 ligands, i.e., PD-L1 and PD-L2, leading to inhibition of T lymphocyte function. TAMs subsequently can polarize in vivo/in vitro towards two types of phenotype, i.e., CD68^+^ M1 and alternatively activated CD68^+^ M2 cells. These types are characterized by a considerably different types of activation, receptor expression, function and cytokine and chemokine production, and can change their functional phenotype in response to changes in microenvironmental influences. The pro-inflammatory, classically activated M1 phenotype TAMs are stimulated by the T helper type 1 (Th_1_) cytokine interferon-γ (IFN-γ). While the anti-inflammatory M2 phenotype TAMs adopts an immunoregulatory role following T helper type 2 (Th_2_) stimulation, and due to their ability to produce immune-inhibitory factors, they are involved in tissue remodelling, angiogenesis and tumour progression [57,122,138,139]. M1 TAMs secrete pro-inflammatory cytokines, such as IL-12, IL-23 and TNF-α, MMP-9 and increased levels of interferon-(INF-γ)-inducible chemokines (CCL2, CCL5, CXCL9, CXCL10 and CXCL16), which induce PD-L1 expression on macrophages and participate in the anti-tumour immune response by contributing to the Th1 activity, by inhibiting proliferation and by exerting cytotoxic ability. Immunocompetent M1 cells play an important role in anti-cancer immunity by enhancing the Th_1_-mediated immune response, which inhibits proliferation and exhibits cytotoxic effects. The M2-like phenotype is induced by Th_2_ cytokines, such as IL-4, IL-10 and IL-13 and is characterized by increased secretion of anti-inflammatory cytokines, such as IL-1 receptor antagonist (IL-1ra), IL-10 and TGF-β, peroxisome proliferator-activated receptor γ (PPARγ), vascular endothelial growth factor (VEGF), arginase-1 (Arg-1) and IDO, which inhibit the activity of CD4^+^ and CD8^+^ Th lymphocytes, and encourage CD8^+^ T cell tolerance [57,122,138,139]. TAMs may also promote CD4^+^CD25^+^Foxp3^+^T_regs_ expansion and lead to metabolic starvation in T cells. TAM blockade enhances the anti-tumour immunity of CD8^+^ CTLs and augments the efficacy of the PD-1 inhibitor [140,141]; it may also inhibit pro-tumour immune evasion, enhance the anti-tumour immunity of CD8^+^ CTLs and augment the efficacy of PD-1 inhibitor [142,143]. Recent studies have confirmed that high levels of infiltration of TAMs in the TME was significantly correlated with a poor outcome in HNSCC patients; therefore, TAM ratio could be used as a potential prognostic parameter [144,145,146].

##### Myeloid-Derived Suppressor Cells (MDSCs)

Myeloid-derived suppressor cells (MDSCs), i.e., immature myeloid CD34^+^ cells, constitute an important heterogenous immunocompetent cell population of the immune suppressive intratumoral niche described in neoplastic disease, and are characterized by a strong immunosuppressive effect in both the adaptive and innate immune systems. MDSCs constitute the leading tumour promoters of carcinogenesis and are a hallmark of cancer immune evasion mechanisms [147,148,149,150,151]. Two subgroups of MDSCs have been identified based on their histological profile and cell-surface expression of specific antigens and various myeloid markers, comprising CD11b, CD33, CD14, CD15 and CD16, but lacking HLA-DR expression. Human MDSCs are generally defined as immune cells with CD3^−^, CD19^−^, CD56^−^, CD11b^+^, CD33^+^ and HLA-DR^−^ antigenicity. The granulocytic subset has a CD11b^+^LY6G^+^LY6C^low^ phenotype and the monocytic subset has a CD11b^+^LY6G^−^LY6^Chi^ phenotype [148,152,153]. Most importantly, MDSC expansion is driven by inter alia macrophage migration inhibitory factor (MIF), cyclooxygenase 2 (COX2), prostaglandins, PGF_2_ (PGLs), stem cell factor (SCF), M-CSF, IDO1, IL-6, IL-1β, IL-4, IL-13, and TNF-α, granulocyte/macrophage CSF (GM-CSF) and vascular endothelial growth factor (VEGF) [147,154,155,156,157,158]. These factors can activate signalling pathways in MDSCs through JAK2/STAT3 (Janus kinase protein family 2 member/signal transducer and activator of transcription 3), STAT6, nuclear factor-κB (NF-κB) and STAT1 or Notch signalling, which are key molecules involved in MDSC cell survival, proliferation, differentiation and apoptosis. Importantly, the exclusion of STAT3 or STAT5 signalling inhibits tumour-infiltrating MDSC activation and T cell-dependent neoplastic growth [147,156,157,158,159,160,161,162]. Immunosuppressive functions of MDSCs are related to the production and secretion of immunosuppressive cytokines, such as IL-10 and TGF-β1 and antigen-specific or antigen-nonspecific suppression of T cell responses. MDSCs can inhibit CD3^+^CD4^+^ T cells, which secrete proinflammatory factors, such as IFN-γ, IL-2, IL-6, IL-17 and TNF-α, by inhibiting IFN-γ and promoting the production of the anti-inflammatory factors, such as IL-10 and TGF-β1. MDSC-mediated immune suppression has been mainly related to the upregulation of arginase (Arg1), nitric oxide synthase (iNOS), peroxynitrite activity, reactive oxygen species (ROS) release and programmed death receptor ligand 1 (PD-L1) expression; this ultimately represses Th_1_ and Th_17_ cell activation and proliferation and causes T cell apoptosis [147,156,157,158,163]. Both iNOS and Arg1 catabolize L-arginine, resulting in arginine starvation and T cell suppression. L-arginine starvation downregulates CD3 ζ-chain in TCR complexes and causes cell cycle arrest of antigen-activated T cells and also acts via inhibiting CD3/CD28 expression on T cells, leading to the suppression of an antigen-specific and non-specific T cell response. In addition, hypoxia-inducible factor 1-alpha (HIF-1α) may affect L-arginine metabolism in MDSCs, by inducing iNOS and Arg1 and enhancing the suppressive function of MDSCs in cancer patients. Moreover, MDSCs can suppress dendritic cells by reducing MHC-II and CD86 expression, inhibit B cells through NO and PGE_2_, and promote Th_17_ cell responses by secreting IL-1β and IL-17 [147,152,158,164].

Interestingly, innate γδT_17_ cells may mobilize MDSCs into the tumour microenvironment to elicit immunosuppression [165]. In vitro treatment with MDSCs was also found to decrease the cytotoxicity of natural killer (NK) cells via the NKp30 receptor but increase the transcription factor Foxp3^+^ in CD4^+^ regulatory T cells. Additionally, MDSCs can differentiate into TAMs, as well as mature DCs and M2 macrophages [147,166]. MDSCs are also responsible for degrading the ECM via production of significant levels of matrix metalloproteinases (MMPs), especially MMP-9, and by the formation of premetastatic niches [122,167]. Importantly, studies in cancer patients indicated that MDSC activity has been associated with low survival rates and higher tumour recurrences [168,169,170,171].

Since MDSCs support tumour growth by suppressing antitumour immune responses and/or inducing immunosuppressive cells, they have become a potential therapeutic target for current cancer treatment strategies [152,172,173,174,175,176]. Therapeutic trials targeting MDSC have so far shown promising results in preclinical studies and are still being extensively investigated in clinical trials, most often in combination with various immune checkpoint inhibitors (ICIs) [166,172,173,174,175,176]. The main approaches to targeting MDSCs include depleting MDSC populations (chemotherapeutics gemcitabine, 5-FU, paclitaxel and doxorubicin, peptibodies; CD33 targeted APC—gemtuzumab, ozogamicin) and preventing MDSC recruitment and migration to the TME; other strategies are based around attenuating the immunosuppressive mechanisms of MDSCs by downregulating the expression of ARG1, iNOS and COX-2, reducing ROS generation, inhibiting MDSC immunosuppression (CCL2i antibodies—cralumab; CXCR2i inhibitors—reparixin; CCR5i inhibitors—maraviroc; CSF1Ri inhibitors—IMC-CS4, GW2580, plexidartinib, emaktuzumab; VEGFRi inhibitors, tyrosine kinase inhibitors—sunitinib; Arg/IDOi inhibitors, PDE5i inhibitor—sildenafil, tadalafil; COX-2i inhibitor—celecoxib), promoting the differentiation of MDSCs into mature non-suppressive myeloid cell-like macrophages and dendritic cells (STAT3i inhibitors—AZD9150, TATRA, TLR agonists, vitamin D3) [147,177,178]. Such strategies include therapies that enhance cell-based anti-cancer therapeutics by stimulating an increase of CD56^bright^ NK cell cytotoxicity by blocking MDSC-mediated immunosuppression [179,180,181]. These anti-cancer therapies include the anti-KIR2DL-1, -2 and -3 antibody IPH2102/BMS-986015 (lirilumab), the anti-NKG2A antibody IPH2201 (monalizumab), and the anti-CD16 innate cell engager AFM13 [181]. Researchers are also examining strategies that target MDSCs with the objective of enhancing the anti-tumour efficacy of ICB therapy. Currently, multiple studies have confirmed that MDSCs induce resistance to immune checkpoint receptor blockade (ICIs) therapy via their immunosuppressive mechanisms in the TME. In addition, non-immunosuppressive mechanisms of MDSCs, such as promotion of angiogenesis and induction of cancer stem cells also affect pro-tumour progression [147,176,182].

##### Tumour-Associated Neutrophils (TANs)

Tumour-associated neutrophils (TANs), also known as polymorphonuclear leukocytes (PMNs), represent another type of immune cell that accounts for up to 70% of circulating leukocytes and infiltrates the TME of many human cancers; by doing so, it regulates innate and adaptive immunity, thus playing an important role in mediation of T cell independent antibody responses, as well as antigen presentation and T cell activation [183,184,185,186]. The identification of TANs is based on the expression of specific surface markers, such as CD11b, CD14, CD15, CD16, CD62L and CD66b [187,188,189].

Due to differences on the roles of neutrophils in cancer development, TANs are divided into N1 TAN (mature/tumour-suppressive) or N2 TAN (immature/tumour-promoting) phenotypes; these are also called high-density neutrophils (HDN) or low-density neutrophils (LDN), respectively. The N2-polarized neutrophils are morphologically similar to granulocytic or polymorphonuclear MDSCs (PMN-MDSCs) and are triggered by the B-cell activating factor to inhibit the immune response in HNSCCs. As such, this may favour the progression of carcinogenesis [186,190,191,192]. Consecutively, N1 TANs mainly promote immune responses, while N2 neutrophils exhibit immunosuppressive effects. Interestingly, the neutrophil phenotype in the TME is related to tumour type and stage of neoplastic disease. N1 TANs are typical in the early stages of cancer, but as the tumour progresses, they assume the immunosuppressive phenotype of N2 TANs [193,194,195]. The N1 TANs are thought to act as phagocytic cells by producing lytic enzymes and reactive intermediates (ROS/RNS), the formers of neutrophil extracellular traps, called NEToses (NETs); they do so by releasing their cytotoxic cytosolic and granule proteins on a scaffold of decondensed chromatin in a cell death process. They also induce the inflammatory response by secreting many cytokines, chemokines and anti-tumour factors. However, under certain conditions, N2 TANs can also produce reactive oxygen species, stimulating pro-tumour mechanisms and thus supporting carcinogenesis, tumour growth and metastasis [183,185,196,197,198,199,200,201,202,203]. N1 TANs are potent anti-tumour effector cells that may also recruit other cells with anti-tumour activity. Indeed, several studies have confirmed that TANs could lyse neoplastic cells via antibody-dependent cell-mediated cytotoxicity (ADCC) in the presence of anti-tumour antibody. N1 phenotype neutrophils also use other mechanisms that may prevent tumour development and further metastasis. For instance, research reported that N1 TAN cytotoxicity is Ca^2+^ ions-dependent and is mediated by TRPM2, a ubiquitously expressed H_2_O_2_-dependent Ca^2+^ channel. Molecule TRPM2 mediates neutrophil killing of disseminated tumour cells [203,204]. Numerous studies also described the various cross-talks between N1 TANs and other tumour-infiltrating immune cell types, emphasizing the active role of neutrophils as regulators of the immune mechanisms, i.e., N1 cells are responsible for leukocyte recruitment and enhance proliferation of T cells [185,204]. In addition, N1 TANs may also suppress tumour cell proliferation via Fas/FasL (Fas ligand) pathway, activation of caspase cascade mediated cell cycle arrest and apoptosis of cancer cells [205]. Since N1 TANs also release matrix metalloproteinases (MMP-8) and reduce the activation of β-integrins, they may promote tumour suppression through its effect on leukocyte migration and inflammation and also display anti-metastatic activity [204].

Recent studies introduced that NETs production has bidirectional effects, both in carcinogenic and anti-carcinogenic mechanisms, by immune regulation and cell behaviour modulation. Unfortunately, it is not currently clear which effect is dominant in HNSCCs and other human cancers [205,206]. However, over the years, neutrophils have been shown to demonstrate functional plasticity, driven by multiple factors present in the TME and display both pro-tumour and anti-tumour effects, emphasizing an unexpected cellular heterogeneity in cancer [185,207,208,209,210]. Moreover, the tumour microenvironment also controls neutrophil recruitment and in turn TANs help tumour progression.

Indeed, it has been proven that N2 TANs can produce NETs to participate in the progression of oropharyngeal carcinoma [205,206]. NETs consist of a chromatin backbone, which acts as a carrier for antimicrobial and anti-antigen peptides and toxins, which are thrown out to attack neutrophils. NET deposition facilitates tumour cell growth and proliferation, immunosuppression of the cellular components of the cancer milieu and cancer-associated thrombosis. Recent research indicates that the neutrophils isolated from oropharyngeal carcinoma have a strong ability to produce NETs. This phenomenon intensifies the accompanying characteristic changes of components, such as citrullinated histone H3 and myeloperoxidase (MPO), which stimulate the immune cells to secrete TNF-α and induct changes in both stroma and epithelial cells, via oral malignant transformation and tumour matrix degradation downstream of TNFα/TNFR1 [187,205]. Moreover, NETs favour the metastasis by contributing to EMT and enhance the potent migratory and invasive abilities of neoplastic cells. Circulating NETs capture circulating cancerogenic cells and increase vascular permeability, thereby promoting tumour cell intravasation and micrometastasis. Moreover, accumulating evidence confirms that TANs may also help in tumorigenesis initiation and cancer progression. It is well known that TANs are attracted by CXCR2 ligands and infiltrate the tumour microenvironment to become intra-tumoral immune cells. They mainly support cancer development through three pathways: by supporting cancer initiation through the secretion of reactive oxygen species, reactive nitrogen species and some proteases, by facilitating metastasis through the suppression of natural killer cells, enhancement of extravasation, and by escorting tumour cells to spread and seed. The TANs also support tumour growth by inhibiting CD3^+^CD8^+^ T cells and promoting angiogenesis due to VEGFA secretion; this is connected with dampening the CTL response via arginase-1 (Arg-1), upregulation of cellular proliferation through a neutrophil-secreted neutrophil elastase (NE), degradation of the basement membrane and ECM via NE, cathepsin G and matrix metalloproteinases (MMP-8/9), chemokine CXCR4, as well as the upregulation of angiogenesis by VEGF and HGF, oncostatin-M (OSM) and ICAM-1 dependent tumour intravasation, immune protection in circulation, secretion of PGE2 for tumour progression and extravasation into distant, metastatic tissue milieu [201,211,212,213,214].

As TANs play such an ambiguous role in cancer, a deep knowledge of the mechanisms of TAN functions in tumorigenesis is necessary to define appropriate therapeutic strategies that can induce and maintain an anti-tumour microenvironment. The available literature provides four categories for strategies targeting TANs and granulocytic/polymorphonuclear myeloid-derived suppressor cells (G/PMN-MDSCs): depletion of existing TANs/PMN-MDSCs, blockade of the development of TANs/PMN-MDSCs, blockade of TANs/PMN-MDSC recruitment, inhibition of immunosuppressive function [215,216,217,218]. For example, the molecule with the most promising benefits is a selective and reversible antagonist of CXCR1/2 named SX-682, which is currently in non-randomized interventional phase I clinical trials (Syntrix Biosystems Inc., clinicaltrials.gov identifier: NCT03161431). The blockade of these molecules consequently leads to the inhibition of CXCR1/2 activation by chemokines secreted by neoplastic cells, resulting in a decrease of neutrophil recruitment to the TME. Subsequently, this led to a decrease in inflammatory phenomena in the tumour milieu and immunosuppressive effects of TANs with a decrease in cancer growth and invasion [218,219].

In addition to TAN’s diverse functional properties, the neutrophils present in the bloodstream or TME are also indicated as biomarkers in several solid tumours. Indeed, increasing clinical evidence shows that an elevated amount of peripheral blood neutrophils may be a promising prognostic marker associated with a poorer overall outcome and progression-free survival [204,220,221].

##### Carcinoma-Associated Fibroblasts (CAFs)

Cancer-associated fibroblasts (CAFs) are elongated, spindle-shaped activated fibroblasts characterized by some positive markers, such as alpha-smooth muscle actin (α-SMA), fibroblast activation protein (FAP) and fibroblast specific protein 1 (FSP-1), as well as caveolin-1 (CAV-1), FSP-1, PDGFR-α, PDGFR-β and Thy-1 [222,223]. CAFs can also be characterized by the absence of epithelial and endothelial markers, such as CD31 and cytokeratin [122,224,225,226,227]. Two subsets of CAFs—senescent CAFs (SA β-Gal/p16^INK4A high^) and non-senescent (SA β-Gal/p16^INK4A low^) phenotypes—constitute the most abundant subpopulation of immune-active cells in the tumour microenvironment reported to support carcinogenesis by stimulating tumour initiation, angiogenesis, cancer cell growth and proliferation, as well as tumour invasion, progression and metastasis in most solid tumours, including HNSCC. Importantly, senescent CAFs proliferate less but are thought to be more tumour supportive than non-senescent ones [122,224,225,226,227,228]. Studies have shown that CAFs could be responsible for the pro-tumorigenic milieu. On the contrary, some studies indicate that CAFs play a tumour-suppressive role in certain cancer models. Recent observations from genetically engineered in vitro models and clinical studies in the cancer niche indicate that at least two functionally different populations of CAFs exist, viz. cancer-promoting CAFs (pCAF) and cancer-restraining CAFs (rCAF) [229,230,231,232,233,234,235,236]. Due to their considerable immunogenic activity, CAFs play a key role in the proliferation, invasion and metastasis of various origin tumours [237,238,239]; however, fibroblasts have also been found to have an inhibitory effect on cancer cell proliferation [229,230].

For many human cancers, the presence of CAFs is related to higher malignancy as they interact with neoplastic cells and other components of the TME to shape a tumour-supportive environment. Modification of the TME conducive to neoplastic development is connected with the secretion of important CAF-secreted pro-neoplastic proteins, enzymes, cytokines, metabolites and pro-inflammatory factors that have a functional impact on head and neck cancer cells. These include interleukins IL-6, IL-8 and IL-17A, TGF-β1 and 2, MCP1, Ly6c, COX2 and PGE2, epidermal growth factor (EGF), hepatocyte growth factor (HGF), vascular endothelial growth factor (VEGF), brain derived neurotrophic factor (BDNF), periostin, microfibrillar-associated protein 5 (MFAP5), chemokine ligands, i.e., CXCL1, CXCL12, CXCL14, CCL2, CCL5, CCL7, granulocyte-macrophage colony-stimulating factor (GM-CSF), fibroblast activation protein (FAP), fibroblast-specific protein-1 (FSP-1), platelet-derived growth factor receptor α/β (PDGFR α/β) and vimentin. These immunologically active ligands promote cancer cell growth and proliferation, migration, invasion, angiogenesis phenomena and various inflammatory processes, which are critical for carcinogenesis and even drug resistance [122,224,225,226,232]. Consequently, the activity of CAFs is associated with many crucial pro-neoplastic mechanisms, including (1) induction of ROS and subsequent activation of PI3K/AKT/mTOR pathway in cancer cells and CAFs, (2) increase in cyclin D/E and inhibitory cyclin-dependent kinase 4 (CDK4) activity by CCL2, (3) interaction with oncogenic tyrosine-protein kinase Met (c-met) and phosphorylation of signal transducer and activator of transcription 3 (STAT3) via a JAK1/2-dependent route and stimulation of nuclear factor NF-ĸB(p65) in cancer cells, (4) activation of HNSCC cell growth and migration via activation of MAPK and AKT pathways, (5) induction of the expression of key glycolytic enzymes and lactate efflux by HGF and (6) activation of tyrosine-protein kinase-like 7 (PTK7) and Wnt/β-catenin signalling in cancer cells by periostin, as well as affect immune cells via: (7) induction of CD4^+^ and CD8^+^ T cell apoptosis, (8) increase the proportion and migration of the regulatory T cells (CD4^+^CD25^+^Foxp3^+^T_regs_) that inhibit the T cell-dependent anti-tumour response, and (9) attract and induce monocyte differentiation [122,224,225,226,240,241,242,243,244].

Importantly, CAFs also play an important role in modulating the microenvironment by remodulating and degrading the TME and contribute to epithelial to mesenchymal transition (EMT) via the transglutaminase 2-dependent IL-6/IL6R/STAT3 axis, matrix-metalloproteinase (MMP) production, i.e., MMP-2, MMP-3; they also increase vimentin expression and downregulate the adhesion molecules e-cadherin, desmoglein 1 and 3, desmoplakin and desmocollin, which results in the promotion of the invasive phenotype of cancer cells in HNSCC. CAFs may also contribute to bone resorption and infiltration in head and neck cancer by activating osteoclasts through the interaction of NF-ĸB(p65) receptor activator (RANK) and its ligand secreted by osteoblasts (RANKL). Furthermore, CAF-secreted exosomes carry various microRNA that change cancer cell behaviour, i.e., CAF-exosomes are rich in miR-196a, which targets *ING5* and *CDKN1B* mRNA, resulting in the reduction of ING5 and p27^KIP1^, CDK2/4 and cyclin D1/E activity, respectively, and they are deprived of miR-34a and miR-3188 that leads to the activation of AKT/β-catenin and Bcl-2 and their downstream pathways [122,224,225,226,245].

Importantly, the identification of cancer-promoting CAFs (pCAF) and cancer-restraining CAF (rCAF) subpopulations may contribute to the development of new diagnostic and therapeutic methods. Unfortunately, their phenotypic and functional heterogeneity makes clinical application difficult [246,247]. It has been reported that CAFs may create a complex signalling network to promote drug resistance in neoplastic cells after drug treatment. They form a survival niche to achieve tumour formation and chemoresistance by sustaining cancer stemness [147,248]. This comes about by inter alia phosphorylation and acetylation of activated NF-κB(p65), expression of wide panel of exosomal miRNAs, i.e., miR-214, miR-92a-3p, miR-196a, miR-3188, cytokine IL-8 and IL-11 secretion; such activity induces STAT3 phosphorylation and increases the expression of anti-apoptotic proteins Bcl-2 and survivin, and results in the release of exosomes containing the chemoresistance inducing transcription factor Snail, thus downregulating E-cadherin and inducing TME. These mechanisms also protect cancer cells from cisplatin-induced apoptosis, thereby promoting chemoresistance [249,250,251,252,253].

The recent advances in CAF-based therapy are based on targeting the markers to ablate CAFs, restoring activated CAFs to quiescent ones, and blocking the signalling between CAFs and tumour cells, such as JAK1/JAK2 and CXCL12/CXCR [147,254,255,256]. For instance, the use of a JAK inhibitor (ruxolitinib) and DNMT inhibitor (5-azacytidine) combination could restore the fibroblast phenotype and reverse the pro-invasive activity of CAFs in lung cancer and head and neck carcinomas (HNSCCs). The results clearly show that patients treated with ruxolitinib had longer overall survival and better prognosis, supporting the potential clinical benefit of JAK1/JAK2 inhibitor [147,254,255,256]. Moreover, from the clinical point of view, several studies also indicate that the presence of cancer-associated fibroblasts in the tumour microenvironment could not only secrete a variety of pro-neoplastic factors to facilitate tumour growth and immunosuppression mechanisms, but also they may favour chemotherapy resistance and extend the viability of HPV*^−ve^* cancer cells in squamous cell carcinomas of the oral cavity, pharynx and larynx after cisplatin treatment. Furthermore, CAFs may also promote the proliferation of HPV*^+ve^* and negative tumour cells after cetuximab and mTOR inhibitor treatment [257,258,259,260].

##### Natural Killer Cells (NK)

Natural killer cells (NK cells) are population of tumour-antagonizing immune cells that mediate the immunosurveillance of the neoplasm and constitute large granular CD3^−^CD56^+^ lymphocytes that can be classified into two subpopulations, depending on their expression of surface markers CD16 and CD56 and they do not require prior sensitization or stimulation for their effector function. Based on CD16 and CD56 expression levels, they are classically distinguished in two subsets. The first subset, CD56^dim^/CD16^bright^ NK cells, constitutes approximately 90% of all peripheral NK cells, and is responsible for high natural cytotoxicity and the ability to quickly detect and kill virus-infected or malignant cells. This subpopulation within TME is similar to the CD8^+^ T cells and expresses high levels of the killer-cell immunoglobulin-like receptor (KIR), which recognises HLA class I, the maturation marker CD57. It also promotes natural and antibody-dependent cellular cytotoxicity, exhibiting high levels of perforin/granzyme B and IFN-γ, which activates and promotes infiltration of Th_1_ cells and MDSCs; this enhances killing and thus limits primary tumour growth [122,183,261,262,263,264]. The second subset, CD56^bright^/CD16^dim^, is characterized by a higher expression of immunomodulatory cytokines. These cells are characterized by the heterogenic NKG2-A/C type II lectin integral membrane protein (NKG2A and NKG2C), low levels of perforin and granzyme B, and are primarily specialized for the production of pro-inflammatory cytokines and chemokines, such as IFN-γ, TNF-α, GM-SCF, IL-5, IL-8, IL-10, IL-13, CCL2, CCL3, CCL4, CCL5 and CXCL10. This allows their preferential recruitment to secondary lymphoid organs, tumour milieu and inflamed tissues to promote anti-tumour activity [122,183,262,263,264,265,266,267,268,269].

The anti-cancer activity of NK cells is closely related to stimulation via signals from two different types of receptors present on the cell surface—activating and inhibiting receptors. MHC class I molecules that are typically found on non-cancerous cells act as inhibitors of NK cell activation. Malignant or virus-infected cells downregulate the MHC-I molecule to escape cytotoxic T cells; however, this phenomenon facilitates the recognition of pathological antigens by natural killer cells. NK cell activation leads to targeted cell apoptosis induction, exocytosis, secretion of perforin and granzymes, expression of Fas ligand (FasL), TNF-associated apoptosis-inducing ligand (TRAIL), or antibody-dependent cellular cytotoxicity (ADCC), which is initiated after the binding of the Fc portion of an IgG antibody to Fc-γRIII/CD16 receptor [39,122,183]. Moreover, some research has examined the involvement of long noncoding RNAs (lncRNAs) in NETosis, the process of neutrophil extracellular trap (NET) formation in HNSCC. Research indicates that the NETosis-related lncRNA signature inclusive of specific lncRNAs facilitate patients to be classified as high-risk and low-risk groups of a good or poor prognosis [270,271,272,273].

Recent research also points to the role of a distinct subpopulation of NK-like cells called NK T cells (NKT). NKT cells are a lineage of lymphoid cells that share the morphological and functional characteristics of T cells and NK cells. They are defined by the expression of both T cells and NK cell surface markers and they are a subset of CD1d-restricted T cells at the interface between the innate and adaptive immune system. Importantly, NKT cells require prior priming to develop antigen-specific immune memory [274,275,276]. There are two major types of NKTs: Type I NKTs (NKTIs) and Type II (NKTIIs) cells. Type I NKT cells express a semi-invariant T cell receptor and have the ability to recognize the common lipid prototype, α-galactosylceramide (α-GalCer). In contrast, Type II NKT cells are T cells characterized by limited lipid-specific recognition of CD1d and hence use other receptors, but do not recognize α-GalCer [39,274,276]. Both NKT subpopulations play important regulatory roles in anti-cancer immunity. NKTIs are typically anti-tumour and promote anti-cancer immunity. Most NKTIIs cells are predominantly pro-tumour and promote the activity of other cells, including regulatory T cells (T_regs_) and myeloid-derived suppressor cells (MDSCs), and may induce MDSCs to secrete TGF-β, one of the most immunosuppressive cytokines known [39,274,276]. Moreover, NKT cells constitute active immune regulators, since they can regulate immune responses toward both inflammation and tolerance by secreting either T helper (Th_1_, Th_2_, Th_17_, T regulatory T_reg_), or follicular helper (T_fh_) cell-associated cytokines. Th_1_-like NKT cells have the potential to induce an anti-tumour response while Th_2_- and T_reg_-like NKT cell subsets facilitate immune escape and tumour progression. This can result in overstimulation of NKT cells during neoplastic growth that can lead to the induction of anergy and the dominance of Th_2_- and T_reg_-like subpopulations, thus facilitating tumour progression and immune escape [39,274,276]. Interestingly, another subset of NKT cells, the invariant natural killer T cells (iNKT cells), express a highly restricted invariant aβ T cell receptor (aβTCR), and low levels of these iNKT cells in peripheral blood predict a poor outcome in HNSCC patients [277,278,279]. Several studies have also shown that activation of IFN-γ-producing NKT cells may activate the iNOS^+^ M1-macrophages and reduce the iNOS^−^ M2-macrophages. They may also trigger the effector Th_1_ cells and CD8^+^ T cells, as well as favour the conversion of Th_2_ to Th_1_ in the secondary lymphoid tissues and tumour microenvironment, thus inhibiting cancer growth. These findings suggest that activation of NKT cells may provide an effective anti-cancer outcome [280,281,282].

Importantly, in HNSCC, tumour-infiltrating NK cells often occupy the CD56^bright^/CD16^dim^ subset; these showed an immature phenotype predominantly characterized by the decreased expression of DX5, CD27^low^CD11b^low^, increased NKG2A and lower levels of Siglec-7, NKG2D and natural cytotoxicity receptors, such as NKp30, NKp44, NKp46, perforin, granzyme B and CD16. Unfortunately, this dysfunctional phenotype is associated with a decreased ability to perform antibody-dependent cellular cytotoxicity (ADCC) [283]. Nevertheless, literature data indicate a wide range of tumour immune escape strategies from NK cell-dependent immunosurveillance. Cancer cells can evade the NK cell-dependent anti-cancer immune response by several mechanisms: changes in DNA modifying enzymes, such as HDAC or microRNA involved in epigenetic gene regulation and suppression of MHC class I polypeptide–related sequence A/B (MICA/B) expression; the persistent ligand expression and sustained triggering of NKG2D, leads to lower NK cell activity and reduced ADCC due to reduced NKG2D expression and reduced IFN-γ production; tumour release of cytokines, such as TGF-β and IFN-γ, which inhibit MICA/B expression, downregulate NKG2D and IFN-γ production by NK cells, which promotes the conversion of CD4^+^ T cells into regulatory T cells (CD4^+^CD25^+^Foxp3^+^T_regs_), which inhibit immune surveillance; enhanced expression of matrix metalloproteinase enzymes (MMPs) and ADAMS promotes inhibition of activating ligands, such as MICA, which bind to NKG2D on CD4^+^ T cells and NK cells, leading to suppression of NK cell responses due to NKG2D degradation and conversion of CD4^+^ T cells to CD4^+^CD25^+^Foxp3^+^ regulatory T cells [284].

In the last decade, numerous pre- and clinical studies have been conducted using the innate anti-cancer function of NK cells to treat solid tumours, including squamous cell carcinoma of the head and neck. Unfortunately, there are a number of immune evasion strategies used by neoplastic cells, including inhibition of the anti-tumour effect of NK cells infiltrating HNSCC. Knowledge of the key mechanisms underlying NK cell activation may contribute to improving HNSCC treatment outcomes, e.g., by selecting patients most likely to respond appropriately to NK cell-based immunotherapy [285,286]. Numerous studies indicate that in patients with HNSCC, the number of NK cells is significantly increased, which correlates with improved patient outcomes [287]. Therefore, research into the role of NK checkpoint receptors is gaining great interest in cancer immunotherapy [287,288,289,290,291]. Importantly, signalling via inhibitory immune checkpoints, e.g., CTLA-4, PD-1, TIM-3, KIR2DL-1/2/3, NKG2A, CD96 and TIGIT inhibits NK cell function. Blocking these pathways increases the anti-cancer potential of natural killer cells [84,183]. Indeed, current research focusing on combining PD-1/PD-L1:CTLA-4 axis inhibition with lirilumab, a monoclonal antibody targeting killer cell immunoglobulin-like receptors (KIRs), confirms that these receptors are key inhibitory molecules for NK cytotoxic cell-mediated immunity [292]. Another recent study found that the blockade of the checkpoint receptor TIGIT prevents NK cell depletion and triggers strong anti-cancer immunity [183,293]. Additionally, researchers propose the use of natural killer cells in combination with chemoradiotherapy (CRT) against head and neck cancer to obtain a synergistic anti-tumour effect. The study results indicate that CRT induces NK cell activation ligand (ULBP2) and adhesion molecules (ICAM-1, -2 and -3) on HNSCC, leading to enhanced cytotoxicity of NK cells against HNSCC; this correlated with increased NK cell infiltration and better overall survival in patients with HNSCC [294]. Another study also confirms that CDX2/p300 enhances NK cell-mediated immunotherapy against head and neck squamous cell carcinoma in vitro; the findings indicate that CDX2 stimulated NK cell migration, cytotoxicity and infiltration by upregulating CXCL14 in HNSCC tissues [295]. Interestingly, preliminary evidence also confirms that low numbers of peripheral NK T cells (NKT) are associated with poor prognosis in HNSCC patients, which has been implicated in the suppression of adaptive immunity [277].

##### T Helper 17 Cells (Th_17_)

Human tumour-associated T helper 17 cells (Th_17_) are a subset of pro-inflammatory T helper cells (Th) defined by their production of interleukin 17 (IL-17). A typical feature of these cells is to induce a much stronger anti-tumour response compared to effector and central memory T cells. Furthermore, Th_17_ cells have the ability to maintain a stem cell-like phenotype by activating the HIF-1α/Notch/Bcl-2 axis, also indicating a correlation between Th_17_ activity and improved immune responses to tumour antigens [60]. They also can give rise to distinct Th lineages, determine the function of immunosuppressive regulatory T cells (CD4^+^CD25^+^Foxp3^+^T_regs_) and inhibit T_reg_ differentiation. These regulatory Th_17_ cells can be generated from naïve T cells by stimulation through TGF-β and IL-6 in vitro (Th_17_ cells induced by TGF-β and IL-6 are termed as T_reg_17 cells). Furthermore, IL-21 and IL-23 contribute to Th_17_ formation in animal models and humans [60,296]. Several cytokines, including type I and type II INF, IL-2, IL-4, IL-12 and IL-27 inhibit Th_17_ differentiation and downregulate their function. In the absence of IFN-γ and IL-4, cytokine IL-23 induces naïve precursor cells to differentiate into Th_17_ cells independently of the transcription factors STAT1, T-bet, STAT4 and STAT6. Human Th_17_ cells act by inducing the release of the cytokines IL-8, IL-6, COX-2, MMP-1, MMP-3, CXCL1 and NOS-2 by surrounding cells, such as fibroblasts, macrophages, endothelial and epithelial cells. In addition, Th_17_ cells can produce the cytokines IL-17F, IL-21, IL-22 and IL-26, and express the surface receptors CD161, CCR6, IL-23R and the orphan nuclear receptor (ROR2C) [297]. The production of IL-17 induces the secretion of chemokines and matrix metalloproteinases (MMTs) in the cancer milieu, which promotes the formation and maintenance of inflammation and the recruitment of neutrophils and macrophages to TME. The researchers also indicate the effect of IL-21 expression on tumour growth by enhancing antibody-mediated neoplasm destruction and activating NK and CD3^+^CD8^+^ T cells. However, it has been also proven that IL-21 not only promotes carcinogenesis and tumour growth, but also, under certain conditions, may have anti-cancer effects and inhibit the angiogenesis process. Interestingly, HNSCCs constitute a tumour environment that promotes Th_17_ cell activity due to the release of IL-23 and IL-6 by the neoplastic cells themselves and by TIL and IL-1β by immune cells infiltrating the cancer niche [297]. The summary for different types of immune cells within TME and their related anti- and pro-tumour functions are shown in Table 1.

##### HPV Infection as a Pivotal Modulator of the Immune Response in HNSCC

Large amounts of literature data indicate the importance of HPV infection as a key factor in modulating the immune response to tumour antigens in HNSCC. Despite the prevailing immunosuppressive character, the pattern of the immune cell infiltration markedly differs between HPV-associated and HPV-negative tumours [31,298,299,300,301,302,303]. Although studies on HNSCC confirm that both HPV*^−ve^* and HPV*^+ve^* cancers are characterized by high levels of immune cell infiltration, HPV-positive HNSCC generally have a significantly higher density of tumour-infiltrating lymphocytes (TIL) and are among the immunologically “hottest” of all types of cancer [304,305,306,307]. Furthermore, it has been noted that HPV^+^ neoplasms contain significantly higher densities of CD3^+^, CD4^+^, CD8^+^, CD20^+^ antigens and PD-1^+^ cells compared to HPV^−^ tumours, with a trend towards the increased density of Foxp3^+^T_reg_ cells [308]. This feature was reported to be positively correlated with patient survival in a wide range of solid malignancies [62,298,299]. Moreover, it has also been reported that HPV*^+ve^* HNSCC may typically be immunologically “colder”, with low TIL levels and significantly worse clinical outcomes [309,310,311,312,313]. This condition leads to an immunocompromised niche that favours proliferation and cancer cell resistance to immunotherapy that may occur in HPV-positive HNSCC. Tumour microenvironment (TME) with a confirmed presence of HPV infection can enhance anti-tumour immunity against tumour associated antigens (TAA) and tumour specific antigens (TSA) [314]. In HPV-positive HNSCC, an increase in the infiltration of immunocompetent cells, i.e., T cells, including CD3^+^, CD4^+^, CD8^+^ IFN-γ cytotoxic T cells (CTLs), CD19^+^/CD20^+^ B cells, CD56^dim^/CD16^bright^ activated NK cells, tumour infiltrating antigen presenting cells/dendritic cells (APCs/DCs) and myeloid and plasmacytoid dendritic cells (MDCs), and a decrease of CD4^+^CD25^+^Foxp3^+^ regulatory T cells in TILs is observed, which enhances the stimulation of cellular immunity to tumour antigens [298,304,315]. The interaction between HPV*^+ve^* HNSCC cells and cancer-associated fibroblasts (CAFs) stimulates the generation of chemokines via the IL-1/IL-1R axis, also leading to enhanced chemotaxis of immunocompetent cells [314]. Moreover, HPV^+^ HNSCC exosomes, which carried E6/E7 oncoproteins, p16^INK4a^/CDKN2A (cyclin-dependent kinase inhibitor 2A) and survivin, promoted DC maturation triggering the activation of anti-tumour immune responses, thereby improving outcome in patients with HPV*^+ve^* HNSCC [316]. Conversely, smoking/drinking-related HPV-negative HNSCC is most often characterized by an impaired local and general immune response, dysfunction of immunocompetent cells, i.e., CD3^+^ zeta chain T cells (CD3ζ), CD8^+^ IFN-γ T cells (CTLs), CD45^+^ cells, CD19^+^/CD20^+^ B cells, APCs/DCs and MDSCs, and increased intra-tumour infiltration of regulatory T cells, dysfunctional CD56^dim^ NK cells, increased CD4^+^CD25^+^Foxp3^+^T_reg_/ CD8^+^ T (CTLs) cell ratio and increased T cell exhaustion markers, i.e., PD-1, CTLA-4, TIM-3, LAG-3, IDO-1, KIR and TIGIT [121,304]. Moreover, HPV infection significantly increased T cell infiltration, immune effector cell activation and the diversity of T cell receptors. Notably, HPV-positivity was related to the increased immune cytotoxic function of CTLs and a T cell proinflammatory cytokine expression profile [317]. Interestingly, HPV*^+ve^* HNSCCs increased the expression of both PD-1 ligands (PD-L1 and PD-L2) in the stromal microenvironment, i.e., in both tumour cells (fibroblasts and macrophages) and stroma in situ in treatment-naïve primary HSNCC cell lines; however, no increase was observed for HPV*^−ve^* cells. It was demonstrated that HPV-positive HNSCC cell lines upregulate PD-L1 and PD-L2 expression in an in vitro model via a TLR9-dependent mechanism, whilst this was not observed in HPV-negative HNSCs [318]. Furthermore, HPV*^+ve^* HNSCC patients demonstrated improved outcomes with PD-1/PD-L1 axis blockade, as compared to those with HPV*^−ve^* tumours. These improved outcomes are likely driven to a greater extent by anti-PD-L1 inhibitors [319]. Indeed, metastatic HNSCCs patients receiving anti-PD-1-based therapy were at significantly less risk of death. Recent studies also confirmed that therapeutic use of anti-PD-1/PD-L1 checkpoint monotherapy substantially reduces the risk of death in recurrent or metastatic head and neck cancer (R/M HNSCCs) patients [320,321,322,323]. Thus, the level of expression of PD-L1 may be a useful biomarker for selecting patients with a better response to anti-PD1/PD-L1 monotherapy. The individual features of the tumour-infiltrating immune cell populations and an immune system modulation between HPV*^−ve^* and HPV*^+ve^* HNSCC, are presented in Table 2.

## 2. Materials and Methods

A corpus of studies encompassing a wide range of molecular, observational and intervention studies in humans was searched to determine the role of selected subpopulations of immune cells in the pathogenesis of head and neck squamous cell carcinoma (HNSCC), as well as the regulatory mechanisms of pro- and anti-cancer activity and their impact on immunotherapy. The final search included articles published from January 2010 to January 2023 (conducted on 31 January 2023), all of which are accessible via the PubMed/Medline/EMBASE/Cochrane Library database. The following keywords were used as search criteria: “head and neck neoplasm or HNSCC”, “squamous cell carcinoma of head and neck”, “head and neck cancer”, “head and neck carcinoma”, “HPV-related tumours”, “immune system”, “immune cells”, “subpopulations of T cells”, “tumour immune escape”, “immunotherapy”, “immune checkpoints”, “combination therapy, immunotherapy”, “immunotherapy, biomarkers”, “adoptive cell therapy“. The search terms also included “pembrolizumab”, “nivolumab”, “durvalumab”, “anty-PD-1”, “anty-PD-L1”, anty-CTLA-4”, “PD-1/PD-L1”, “immune checkpoint inhibitor”, “combination therapy, immunotherapy”, “adverse event”, “toxicity”. There was no restriction on language or research group characteristics. No exclusion criteria were employed.

This article extensively discusses immune checkpoints, biological mechanisms of tumour immune escape and its implications on the effectiveness of immunotherapy in HNSCC or its limitations. It also focuses on the crucial results of molecular studies followed by latest pre- and clinical early trials (I/II phase) and phase-III clinical trials, i.e., studies on in vitro models of head and neck carcinoma; it also pays particular attention to the most clinically important long-term observational and intervention population studies and key opinion-forming systematic reviews. This work discusses the issues in detail, in order of increasing clinical credibility.

## 3. Results

### 3.1. Immunoediting and Immune Surveillance, Biological Mechanisms of Tumour Immune Escape

#### 3.1.1. Immunoediting and Immune Surveillance

Despite the huge progress made in understanding the mechanisms of cancer pathogenesis and the role of immunocompetent cells, and despite the promising results of pre- and clinical trials of molecular therapy and significant successes of immunotherapy, still the factors and processes occurring in the tumour environment which enable the anti-cancer immune defence in the host remain elusive. At each stage of carcinogenesis, the immune system generates mechanisms aimed at inhibiting the promotion, development and progression of the malignant tumour. This phenomenon is known as cancer immunoediting. Unfortunately, many human cancers of various origin have developed biologically proven escape mechanisms from immune surveillance, which may affect the unlimited growth of tumour cells and contribute to the ineffectiveness of modern forms of immunotherapy, i.e., therapy based on immune checkpoint blockade, therapeutic vaccines, adoptive T cell therapy, combination therapies and treatment targeting immunocompetent suppressive cells [324,325,326,327,328].

The development of new therapeutic strategies in cancer immunotherapy is directly related to the growing knowledge of the cancer immunity cycle, which constitutes the steps for the effective elimination and inhibition of tumour cells by immunocompetent cells, such as tumour infiltrating lymphocytes (TILs) and circulating different subpopulations of immunocompetent cells, such as neutrophils, cancer-associated macrophages (TAMs/MФ), cancer-associated fibroblasts (CAFs), regulatory T cells (CD4^+^CD25^+^Foxp3^+^ T_regs_), cytotoxic CD3^+^CD8^+^ T cells (CTLs) and CD3^+^CD4^+^ T helper type ½/9/17 (Th_1_/Th_2_/Th9/Th_17_) lymphocytes, T follicular helper cells (Tfh), CD56^dim^/CD16^bright^ activated natural killer cells (NK) as well as carcinoma-associated fibroblasts (CAFs), myeloid-derived suppressor cells (MDSCs) and tumour-associated macrophages (M1/M2 TAMs). The cancer immunity cycle comprises the following stages: (A) immunogenic cell death and tumour antigen release from the neoplastic cells, (B) antigen processing and presentation by antigen presenting cells (APCs), i.e., mature dendritic cells (DCs) to naïve immune cells in tumour-draining lymph nodes and activation of CD4^+^ Th cells, (C) immunocompetent T cell priming and activation of CD4^+^ Th-cells and subsequently effector CD8^+^ T cells (CTLc) in the local lymph nodes, (D) displacement of T cells to the TME via the blood stream and infiltration to the tumour, (E) recognition of tumour cells by immune cells, CD8^+^ T cells (CTLc), (F) T cell mediated immune response, tumour cell recognition and initiation of cytotoxicity (killing of cancer cells) [50,324,325,326].

The schema of cancer immunity cycle and the steps in cancer immunoediting is shown in Figure 1A–F.

#### 3.1.2. Biological Mechanisms of Tumour Immune Escape

##### Deficiency in the Tumour Antigen Release from the Tumour Cells (TM)

The condition for effective activation of immune cells in the tumour microenvironment and induction of an anti-neoplastic immune response is the release of tumour antigens, occurring as a result of the preceding cell apoptosis. The released cancer neoantigens are presented to effector T cells via the major histocompatibility complex (HLA), i.e., MHC class I molecules (MHC-I restriction). The release of these antigens into the microenvironment determines an effective and rapid T cell immune response. Therefore, abnormalities in the expression of tumour antigens impair their presentation by the cells, which may lead to evasion of immune surveillance. Importantly, a lower tumour mutation burden (TMB) load is associated with a lower production of mutation-associated neoantigens, and hence less recognition of them as foreign antigens. Consequently, the effectiveness of CD4^+^ T helper (Th_1_ lymphocytes) activation is decreased, as is the cytotoxic tumour cell killing reaction by CD8^+^ cytotoxic cells (CTLs). This can lead to immune evasion by tumour cells and reduced efficacy of immunotherapy due to an unsatisfactory response to therapeutic agents. This of course results in worse clinical benefits, i.e., further worse outcomes and shorter survival time [329,330,331]. The results of clinical trials indicate that the efficacy of treatments using nivolumab alone or in combination with ipilimumab correlate with tumour mutation burden. Lower objective response rates and overall survival rates were noted in patients with low TMB, suggesting that increasing the mutation load may contribute to further improving the efficacy of immunotherapy [330,332,333]. Another factor contributing to cancer initiation, development and local/regional spread is the loss of neoplastic antigens, i.e., tumour-associated antigens (TAAs) or tumour specific antigens (TSAs), and immunoselection of tumour cell clones with poor immunogenicity. This phenomenon can lead to acquired resistance to immunotherapy by neoplastic cells, by reducing or inhibiting the interaction between immune-stimulating antigens and immunocompetent cells, and even result in immune “blindness” to cancer antigens [334,335,336,337] (Figure 2A).

##### Disturbances in the Maturation and Function of Dendritic Cells (DCs)

Dendritic cell (DC) dysfunction may be another important mechanism facilitating tumour escape from immune surveillance. Disturbances in the DC maturation process and the abolition of their function as APC cells, which prevents the proper presentation of tumour antigens to naïve immune cells in tumour-draining lymph nodes, and the subsequent priming of immunocompetent cells and activation of CD4^+^ T cells, may occur. This has been attributed to the inhibition of the expression of co-stimulating molecules CD80/CD86 and other cell membrane molecules such as HLA-DR, CD83, CD40 on DCs by interleukins, i.e., IL-6 and IL-35; this process may act by blocking STAT1/STAT3 and inhibiting the mitogen-activated protein kinase p38 MAPK/NF-ĸβ(p65) pathway, and by downregulating MHC-II class αβ dimer level, CD86(B7-2) and IL-12 expression by activation of the IL-6/STAT3 signalling pathway [329,338,339,340,341]. These changes inhibit the activation of naïve T lymphocytes and their transformation into the Th_1_ subpopulation, as well as decreased synthesis of IFN-γ and IL-2, and a subsequent decrease in the effectiveness of CTL function [339,342,343] (Figure 2B).

##### Defects in Immunocompetent Cell Priming and Activation of T cells in Local Lymph Nodes

T cell priming is a prerequisite for the activation of CD4^+^ Th_1_ lymphocytes, which are stimulated by the interaction between the T cell receptor (TCR) and MHC class II molecules, and between CD28 on T cell and CD80(B7-1)/CD86(B7-2) co-stimulating proteins present on the surface of APCs (DCs) and CD4^+^ lymphocytes. Following CD4^+^ T cell stimulation, CD8^+^ cytotoxic lymphocytes (CTLs) activate the T cell mediated immune response, resulting in the killing of tumour cells, as a result of the interaction between TCR molecules and the MHC class I complex αβ dimer and β2 microglobulin (B2M). Immune check-point proteins, e.g., cytotoxic T cell antigen 4 (CTLA-4/CD152) also participate in immune priming. Unfortunately, activation of CTLA-4, a factor that inhibits the proliferation and stimulation of T cell effector activity, is associated with an increase in the affinity of the CTLA-4 molecule to co-stimulate CD80(B7-1)/CD86(B7-2) on DCs compared to CD28. Increased expression and binding of CTLA-4 to CD28 contributes to the activation of intracellular mechanisms responsible for the immunosuppression of the cytotoxic tumour cell killing reaction by CD8^+^ cytotoxic cells (CTLs) and the immunotolerance of tumour neoantigens [329,344,345,346]. Tumour immune escape at this stage of the cancer immunity cycle occurs most often as a result of the inhibition of several key mechanisms and proteins, i.e., CTLA-4-mediated intracellular regulatory pathways, such as p38 MAPK/NF-ĸβ(p65), PI3K/AKT and Mdm2/p53, cyclin-dependent kinase inhibitors p27^KIP1^, p21^CIP1/WAF1^, CDK (cyclin-dependent kinase) inhibitors CDK4/6 and G1/S-specific cyclin-D3 [347,348,349,350]. The immunosuppressive activity of CTLA-4 is also associated with the induction of indoleamine 2,3-dioxygenase enzyme 1 (IDO1) activity and the suppression of the ζ(zeta)-chain-associated protein kinase 70 (ZAP70), which inhibit the activation of T cells and play a critical role in T cell signalling [329] (Figure 2C).

##### Disturbed Trafficking of T Cells to the Tumour Microenvironment (TME) and Defect in the Recognition of Tumour Cells by Immunocompetent Cells

The vascular endothelial growth factor (VEGF) family, particularly VEGF-A, serve as signal proteins that stimulate the formation of blood vessels from pre-existing vasculature or the de novo development of tumour microcirculation after VEGFR receptor activation. VEGF/VEFR overexpression is often found in human solid tumours, which, apart from activating angiogenesis, reduce vascular permeability and inhibit the expression of adhesion molecules and coactivators, i.e., VLA-4/VCAM-1 and LFA-1/ICAM-1, thus promoting a decrease in adhesion of CTL effector lymphocytes to the endothelium and preventing the colonization of tumours by immunocompetent cells (TILs). Moreover, the presence of the suppressive cytokine IL-10 and prostaglandin E2 (PGE_2_) contributes to the upregulation of the FasL molecule on endothelial cells, leading to apoptosis of CD8^+^ cytotoxic T cells and inhibition of tumour microenvironment (TME) infiltration by T effector cells in Fas/Fas-L (CD95) [351,352,353,354,355]. A similar functional effect in tumour development occurs as a result of overexpression of the endothelin B receptor (ET_B_), which stimulates upregulation of the ICAM-1 molecule on endothelial cells, leading to reduced efficiency of the trafficking of T cells to the tumour and enhanced apoptosis, as well as more intense proliferation, migration and invasiveness of tumour cell lines [356,357,358,359].

Disturbances in the effective presentation of tumour neoantigens to CTL effector cells may result from defects in the expression of the MHC-I component, i.e., β2 microglobulin (B2M), or other factors involved in the presentation of cancer antigens, e.g., TAP-associated glycoprotein (TAPBP, tapasin) and a member of the ATP-binding cassette transporter family: a transporter associated with antigen processing 1 (TAP1). Downregulation of B2M in the MHC-I/TCR complex allows for the tumour cells to avoid immune destruction mechanisms [360,361]. Genetic changes, such as homozygous or biallelic gene loss for B2M, lead not only to decreased HLA class I expression on tumour cells, but also increased resistance to immune checkpoint receptors targeting immunotherapy, e.g., anti-PD-1 and anti-CTLA-4 treatment [362,363]. Furthermore, a knock-down of TAP-1 gene or epigenetic changes and TAP-1 protein downregulation, as well as the loss of tapasin, an essential component of the peptide-loading complex (PLC) in human cancer cells, elicits tumour immune escape by inhibiting CTL surveillance against the tumour. This results in a poor prognosis for the patient [364,365,366,367,368,369]. Studies on the role of cytotoxic CD8^+^ T cells (CTLs) in the immune surveillance escape mechanisms also suggest an accumulation of positively stimulated CD4^+^CD25^+^Foxp3^+^T_regs_ cell activity in the CD8^+^ T cell-dependent mechanism associated with the production of the CCL2 chemokine and CCR4 that directly recruit immuno-inhibitory regulatory T cells [59] (Figure 2D).

##### Lack of T Cell Mediated Immune Response and Deficiency in the Killing of Tumour Cells

Initiation of the cancer cell killing process by immunocompetent cells present in the tumour microenvironment can be significantly impaired by several mechanisms, i.e., increased activity of regulatory immunosuppressive enzymes, inhibition of checkpoint receptors and tumour-activated co-inhibitors for infiltrating lymphocytes (TILs), i.e., T cell exhaustion, tumour induced T cell anergy and tumour-associated immunocompetent cells in the tumour microenvironment (TME) [329].

Indoleamine 2,3-dioxygenase 1 enzyme (IDO1) activity is involved in cell tryptophan metabolism and catalyses the first and rate-limiting step in the kynurenine pathway, the O_2_-dependent oxidation of L-tryptophan to N-formylkynurenine, the others being indolamine-2,3-dioxygenase 2 (IDO2) and tryptophan 2,3-dioxygenase (TDO) [370,371]. Human IDO (hIDO) is a key immune checkpoint molecule acting as an immunomodulatory enzyme produced by alternatively activated tumour associated macrophages (TAMs/MФ) and other immunoregulatory cells and it has strong immunosuppressive action because of its ability to limit T cell function and engage mechanisms of immune tolerance. IDO1 is also known to suppress T and NK cells, generate regulatory T cell (CD4^+^CD25^+^Foxp3^+^T_regs_) and myeloid-derived suppressor cells (MDSCs), and also support angiogenesis process. These mechanisms are crucial in the process of carcinogenesis. IDO1 allows tumour cells to escape the immune system by two main mechanisms. The first one is based on L-tryptophan (L-Trp) depletion from the tumour microenvironment. A decrease in L-Trp conversion causes an increase in the production of kynurenine and its downstream catabolites; this promotes cell cycle arrest, leading to the loss of immunocompetent cell function and increased activation of immune suppressor cells, i.e., Treg and MDSCs. The second mechanism is based on increasing production of kynurenines, which are cytotoxic for T lymphocytes and NK cells [372,373]. Increased expression of kynurenine on CD8^+^ cytotoxic lymphocytes also stimulates the expression of the programmed cell death protein 1 receptor (PD-1) on CTLs through a kynurenine-acryl hydrocarbon receptor pathway, leading to the apoptosis of effector T cells [374]. Recent studies also indicate that IDO1 plays a significant role in enhancing tumour growth and inhibiting tumour cell apoptosis via the PI3K/AKT signalling pathway [375]. Overexpression of hIDO1 is described in a variety of human solid cancer cell lineages and is often associated with poor prognosis [376]. Emerging clinical studies suggest that the use of IDO inhibitors, e.g., d-1-methyl-tryptophan and INCB24360 with classical chemo- or radiotherapy, and PCC0208009, INCB024360 and NLG919 could restore a correct immune response and provide a therapeutic response to generally resistant human neoplasms. IDO1 forms the basis of various research and clinical uses, particularly in combination with immune checkpoint inhibitors [377,378,379,380].

Furthermore, tumour-induced T cell anergy may be also one of the crucial immune evasion mechanisms in patients with cancer. Indirect evidence indicates that T cell anergy may lead to T cell-intrinsic dysfunction contributing to tumour immune escape. Anergic T cells are induced hyporesponsive T cells or naïve T cells stimulated with low co-stimulatory and/or high co-inhibitory signalling. These T cells are unresponsive to subsequent activating conditions with limited IL-2 expression that induce tolerance in the periphery and cell cycle arrest at the G1/S phase, and promote the development of neoplastic diseases. It has been hypothesized that the anergy phenomenon is initiated by mTOR/Ras/MAPK pathway in T cells and related to the downstream of TCR/CD28 engagement. It is suggested that early growth response gene 2 (Egr2) may be a key transcription factor that regulates T cell anergic state in vitro and in vivo by focusing on key target gene diacylglycerol kinase α (DGK-α) expression, a focal protein of T cell anergy. Moreover, an active RAP-1 (Ras-proximate-1 or Ras-related protein 1) molecule has been found to be constitutively present in anergic T cells, where it reduces the expression of IL-2 and inhibits normal TCR signalling. This leads to genetic reprogramming of T cells mediated by NFAT (nuclear factor of activated T cells) homodimer formation and transcription of anergy-inducing genes [59,60].

In addition, escape from immune surveillance can be facilitated by molecules that inhibit immune checkpoints (ICIs), i.e., PD-1, LAG-3, TIM-3, TIGIT, VISTA, glucocorticoid-induced TNFR-related protein (GITR) and killer immunoglobulin-like receptor (KIR), as well as novel co-inhibitory checkpoints, e.g., OX-40, CD40, ICOS, 4-1BB, KIR2DL-1/2/3, NKG2A, CD96 and killer cell lectin-like receptor subfamily G, member 1 (KLRG-1) [329,381].

Programmed death ligand (PD-1) is a well-known molecule that inhibits the function of activated T lymphocytes and represents the point of interaction with the corresponding PD-L1 (B7-H1) costimulatory molecule expressed on the tumour cells. The interaction of PD-L1 on the tumour cells with PD-1 on a T cell reduces T cell function signals to prevent the immune system from attacking the tumour cells. The PD-1/PD-L1(B7-H1) interaction leads to the inhibition of local anti-tumour immunity through the attachment of a phosphate residue to PD-1 and the recruitment of protein tyrosine kinases, e.g., protein tyrosine phosphatases (PTPs), including SHP2, these being the enzymes that remove groups from phosphorylated tyrosine residues on proteins of key cell pathways, such as PI3K/AKT/mTOR and Ras-Raf/MEK/ERK. Because of this, PTPs have been implicated in the regulation of many cellular processes, including the inhibition of T cell activation, proliferation and viability, reduced target cell lysis, altered lymphocyte motility, and metabolic reprogramming; they have also been associated with inducing differentiation to a regulatory T cell (CD4^+^CD25^+^Foxp3^+^T_regs_) phenotype and the production of immunosuppressive cytokines (IL-10, TGF-β) [324]. Moreover, stimulation of the PD-L1 receptor on cancer cells inhibits the programmed cell death (apoptosis) signal, thus inhibiting mediating cytotoxic CD8^+^ T lymphocyte (CTLs) anti-tumour cytotoxicity and the activation of death-inducing complex (DISC), as well as subsequent Fas-mediated target tumour killing. Importantly, up to 60% of HNSCC demonstrate increased expression of PD-1 within the TME, indicating that this is a key mechanism for immune escape in HSNCC [51].

Interestingly, functional in vitro and in vivo experiments have shown the existence of higher stemness of PD-L1*^hi^* in TME of cancers of different origins compared with PD-L1*^low^* cells [75]. Moreover, stimulation of PD-L1 allows neoplastic cells to avoid the anti-tumour toxicity of IFN-γ released by cytotoxic T cells and type-1 T helper cells (Th_1_); it also activates angiogenesis by upregulating angiogenic stimuli deriving from tumour cells [382]. Importantly, immunotherapy with immune checkpoint inhibitors appears to be the most promising drug category for many human cancers. The use of inhibitors that block the interaction of PD-L1 with the PD-1 receptor may thus prevent the cancer from evading the immune system. As such, several PD-1 and PD-L1 inhibitors are currently being tested in the clinic for use in a wide variety of cancers, including HNSCC [383]. Furthermore, several anti-PD-1 and PD-L1(B7-H1) antibodies have been approved for the treatment of an array of cancer types by The United States Food and Drug Administration (FDA), e.g., nivolumab (2014), pembrolizumab (2014), atezolizumab (2016), avelumab (2017), durvalumab (2017), cemiplimab (2018), dostarlimab (2021). Current clinical trials are also evaluating anti-PD-1 and PD-L1 drugs in combination with other immunotherapy drugs, i.e., antibodies targeting lymphocyte-activation gene 3 (LAG-3), type I transmembrane protein B7-H3, killer-cell immunoglobulin-like receptors (KIRs), poly (ADP-ribose) polymerase (PARP), a member of the TNF-receptor superfamily (CD27) and inducible T cell co-stimulator (ICOS) [384].

Lymphocyte-activation gene 3 (LAG-3) is a cell surface molecule with diverse biologic effects on T cell function and it is expressed on activated CD4^+^ T_h_ cells, cytotoxic CD8^+^ T_c_ cells (CTLs), regulatory T cells (CD4^+^CD25^+^Foxp3^+^T_regs_), CD56^dim^ natural killer cells (NK) and B cells and plasmacytoid dendritic cells. LAG-3 constitutes the main co-inhibitory ligand for MHC class II, to which it binds with higher affinity than CD4 molecule. Other known ligands are also galectin-3, fibrinogen-like protein 1 (FGF-1) and L-selectin presence on tumour cells. The LAG-3 receptor negatively regulates cellular proliferation, function and homeostasis of T cells, causing a reduction of INF-γ synthesis and acting in a similar fashion to CTLA-4 and PD-1, and also by reducing production of cytokines and other inhibitory receptors. Moreover, LAG3 also helps maintain CD8^+^ T cells in a tolerogenic state and has been reported to play an issue role in CD4^+^CD25^+^Foxp3^+^T_regs_ suppressive function, thus favouring the escape of the tumour from immune surveillance in various human cancers [385,386,387,388,389]. Interestingly, preclinical studies in vitro showed that the use of an anti-LAG-3 antibody (REGN3767), alone or in combination with cemiplimab (REGN2810, a human anti-PD-1 antibody), resulted in increased efficacy and higher pro-inflammatory cytokine secretion by tumour-specific TILs, as well as improved survival and eradication of neoplastic cells in a mouse tumour model [390,391,392].

T cell immunoglobulin and mucin-domain containing-3 (TIM-3) or HAVCR2 receptor (HAVCR2/TIM-3) is a relevant cell transmembrane protein expressed on IFN-γ producing CD4^+^ Th_1,_ CD8^+^ cytotoxic Tc_1_ cells, Th_17_ cells and mast cells, as well as regulatory T cells (CD4^+^CD25^+^Foxp3^+^T_reg_) and myeloid-derived suppressor cells (MDSCs). Expression of TIM-3 has been also detected in innate immune cells, i.e., dendritic cells (DCs), NK cells and monocytes [393]. The HAVCR2/TIM-3 molecule has been described as a checkpoint inhibitor and key regulator of the immune response acting by inhibiting T cell responses in both autoimmunity and cancer initiation and progression, as well as suppressing macrophage activation following PD-1 inhibition. This protein interacts with multiple components of the T cell receptor (TCR) complex; it mediates intercellular signalling pathways and negatively regulates TCR function, leading to the depletion of CD8^+^ T cells for the proliferation and secretion of cytokines, such as TNF-α, IFN-γ and IL-2 and NK cell exhaustion [394,395,396]. HAVCR2/TIM-3 also has a free form (soluble form) lacking mucin and the transmembrane domain; however, its function is unknown [393]. HAVCR2/TIM-3 expression is upregulated in tumour-infiltrating lymphocytes in head and neck cancer and other neoplasms of different origin [397]. Upregulation of TIM-3 was also observed in tumours progressing after anti-PD-1 therapy, and overexpression of HAVCR2/TIM-3 was an indicator of poor prognosis. This seems to be a form of adaptive resistance to immunotherapy [398,399,400,401]. Numerous multicentre phase I and II clinical trials are underway using anti- HAVCR2 monoclonal antibodies in combination with anti-PD-1 or anti-PD-L1 therapies (LY3321367, Eli Lilly and Company; MBG453, Novartis Pharmaceuticals; TSR-022, Tesaro, Inc.). Research shows that blocking this receptor may improve the anti-tumour immune activity in many human cancers [402,403,404].

An immune receptor/immune checkpoint present on T cells, regulatory T cells (CD4^+^CD25^+^Foxp3^+^T_regs_) and CD56^dim^ NK cells is a T cell immunoreceptor with Ig and ITIM domains (TIGIT protein), also known as WUCAM/Vstm3. It inhibits effector T cell activation in vivo and NK cytotoxicity by binding to CD115 and CD112 molecules on dendritic cells (DCs) and macrophages (TAMs/MФ), but also to ligands on the tumour cell surface, thus facilitating the mechanisms of immune escape in the cancer immune cycle [405,406,407,408]. Importantly, TIGIT and programmed death-ligand 1 (PD-1) have been shown to be upregulated on tumour antigen-specific CD8^+^ T cells and CD8^+^ tumour infiltrating lymphocytes (TILs) in both HNSCC patients and mouse models; in addition, their level was correlated with other immune-checkpoint molecules, i.e., PD-1, TIM-3 and LAG-3 [409,410,411]. Therefore, a few clinical trials on TIGIT-blockade have recently been initiated, predominantly in combination with immune checkpoint co-blockade therapy against PD-L1 used in different human solid cancers [410,412,413,414]. Blockade of these relevant immune checkpoint axes have been observed to enhance antitumour immune responses by activating T cell proliferation and function, cytokine production, degranulation of CTLs CD8^+^ T cells and TIL CD8^+^ T cells, as well as reducing the CD4^+^CD25^+^Foxp3^+^T_reg_ population and TGF-β1 secretion [407,409,415,416]. Interestingly, dual blockade of TIGIT/CD155 signalling was found to reverse T cell exhaustion and enhance antitumour capability in HNSCC, which shed light in the therapeutic effect of the TIGIT/CD155 pathway in a transgenic mouse model [409]. Furthermore, the CITYSCAPE clinical trial (NCT03563716) evaluated the combination of the anti-TIGIT antibody tiragolumab in combination with the anti-PD-L1 antibody atezolizumab in patients with newly diagnosed non-small cell lung cancer whose tumours expressed PD-L1. The combination has significantly reduced the risk of cancer progression or death compared to anti-PD-1 monotherapy [413]. However, the randomized phase III study failed to confirm these results [417].

V-domain Ig suppressor of T cell activation (VISTA protein) is a type I transmembrane protein that functions also as a negative immune checkpoint regulator. VISTA is expressed on tumour-infiltrating lymphocytes (TILs), myeloid-derived suppressor cells and regulatory T cells (MDSCs), regulatory T cells (CD4^+^CD25^+^Foxp3^+^T_reg_), dendritic cells (DCs), tumour associated macrophages (TAMs/MФ) and neoplastic squamous carcinoma cells [418,419]. There is evidence that VISTA can act as both a ligand and a receptor on effector T cells to inhibit T cell proliferation and function, reduce cytokine generation, i.e., IL-2, IL-17 and IFN-γ, and maintain peripheral tolerance [420]. Preliminary phase I clinical trials of anti-cancer immunotherapy using a monoclonal antibody directed against a VISTA molecule in advanced human cancers are ongoing. Since VISTA protein may also be expressed on the tumour cells in various human neoplasms, where it correlates with an acquired resistance to anti-PD-1 immunotherapy and poor patient survival, co-blockade therapy against PD-L1 and VISTA has been proposed. The latest phase I research on the use of the programmed death-ligands 1 and 2 (PD-L1 and PD-L2)/VISTA pathway co-antagonists (drug: CA-170) show a relevant anti-neoplastic effect in patients with advanced solid tumours or lymphomas who have progressed or are non-responsive to available therapies and for whom no standard therapy exists (NCT02812875) [421] (Figure 2E).

### 3.2. Current Perspectives and Future Prospects of Immunotherapeutic Strategies for Overcoming Immune Escape of HNSCC

The standard therapies used to treat patients with squamous cell head and neck cancer mainly include surgical removal of the tumour, chemotherapy and radiotherapy. Unfortunately, a large proportion of HNSCC subjects are refractory to these therapies, and this population is characterized by unacceptably low survival rates and high recurrence rates [422,423]. Therefore, new therapies are needed to treat HNSCC to achieve better efficiency, less toxicity and better quality of life [424,425]. The HNSCC tumour landscape is permissive of these tumours’ aggressive nature, fostered by the actions of the immune system. Indeed, the immune system has a vital role in the tumorigenesis, development, and progression of these heterogeneous, aggressive and genetically complex collection of malignancies. Different immune escape mechanisms facilitate HNSCCs and allow their progression. An understanding of this phenomenon provides the basis for improved, novel anti-tumour therapies and better outcomes for patients.

The tumour microenvironment (TME) of HNSCC consists of many different subpopulations of immune cells that infiltrate neoplastic tissue and interact with tumour cells or with each other via various pathways. Immune cells involved in both innate and adaptive immunity play a key role in mediating immune surveillance and controlling tumour initiation and growth. In the TME tumour microenvironment, immunosuppression occurs as a result of complex interactions between immunologically heterogeneous cell populations, such as TAMs, CAFs, MDSCs, NKs and T_regs_. Therefore, attempts are being made to effectively target these cells by inhibiting their activation or function in order to reverse clinically unfavourable local and systemic immunosuppression, to achieve potent anti-tumour immunity and reverse treatment resistance. HNSCC is characterized by strong immunosuppression and therefore is expected to respond well to current and modern methods of immunotherapies, e.g., modulation of immune checkpoints, including blockade of inhibitory checkpoints (ICIs) and agonists of stimulatory checkpoints by specific antibodies and adoptive cell transfer, including TCR/CAR-T transfer and NK cell adoptive transfer. Promise is also offered by DC-based vaccination including targeting DCs in vivo with cytokines, such as GM-CSF, and agonistic antibodies, such as CD40, as well as antigens that bind to C-type lectin receptors (CLRs), such as DEC-205. Ex vivo approaches can also be used, by inducing the immature DCs originating from the patients’ monocytes into mature DCs by loading peptide, protein or tumour lysate as antigens; in addition, some approaches use nanotechnology-based immunotherapy [367,368,369,426,427,428,429,430]. Consequently, a key goal of immunotherapy strategies is to overcome the barriers of immune suppression or tolerance and inhibit the mechanisms that facilitate tumour escape from immune surveillance in HNSCC. Numerous clinical trials are underway to further refine the application of immunotherapy and develop new immunotherapy approaches in various human solid tumours, including squamous cell head and neck carcinoma (each research available on the ClinicalTrials.gov website: www.clinicaltrials.gov;). ClinicalTrials.gov is a database of privately and publicly funded clinical studies conducted around the world.

The subsequent steps in cancer immunoediting in relation to selected drugs and compounds currently available for immunotherapy, as well as immunotherapeutic agents incorporated within novel combination therapy strategies in patients with HNSCC, are presented in Figure 2A–F.

#### 3.2.1. HNSCC Immune Checkpoints Receptors and Their Targeting by Immunotherapy (Immune Checkpoint Inhibitors, ICIs)

Immune checkpoint inhibitors (ICIs) of the adaptive immune system are currently critical targets for immunotherapy due to their therapeutic potential in cancers of various origins, including HNSCC. The properties of checkpoints determine the strategy of anti-cancer management by suppressing inhibitory checkpoints or activating stimulatory checkpoints with specific antibodies. Therefore, few interventional clinical trials have investigated the treatment with checkpoint inhibitors for HNSCC used in monotherapy or combined immunotherapy, i.e., CheckMate 141 (ClinicalTrials.gov nr NCT02105636), KEYNOTE-048 (NCT02358031), KEYNOTE-012 (NCT01848834) and KEYNOTE-055 (NCT02255097) and KEYNOTE-158 (NCT02628067), or novel combination therapy strategies, i.e., Active8 (ClinicalTrials.gov, number NCT01836029), KEYNOTE-048 (NCT02358031), CheckMate-651 (NCT02741570), CONDOR (NCT02207530), KESTREL (NCT02551159), EAGLE (NCT02369874), MASTERKEY-232 (NCT02626000), KEYNOTE-184 (NCT02521870), etc. [42,43]. Key immunotherapy trials incorporating ICIs in an HNSCC setting, based on The Society for Immunotherapy of Cancer consensus statement on immunotherapy for the treatment of HNSCC, are summarized in Appendix A.

##### PD-1/PD-L1 Inhibitors

Binding of the PD-1 protein, which is expressed mainly in T lymphocytes, to the PD-L1 molecule present on cancer cells, leads to the inhibition of the T cell-dependent immune response and is one of the important mechanisms of the tumour escaping the surveillance of the immune system. In HNSCC, PD-L1 expression is observed in 50–100%, which is relatively high [431]. Anti-PD-1/PD-L1 ICIs block suppressive signalling through the PD-1/PD-L1 pathway, thus enhancing immune cell activity [432]. The increase in T cell anti-tumour action and efficacy is intended to enhance the response to tumour neoantigens and to convert HNSCC, generally known as “cold tumours” with low local immune activity, to “hot tumours” with higher local immune response [383,433]. The approved immune checkpoint inhibitors so far are the PD-1/PD-L1 blocking antibodies (e.g., pembrolizumab, an IgG4 monoclonal antibody (mAb), Keytruda, Merck and nivolumab, Opdivo, Bristol-Meyers Squibb—anti-PD-1, or avelumab and atezolizumab, duralumab—anti-PD-L1, all the IgG1 mAbs) and the blockade of CTLA-4 signalling using the CTLA-4 antibody (e.g., ipilimumab, an IgG1k mAb, tremelimumab, an IgG2 mAb) [434]. Since 2016, for patients with head and neck cancer, the U.S. Food and Drug Administration (FDA) has approved both nivolumab and pembrolizumab for resistant R/M HNSCC tumours that have not responded to platinum-based therapy. Subsequently, in 2017, the European Commission approved nivolumab for the treatment of the same patient population with HNSCC, followed shortly thereafter by pembrolizumab as monotherapy in recurrent or metastatic PD-L1-expressing HNSCCs with a tumour proportion score of ≥50% and in cases where progression occurred during or after platinum-containing chemotherapy [292,383,435].

An example of a clinical trial using ICIs in monotherapy in patients with recurrent or metastatic squamous-cell carcinoma of the head and neck (R/M HNSCC) after platinum chemotherapy is CheckMate 141, ClinicalTrials.gov number, NCT02105636). The median overall survival was found to be 7.5 months (95% CI: 5.5–9.1) in the nivolumab group vs. 5.1 months (95% CI: 4.0–6.0) in the group that received standard therapy. Overall survival was significantly longer with nivolumab than with standard therapy (HR 0.70; 97.73% CI: 0.51–0.96; *p* = 0.01). The response rate was also higher, i.e., 13.3% vs. 5.8% in the nivolumab group and the standard-therapy group, respectively. These data support the approval of nivolumab as a type of monotherapy for HNSCC patients with a higher progression stage of neoplastic disease [436]. Recently, researchers also assessed the response to immunotherapy based on the Response Evaluation Criteria in Solid Tumors (RECIST), Version 1.1. The randomized, open-label, phase 3 CheckMate 141 trial (NCT02105636) evaluated nivolumab in R/M HNSCC after platinum therapy in patients treated beyond first RECIST-defined progression (TBP) and confirmed that tumour burden reduction was noted in a proportion of patients who received TBP with nivolumab (25% had stable disease, 25% showed a reduction in target lesion size and 5% had reductions >30%) [437].

pembrolizumab is also indicated as a mAb with significant anti-tumour activity in HNSCC, resulting in improved overall response rates (ORRs) with moderate toxicity. In the randomised, phase III study KEYNOTE-048 (NCT02358031), the effectiveness of pembrolizumab treatment (pembrolizumab alone vs. pembrolizumab plus a chemotherapy, i.e., platinum and 5-fluorouracil vs. cetuximab with chemotherapy) was related to the PD-L1 cumulative positive score (CPS). This score combines tumoral together with the PD-L1 expression of immune cells. The study confirmed that pembrolizumab alone improved OS compared with cetuximab plus chemotherapy in the CPS (HR 0.61; 95% CI: 0.45–0.83, *p* = 0.0007). Furthermore, pembrolizumab with chemotherapy also improved OS in comparison with cetuximab plus chemotherapy in the total population (HR 0.77; 95% CI: 0.63–0.93, *p* = 0.003). Based on the results of the KEYNOTE-048 study, pembrolizumab alone or in combination with chemotherapy is now approved as first-line therapy for all R/M HNSCC patients with a cumulative positive score (CPS) ≥1 [438].

Results from the phase Ib KEYNOTE-012 (NCT01848834) and the phase II study KEYNOTE-055 (NCT02255097) expansion cohort for Keytruda pembrolizumab show continued benefit in ORRs and duration of response lasting up to 30 months in patients with previously treated R/M HNSCC. In KEYNOTE-012, for the primary endpoint, findings disclosed an ORR of 18% (95% CI: 13–24). The secondary endpoint results displayed a median overall survival (OS) rate of eight months (95% CI: 6–10) and a 6-month progression free survival (PFS) of 23%. This study also revealed an ORR of 24% (95% CI: 13–40) in patients with HPV*^+ve^* disease. In contrast, patients who had HPV^−^ tumours showed an ORR of only 16% (95% CI: 10–23) [439,440]. For the second study, KEYNOTE-055, which enrolled patients regardless of PD-L1 tumour status, an analysis showed an ORR in 18% (95% CI: 9–31). KEYNOTE-055 support the safety, tolerability and anti-tumour activity of pembrolizumab as a monotherapy in patients with HNSCC with disease progression on platinum-based and cetuximab (EGFR mAb) therapy [441].

The confirmatory phase III study KEYNOTE-040 (NCT02252042) has compared the anti-cancer effect of pembrolizumab to standard therapies with methotrexate, docetaxel or cetuximab in patients with platinum-resistant R/M HNSCCs. In a multicentre clinical trial, it was confirmed that the use of pembrolizumab extended the median OS to 8.4 months (95% CI: 6.4–9.4) and to 6.9 months (95% CI: 5.9–8.0) for standard therapy (HR 0.80; 0.65–0.98, *p* = 0.01). Furthermore, the median duration of response was 7.0 (95% CI: 2.1–11.1) months. The response rate was higher in HPV*^+ve^* patients. Moreover, treatment with anti-PD-1 antibodies reduced the rate of grade ≥3 treatment-related adverse events [442,443].

The relationship of tumour mutational burden (TMB) with outcomes in patients with advanced solid tumours, also with R/M HNSCCs, treated with pembrolizumab in monotherapy, was analysed in the multicohort phase II KEYNOTE-158 study (NCT02628067). Research disclosed that objective responses were observed in 29% (95% CI: 21–39) among patients in the TMB-high group and only 6% (95% CI: 5–8) of 688 in the non-TMB-high group. Moreover, in HNSCC individuals, high TMB level was also correlated to higher ORR and longer OS rate after immunotherapy, particularly in HPV*^−ve^* tumours, whereas this association was not significant in HPV*^+ve^* neoplasms [444].

A few recent studies support that the cancer immunotherapy targeting PD-L1 as a helper T cell antigen would be a rational strategy for HNSCC patients [445,446]. Importantly, compared to traditional therapies, the analysed immunotherapies with antibodies blocking the PD-1/PD-L1 pathway show more satisfactory therapeutic effects and lower toxicity, and improves treatment outcomes in patients with advanced squamous cell carcinoma of the head and neck [323,447]. For example, atezolizumab, durvalumab and avelumab are known IgG1 mAbs targeting PD-L1 molecule in tumour cells, also in advanced HNSCCs [383,446,448].

In the HNSCC cohort of the phase Ia PCD4989g clinical trial (NCT01375842), the patients were monitored for safety and tolerability and were evaluated for a response for atezolizumab administered intravenously. In this pre-treated advanced HNSCC population, PD-L1/PD-1 signalling pathway inhibition by atezolizumab had a tolerable safety profile and encouraging activity, with responses observed regardless of HPV status and PD-L1 expression level. Namely, objective responses by the Response Evaluation Criteria in Solid Tumors Version 1.1 (RECIST v1.1) occurred in 22% of patients and the PFS score was on average 2.6 months, and median OS was six months. Importantly, treatment responses showed no association with HPV status or PD-L1 expression level [448].

Interestingly, the biomarker of programmed cell death ligand-1 (PD-L1) could predict the treatment efficacy and prognosis in head and neck cancer patients. The meta-analysis based on recent clinical trials revealed that the high/positive expression of PD-L1 predicted better 6- and 12-month OS in HNSCC (RR 1.30; 95% CI: 1.02–1.65, *p* = 0.03; and RR 1.31, 95% CI: 1.05–1.62, *p* = 0.01). Furthermore, PD-L1 positivity were also connected to higher ORR in the study population who had treatment of PD-1/PD-L1 inhibitors compared to negative expression of the PD-L1 molecule (RR 1.84; 95% CI: 1.41–2.41, *p* < 0.00001) [449].

##### CTLA-4 Inhibitors

Targeting CTLA-4 as an immune checkpoint inhibition constitutes a new type of immunotherapeutic approach. CTLA-4 is a molecule that can bind to B7, thus preventing the interaction between B7 and the co-stimulatory molecule CD28 and inhibiting T cell proliferation and the production of functional IL-2 [450,451]. On the basis of several preclinical murine models showing improved tumour control after CTLA-4 blockade, anti-CTLA-4 human mAbs were constructed. In HNSCC patients, two CTLA-4-targeting antibodies have been developed: ipilimumab (IgG1k mAb, also known as MDX-010, MDX-101 and BMS-734016) and tremelimumab (IgG2 mAb, previously known as ticilimumab or CP-675,206). mAb, ipilimumab and tremelimumab are currently being tested and verified in phase II and III clinical studies in combination with anti-PD-1 (nivolumab) and anti-PD-L1 (durvalumab) blockade in R/M HNSCCs [43].

Intracellular immune checkpoint receptors PD-1/PD-L1 and CTLA-4 signalling and functional effects of their activity and blockage by ICIs in HNSCC are shown in Figure 3A.

##### ICI Combination Therapy

The dual checkpoint blockade of PD-1/PD-L1 and CTLA-4 in advanced head and neck cancer patients and low or no PD-L1 tumour cell expression is currently being investigated in different studies, e.g., CheckMate-651 (NCT02741570), CONDOR (NCT02319044), EAGLE (NCT02369874) and KESTREL (NCT02551159) [452,453,454,455,456,457,458].

The most promising combination appears to be ipilimumab plus nivolumab, a PD-1-blocking antibody, which is currently being tested in a phase III randomized trial CheckMate-651 (NCT02741570). This clinical study evaluated nivolumab plus ipilimumab vs. EXTREME regimen (cetuximab plus cisplatin/carboplatin plus fluorouracil, then cetuximab maintenance) for platinum-eligible R/M HNSCCs. Unfortunately, in conclusions, N+I treatment vs. EXTREME did not show significant OS improvement in all randomized (HR 0.95; 95% CI: 0.80–1.13, *p* = 0.49) [458].

The CONDOR (NCT02319044) clinical trial was a phase II, randomized, open-label study of durvalumab, tremelimumab and durvalumab in combination with tremelimumab in patients with R/M HNSCC. ORR was in average 7.8%, 9.2% and 1.6% for combination arm, durvalumab monotherapy and for tremelimumab monotherapy, respectively. Furthermore, no significant improvement in OS was found for any patients treated [457]. Other phase III studies, such as KESTREL (NCT02551159) and EAGLE (NCT02369874) showed no statistically significant differences in OS between durvalumab or Durvalumab plus tremelimumab vs. standard of care for first-line treatment and second-line treatment for R/M HNSCC patients. However, durvalumab was associated with higher survival rates at 12 to 24 months and better response rates (AstraZeneca https://www.astrazeneca.com/media-center/press-releases/2021/update-on-kestrel-phase-iii-trial-for-imfinzi.html, accessed on 22 September 2022) [453,454].

Reference should also be made to clinical studies investigating the use of checkpoint inhibitor combinations with standard therapies. Several trials are underway to establish the safety and efficacy of ICI combinations with curative-intent chemoradiotherapy in HNSCC [459]. An example is the ongoing phase I/II DUCRO study (NCT03051906) analysing the use of the anti-PD-L1 monoclonal antibody, durvalumab, in combination with cetuximab and radiotherapy in locally advanced HNSCC in a cohort of patients with HPV-positive and HPV-negative HNSCCs. Equally interesting is the phase II/III study EA 3161 (NCT03811015) led by oncologists from the ECOG-ACRIN study group, which evaluates the effectiveness of maintenance treatment with nivolumab after the end of standard cisplatin chemoradiotherapy, in the case of intermediate risk, HPV-positive oropharyngeal cancer (OPC). Currently, recruitment is also underway for a phase II study (NCT03799445) to assess the effectiveness of ipilimumab, nivolumab and radiotherapy in locally advanced HPV-positive OPC [459].

##### Other Checkpoint Modulator Inhibitors

Preliminary clinical trials are currently underway to block other checkpoint modulators, including studies of inhibitory antibodies (such as LAG-3, TIGIT and TIM-3) and agonist antibodies (such as OX-40, CD40, ICOS, 4-1BB and GITR) (ClinicalTrial.gov, representative identifiers, NCT03489369, NCT02559024, NCT03123783, NCT02520791, NCT03364348 and NCT04021043) [299,434,459,460,461].

In humans, LAG-3 is expressed on CD8^+^ tumour-infiltrating lymphocytes (TILs) and peripheral regulatory T cells (T_regs_). This protein was found to be simultaneously co-expressed with other targets, such as PD-L1, TIGIT and TIM-3, in preclinical settings [462]. Research on the effect of blocking LAG-3 alone did not show the restoration of T cell exhaustion, however, the use of the combination of LAG-3/PD-1 mAbs resulted in reduced tumour volume [463]. Recently, a phase I/IIa clinical expansion study CA224-020 (NCT01968109) has been initiated to support the safety, tolerability and effectiveness of anti-LAG-3 IgG4 mAb (BMS-986016 or relatlimab) administered alone and in combination with anti-PD-1 mAb nivolumab in patients with solid tumours; this group includes HNSCCs that have spread and/or cannot be removed by surgery. This clinical trial assumes that dual blockage of PD-1 and LAG-3 can reverse T cell function. Indeed, preliminary results of previous studies indicate that LAG-3 inhibits the proliferation and activation of CD4^+^ and CD8^+^ T cells, and is also expressed on T_regs_, where it facilitates their suppressive function [464,465]. Another study (NCT04326257) is also currently underway on the use of relatlimab in combination with nivolumab in the treatment of recurrent and/or metastatic squamous cell carcinoma of the head and neck. In this phase II clinical trial of personalized immunotherapy in advanced head and neck cancers, gene expression of LAG-3 and CTLA-4 by RNAseq is determined to select the appropriate agent (ipilimumab or relatlimab) to add to nivolumab in patients with R/M HNSCC who have failed prior immunotherapy with anti-PD-1 or PD-L1 mAb therapy. The agent, either ipilimumab or relatlimab is chosen based on the highest relevant immune gene expression (CTLA-4 or LAG-3), as long as the minimum difference required is met [459]. A recurrent or metastatic HNSCC cohort will be also recruited in the TACTI-002 (two active immunotherapies) phase II study (NCT03625323) to assess the safety and efficacy of a combination of soluble LAG-3 fusion protein Eftilagimod alpha (IMP321) and pembrolizumab. Eftilagimod alpha constitutes a soluble molecule that is a mediator of antigen presenting cell (APC) activation and CD8^+^ T cell activation by binding to a subset of MHC class II molecules. Preliminary results of the study confirm the occurrence of a partial immune response in the analysed cohort of patients. The study also indicates a safe and effective anti-cancer effect in the second-line setting for R/M HNSCCs [466].

TIM-3 is a co-receptor belonging to the family of new immune checkpoint molecules that play important and complex roles in regulating immune responses and in inducing immune tolerance. TIM-3 is expressed by human Th_1_ and Th_17_ cells, as well as other immune cells encompassing dendritic cells (DCs), macrophages (TAMs), natural killer (NK) cells, and cytotoxic T cells; its signalling inhibits the proliferation and reduces the secretion of effector cytokines by T cells and the reduction of apoptosis of effector T cells [467,468]. TIM-3 mediates its suppressive activity on immune cells via its several ligands, including C-type galectin-9 (Gal-9), phosphatidylserine (PtdSer), carcinoembryonic antigen-related cell adhesion molecule 1 (CEACAM1) and high-mobility group protein 1 (HMGB1) [469]. Several clinical trials are currently underway to assess the safety and efficacy of TIM-3 inhibition, including a phase I study (AMBER) evaluating TSR-022, TIM-3 mAb in advanced solid tumours (NCT02817633) and a clinical trial evaluating MBG453, a humanized IgG4 antibody with high affinity antibodies against TIM-3 (NCT02608268). This is a first in-human study evaluating the anti-T cell immunoglobulin and TIM-3 antibody TSR-022 in HNSCCs. The second is a phase I-Ib/II study of MBG453 as single agent and in combination with PDR001 (anti-PD1) in patients with advanced malignancies, including HNSCCs. The purpose of this first in-human study of MBG453 was to characterise the safety, tolerability, pharmacokinetics, pharmacodynamics and anti-tumour activity of MBG453 administered intravenously [459].

Another new therapeutic agent is the human anti-TIGIT mAb tiragolumab (NCT04665843), which is currently being evaluated in the SKYSCRAPER-09 phase II interventional clinical trial. The primary objective of this study is to evaluate the efficacy of atezolizumab plus tiragolumab and atezolizumab plus placebo as first-line treatment in recurrent/metastatic PD-L1-positive squamous cell carcinoma of the head and neck on the basis of the confirmed objective response rate. It will also evaluate the safety, pharmacokinetics and immunogenicity of both atezolizumab and tiragolumab [459].

Other costimulatory receptors are a subtype of second signalling molecules that promote T cell receptor pathways and include important molecules belonging to the tumour necrosis factor (TNF) family of receptors, such as protein OX40. Other early phase studies of human OX40 agonists in patients with various advanced HNSCCs include NCT02274155 and NCT02315066. The first phase Ib clinical trial (NCT02274155) investigates the role of MEDI6469, a murine anti-OX40 antibody, in patients with locally advanced HNSCC prior to definitive surgical resection. Administration of MEDI6469 resulted in an increase in the proliferation of tumour-infiltrating T cells, both CD4^+^ and CD8^+^, in tumour biopsies before and after surgery in patients with stage III and IV oral head and neck squamous cell carcinoma (OPC). Another interesting dose-escalation/dose-expansion phase I clinical trial (NCT02315066) is investigating the effect of administration of ivuxolimab (PF-04518600), an OX40 agonist, alone/or in combination with utomilumab (PF-05082566), a 4-1BB agonist in patients with advanced or metastatic cancers, including HNSCC. Initial results support that ivuxolimab plus utomilumab are well tolerated treatment options and demonstrated preliminary anti-tumour activity in selected groups of patients [459,470,471]. Currently, the data of a phase I/II multicentre study (NCT02304393) designed to assess the safety, pharmacokinetics, pharmacodynamics and activity of the CD40 agonist selicrelumab administered in combination with atezolizumab (anti PD-L1) in participants with metastatic or locally in participants with locally advanced and/or metastatic solid tumours [48]. Another phase I study (NCT03329950) evaluated the immunological benefits of the CD40 agonist CDX-1140 against advanced solid tumours, including HNSCC, either alone or in combination with pembrolizumab, chemotherapy or recombinant fms type 3 tyrosine kinase ligand (Flt-3) CDX-301. Furthermore, IPI-549, a selective PI3Kγ inhibitor in combination with nivolumab, showed clinical activity in a phase I study in solid tumours [472]. This therapeutic agent is currently being evaluated as monotherapy also in patients with locally advanced HPV*^+ve^* and HPV*^−ve^* resectable HNSCC (NCT03795610) [48].

The changes in the signalling and functional effects of novel intracellular immune checkpoint receptors LAG-3, TIGIT and TIM-3, following blockade by ICIs in HNSCC are shown in Figure 3B.

#### 3.2.2. Therapeutic Strategies for TME-Directed Therapy—Targeting TAMs, MDSCs, T_regs_, CAFs and NK Cells

##### Strategies for TAMs

Current immunotherapeutic strategies seek to overcome the barriers of immune suppression in HNSCC patients, which is expected to improve the effectiveness of immunomodulatory methods that shape a therapeutically controlled immune response. Since tumour associated macrophages (TAMs) have a multidimensional role in the formation of immunosuppressive TME in HNSCCs, targeting TAMs has become an interesting treatment strategy to inhibit TAM recruitment, or to reverse TAM polarity from the immunosuppressive activity (M2) to a phenotype with tumoricidal properties (M1). For example, clinical trials using targeted therapy with CCL2/CCR2 axis inhibitors, molecules correlated with high infiltration of TAMs in cancer biopsies, such as anti-CCL2 carlumab and anti-CCR2 PF04136309 used in combination with chemotherapy (FOLFIRINOX) (NCT01413022) have shown promising results in solid tumours [473,474]. Blockade of CSF1R, a growth factor involved in TAM migration and recruitment, has been also proposed as a therapeutic strategy. A new CSF1R inhibitor, emactuzumab, is currently being studied in combination with ICI atezolizumab in a phase I study in advanced solid tumours (NCT0232319). Pexidartinib, also a new CSF1R inhibitor that is currently FDA approved, is also being evaluated in advanced solid tumours in a phase I study (NCT02734433) [475,476]. Unfortunately, the use of these promising molecules has not been studied in HNSCC.

A second therapeutic strategy that aims to eliminate TAMs is the repolarization of TAMs through targeting toll-like receptor (TLR), CD40 and PI3Kγ [477]. The use of TLR7 and TLR9 agonists injected into the tumour in combination with PD-1 blockade to improve the efficacy of ICIs, in experimental animal models of HNSCC was associated with an increased M1/M2 ratio and higher recruitment of tumour-specific IFNγ-producing CD8^+^ T cells (CTL), leading to suppression of cancer growth and prevention of metastasis in HNSCC models [100,478]. Currently, a phase I/IIb clinical trial (NCT02521870) is also underway on the use of the TLR9 agonist SD-101 administered intratumorally in combination with pembrolizumab (anti-PD-1 inhibitor) in naïve patients with recurrent or metastatic previously untreated HNSCC. This combination treatment shows good therapeutic toleration and early promising data, revealing a 48% DCR in this cohort [479]. Another ongoing phase Ib clinical trial (NCT03906526) is evaluating the effect of the combination of nivolumab (anti-PD-1 inhibitor) and the TLR8 agonist motolimod (VTX-2337) on the immune response as measured by CD3^+^CD8^+^ T cell count before and after surgery in subjects with resectable HNSCC [48]. Other mechanisms of inhibition of TAMs on promoting tumour progression have also been discussed. Other novel approaches have been used to effectively target TAMs (NCT03329950, NCT03795610), e.g., using the CD40 agonist or PI3Kγ inhibitor in combination with ICIs (nivolumab and pembrolizumab) to increase the efficacy of immunological therapies in HNSCC patients [48].

##### Strategies for MDSCs

Due to the pluralistic role of MDSCs in oncogenesis, various therapeutic approaches targeting MDSCs are being explored. Clinical trials are mainly focused on preventing MDSC differentiation, inhibiting MDSC activation or blocking MDSC immunosuppression, which occurs through expression of IDO-1 and the chemokine receptor CXCR2 [480,481]. A phase I/II multicentre trial (ECHO-202/KEYNOTE-037) performed also in HNSCC subjects, is evaluating treatment with epacadostat, a highly selective IDO-1 enzyme inhibitor plus pembrolizumab (anti-PD-1) in patients with advanced solid tumours. As a result, one of the two patients with HNSCC included in this trial had a stable disease as best response, according to the response evaluation criteria in solid tumours Version 1.1 (RECIST v1.1) [379]. Furthermore, a phase I clinical trial showed that the combination of the IDO-1 inhibitor navoximod (GDC-0919) with atezolizumab (PD-L1 inhibitor) is characterized by good tolerability and acceptable anti-tumour activity in patients with solid tumours, including one patient with HNSCC [482]. Another intervention phase III study Keynote 669/Echo 304 (ClinicalTrials.gov identifier: NCT03358472) evaluating the combination of epacadostat with pembrolizumab and pembrolizumab monotherapy versus the standard EXTREME regimen as first-line treatment for R/M HNSCC, has been completed and the results of the studies are currently awaited. Additionally, a phase II trial of nivolumab (anti-PD-1) and an oral IDO-1 inhibitor BMS986205 (NCT03854032) is currently underway for the treatment of patients with stage II-IV HNSCC [48].

Another strategy for eliminating NK cell-suppressive activity of MDSCs is based on inhibiting MDSC trafficking with SX-682, a CXCR1/2 inhibitor; this can be used to enhance NK cell immunotherapy in head and neck cancer models [483]. Another promising therapeutic approach is the inhibition of phosphodiesterase 5 (PDE5), which functionally inactivates MDSCs. In a phase II study conducted in patients with HNSCC, it was shown that the use of the PDE5 inhibitor tadalafil augments general and tumour-specific immunity and has therapeutic potential in HNSCC patients. Its use was associated with reverse cancer specific immune suppression, promotion of T cell expansion and reduction of MDSC activity in the periphery [484,485]. A phase II multicentre, randomized, prospective trial (NCT01697800) is currently underway to determine the optimal timing and design of PDE5 anti-cancer immunotherapy in combination with conventional HNSCC therapy [48,485]. The ability of tadalafil to enhance the efficacy of the mucin 1 (MUC1) cancer vaccine by inhibiting PDE5 in HNSCC patients is also under investigation (NCT02544880). The researchers hypothesize that by lowering MDSCs and regulatory T cells (T_regs_), tadalafil treatment can prime an anti-tumour immune response and promote a permissive environment; this should increase the efficacy of anti-tumour vaccines in minimal residual disease [486].

##### Strategies for CD4^+^CD25^+^Foxp3^+^T_regs_

Therapeutic strategies that effectively target CD4^+^CD25^+^Foxp3^+^T_regs_ regulatory cells rely mainly on blocking surface markers, such as CD25, Foxp3, OX40, TIM-3 and GITR molecules, as well as inhibiting their immunosuppressive cytokine production. The results of many studies indicate complex mechanisms of immune regulation that favour the initiation and progression of tumours with the participation of T_regs_. Therefore, several investigators propose new approaches to targeting regulatory cells and the therapeutic potential of sequential therapy to overcome tumour resistance mechanisms. For example, radiation therapy (RT) can alter the immune microenvironment and render poorly immunogenic neoplasms sensitive to PD-L1 inhibition [487]. Resistance to RT and PD-L1 blockade have been explored in a murine model of HNSCC, in which that the immune checkpoint receptor, T cell immunoglobulin mucin-3 (TIM-3), a negative regulator of Th_1_ immunity, was found to be expressed by a fraction of TIL CD3^+^CD8^+^ T cells and TIL CD4^+^CD25^+^Foxp3^+^T_regs_ in cancers treated with RT and anti-PD-L1 inhibitors. Treatment with anti-TIM-3 concurrently with anti-PD-L1 plus RT focused on decreased tumour growth, enhanced T cell cytotoxicity, decreased T_regs_ activity and improved survival in animal models of HNSCC. Unfortunately, the response to combination treatment was not sustainable, and recurrence analysis showed T_regs_ repopulation [487]. Moreover, since Tim-3^+^ T_regs_ are functionally and phenotypically different in HNSCC TILs, they are highly effective in inhibiting T cell proliferation, despite the high expression of PD-1. Interestingly, IFN-γ induced by anti-PD-1 immunotherapy may be useful by reversing the suppression of Tim-3^+^ T_regs_ [125].

It is difficult to identify successful tries to target the CD25 molecule in HNSCC and other tumours because it is not selectively expressed on T_regs_, and treatment with anti-CD25 antibodies can inhibit effector cell activity, interfering with the therapeutic target. An example is the phase I/I study of melanoma patients, where administration of the anti-CD25 mAb daclizumab, in combination with a dendritic vaccine, did not result in any clinical response [488].

Another experimental therapeutic target could be methyl-binding domain (Mbd) proteins targeting CpG dinucleotides in the T_reg_-specific demethylation region (TSDR). TSDR naturally undergoes reversal demethylation, a process that mediates the stability of the Foxp3 antigen. Experimental data indicate that Mbd2 has a key role in promoting TSDR demethylation, Foxp3 expression and T_reg_-suppressive function [489]. Furthermore, the therapeutic use of the recently developed antisense oligonucleotide inhibitor Foxp3 (AZD8701) induced enhanced activity in preclinical mouse models, both alone and in combination with immune checkpoint blockade [490]. For example, one ongoing study (NCT04504669) is currently evaluating the safety, tolerability, pharmacokinetics, pharmacodynamics, immunogenicity and anti-tumour activity of anti-Foxp3 inhibitor AZD8701 alone and in combination with durvalumab (MEDI4736) in patients with selected advanced solid tumours, including HNSCC.

Trials targeting receptors for chemokines that regulate CD4^+^CD25^+^Foxp3^+^T_reg_ chemotaxis and function, such as CCR4, represent another promising approach for the treatment of solid tumours. The inhibition of the CCR4 molecule by a humanized anti-CCR4 monoclonal antibody mogamulizumab (KW-0761), with antibody-dependent cellular cytotoxicity activity, was successful in depleting T_reg_ activity in peripheral blood and 40% disease control in a phase Ia study of FoxP3^+^ CD4^+^ T_reg_ depletion by infusion of KW-0761 in solid cancer patients, including oesophageal tumours [491,492]. In summary, the combined use of KW-0761 to deplete T_regs_ and other immunotherapies is a promising approach to augment immune responses in HNSCCs. Unfortunately, in a multicentre, phase I, dose-escalation study (NCT02301130) of the immunological efficacy of the combination of KW-0761 with durvalumab (anti-PD-L1 mAb) or tremelimumab (anti-CTLA-4 mAb), mogamulizumab did not show significant efficacy in patients with advanced solid tumours, including HNSCC. Although it should be noted that mogamulizumab treatment led to almost complete depletion of peripheral T_regs_, as well as a reduction of intratumoral T_regs_ in the majority of patients [493]. Researchers also evaluated the safety and efficacy of a combination of mogamulizumab and nivolumab (anti-PD-1) in previously untreated patients with advanced/metastatic solid tumours. During combination treatment (NCT02476123), populations of effector CD4^+^CD25^+^Foxp3^+high^ T_reg_ decreased and CD3^+^CD8^+^ T cells in TILs increased. Simultaneous administration of an anti-PD-1 antibody, nivolumab, with a T_reg_-depleting anti-CCR4 mAb, mogamulizumab, provides an acceptable safety profile, anti-tumour activity, and a potentially effective option in cancer immunotherapy [494].

OX40 is a costimulatory receptor that promotes the proliferation and activity of TILs and enhances the memory of cytotoxic T cells in solid tumours, including HNSCC [495]. For instance, in a phase Ib clinical trial (NCT02274155), neoadjuvant anti-OX40 (MEDI6469) therapy in patients with HNSCCs led to the activation and expansion of antigen-specific tumour-infiltrating T cells [496,497]. The data provided evidence that an agonistic humanized anti-OX40 mAb used prior to surgery was therapeutically safe and increased activation and proliferation of CD4^+^ and CD8^+^ T cells in the blood and tumour milieu. Furthermore, the study suggested that increases in the tumour-reactive CD103^+^CD39^+^CD8^+^ TIL could serve as a potential biomarker of anti-OX40 clinical activity [496,497]. The results of the phase I study (NCT02205333) with MEDI6469 in monotherapy or in combination with tremelimumab or durvalumab in metastatic solid tumours are also currently awaited. In another phase I study (NCT02318394), the antibody MEDI0562 showed promising therapeutic efficacy through increased proliferation of effector T cells (increased Ki67^+^ CD4^+^ and CD8^+^ memory T cell in the periphery) and decreased intratumoral OX40^+^ CD4^+^CD25^+^Foxp3^+high^ T_reg_ cells in patients with advanced solid tumours, including HNSCC [498]. Another clinical trial (NCT03336606) will also evaluate the safety and feasibility of a humanized OX40 agonist, MEDI0562, in the neoadjuvant setting in the pre-operative setting for patients with HNSCCs [48]. Furthermore, a new human agonistic OX40 antibody, MEDI16383 (NCT02221960), alone or together with MEDI4736, is also currently being tested in a phase I clinical trial in patients with selected recurrent or metastatic solid tumours [48]. Of interest is a first in-human, open-label, non-randomized, 4-part phase I trial (NCT04198766) constructed to determine the safety profile and identify the maximum tolerated dose (MTD) and/or recommended phase II dose (RP2D) of INBRX-106, hexavalent OX40 agonist antibody, administered as a single agent or in combination with the anti-PD-1 pembrolizumab [48].

Other strategies targeting T_regs_ include stimulation of the glucocorticoid-induced TNFR-related protein (GITR) costimulatory receptor, which is ubiquitously expressed on regulatory CD4^+^CD25^+^Foxp3^+^ T_reg_ cells. T cell stimulation through GITR attenuates T_reg_-mediated suppression or enhances tumour-killing by CD4^+^ and CD8^+^ effector T cells, including those secreting IFN-γ, or both. Administration of anti-GITR mAb in animal models resulted in tumour regression and increased infiltration of effector T cells [499]. MEDI1873, a novel GITR-ligand agonist (IgG1 mAb) was recently assessed in a phase I study in patients with advanced solid tumours, including HNSCC. Another example is the novel anti-GITR antibody, REGN6569, which is currently being tested (NCT04465487) in combination with the PD-1 inhibitor cemiplimab in patients with advanced solid tumours, including HNSCCs [48].

##### Strategies for CAFs

Due to the dominance of CAF in TME, a phenotypically and functionally heterogeneous population of immune cells, CAF-targeted therapies represent a new challenge for researchers. Because CAFs regulate the biology of cancer cells and other stromal cells through cell-cell contact, releasing numerous regulatory factors, and synthesizing and remodelling the extracellular matrix, CAFs are key regulators of cancer initiation, progression and development. The premise of CAF-targeted anti-cancer therapies is generally to deplete CAF activity by genetic deletion or pharmaceutical inhibition of cell surface markers. Inhibition of CAF function can also be accomplished by targeting chemokines and CAF-derived ECMs. Moreover, there are attempts to use CAFs as drug delivery vehicles, such as oncolytic adenoviruses [235,236,500,501]. One example of a target is the fibroblast activation protein (FAP), which is overexpressed in activated fibroblasts of the tumour microenvironment, where it plays a key role in ECM remodelling and regulates CAF proliferation and migration [502,503]. In HNSCC xenograft models, treatment with a novel anti-FAP monoclonal antibody-maytansinoid conjugate (FAP5-DM1) resulted in sustained suppression of tumour progression with excellent efficacy and tolerability. Furthermore, a phase I study with the use of RO6874281, an immunocytokine containing of an interleukin-2 variant (IL-2v) fused with an anti-FAP antibody targeted to tumour-associated fibroblasts via binding to FAP, showed a significant increase in the activity of T and NK cells and caused a durable response in one patient with HNSC [504]. Patients are also currently recruiting for an open-label, multicentre phase II study (NCT03386721) evaluating RO6874281 in combination with atezolizumab (MPDL3280A) in ICI-naïve or previously treated, advanced and/or metastatic solid tumours, including HNSCC. Attempts have also been made with the anti-FAP monoclonal antibody sibrotuzumab, but this strategy has not yielded satisfactory results in human cancers, including HNSCCs [48].

Targeting cytokines and other transcriptional factors, such as IL-6 and JAK2/STAT3 pathways is also a promising anti-CAF activity approach. Unfortunately, despite promising results for the chimeric anti-interleukin (IL)-6 mAb (siltuximab) in preclinical models, a satisfactory response was not demonstrated in a phase I/II clinical trial involving patients with advanced solid tumours, including HNSCC. Moreover, serious adverse events were reported in 42% of the population [505]. Another study using C188-9, a small inhibitor of STAT3, showed tumour regression in mice xenografted with radiation-resistant HNSCC lines. A phase I study (NCT03195699) on the therapeutic effect of C188-9 in advanced solid tumours, including HNSCC, are currently underway [48,506]. JAK2 inhibitors have also been shown as a promising strategy for reducing tumour progression in xenografts from patients with HNSCCs. An example is a clinical trial with ruxolitinib (NCT03153982), a JAK1/2 inhibitor that is currently being evaluated in operable HNSCC patients who are planned for definitive surgery. Other studies have examined the potential of targeting matrix components and matrix metalloproteases for anti-cancer therapy. For example, in a phase I pharmacokinetic study of a potent oral noncytotoxic inhibitor of MMPs, primarily MMP-2 and MMP-9 (S-3304), the drug was shown to be safe and well tolerated, but no objective changes in systemic and local immune responses were identified. However, among seven patients with stable disease, one had HNSCC and S-3304 achieved plasma concentrations above those required to inhibit MMP-2 and MMP-9 [507].

The functional effects of CAF cell activity and therapeutic strategies for targeting CAFs in HNSCC are given in Figure 4A.

##### Strategies for NK Cells

As the peripheral blood of HNSCC patients has a reduced population of immunoregulatory CD56^bright^ NK cells and an increased number of TIM-3^+^ and NKG2A^+^ NK cells compared to healthy donors, attempts are being made to increase the anti-cancer activity of the specific NK cell subpopulation and modulate NK cell activity [508,509,510]. Natural killer cells are potential highly effective tumour suppressor immune cells because they can kill cancer cells when activating signals outweigh inhibitory signals. Therefore, various strategies have been used in many clinical and preclinical studies to increase NK cell activity against solid tumours [511]. Unfortunately, the immunosuppressive TME can also effectively inhibit NK cell function, tipping the balance towards pro-cancer pathways. A very important factor in the control of tumour development and progression is the NKG2D molecule and its ligand HLA-E. Because neoplasms are able to release NKG2D ligands MICA/B and ULBP1/2/3 by proteolytic cleavage, and thus evade NKG2D-mediated NK cell cytotoxicity [283,512,513]. The observed functional activity of NK cells within the tumour was reduced, reflecting a dominant inhibitory phenotype, also accompanied by enrichment for PD-1^+^ TIGIT^+^ NK cells and a reduction in CD16, a receptor capable to bind the antibody Fc fragment, thus enabling NK cells to perform ADCC. Therefore, a novel therapeutic strategy being tested in animal tumour models is to use antibodies targeting the proteolytic cleavage site to prevent MICA/B release and reduce cancer spread [513,514]. This immune evasion mechanism has also been described in HNSCC. Preclinical studies indicate that increased plasma levels of soluble NKG2D ligands in HNSCC patients, if depleted, restore the ability to kill the cell’s target in vitro [515,516]. Therefore, blocking NKG2A by anti-NKG2A mAb, monalizumab is an effective strategy for increasing the anti-tumour activity of NK and CD8^+^ T cells in patients with HNSCCs. The use of monalizumab was associated with high levels of HLA-E in primary tumour cells obtained from four patients with HNSCC [517,518].

ADCC phenomenon is triggered by cetuximab, an anti EGFR antibody, which is widely used in cancer immunotherapy, including also locally advanced or recurrent/metastatic HNSCC treated with platinum-based chemotherapy plus cetuximab or panitumumab. The use of cetuximab stimulates the CD16 receptor, activating peripheral NK cells and inducing the secretion of IFN-γ [519,520].

Other studies have also shown that patients with more TIGIT-expressing (T cell immunoglobulin and ITIM domain) natural killer cells produced less IFN-γ in peripheral blood and in the TME. This is due to the activity of TIGIT, which is a co-inhibitory receptor on the surface of the immune cell that binds CD155 and CD112 ligands [521,522]. One therapeutic strategy is to block the interaction of TIGIT with its ligands to reduce the effect of NK cell depletion in vitro and in a mouse model in solid tumours [293,523]. Moreover, several anti-TIGIT monoclonal antibodies are currently under development in phase I/II studies [524].

Increased amounts of GITR^+^ NK cells have been found in TIL in many solid tumours, including HNSCC [525,526]. GITR (glucocorticoid-induced tumour-necrosis-factor-receptor-related protein) is a costimulatory receptor that has a dual anti-tumour effect by promoting the survival and expansion of activated T cells and simultaneously inhibiting the activity of the CD4^+^CD25^+^ regulatory T_regs_ cell population in tumours [527]. The GITR molecule is therefore an important immunomodulatory receptor, which is a therapeutic target whose activity may lead to a decrease in the suppressive potential of T_regs_ inside the tumour and an increase in anti-tumour immunity. Several pre- and clinical trials are testing GITR agonist antibodies, e.g., a FcγR-binding, mIgG2a anti-GITR antibody in mouse tumour models, or humanized GITR agonists in combination with immune checkpoint inhibitors (ICIs) in advanced human cancers. An example is the first in-human phase I trial (NCT01239134) of GITR agonism with the anti-GITR antibody (TRX518) plus PD-1 pathway blockade in patients with advanced tumours. The data demonstrated that T cell reinvigoration with PD-1 blockade can overcome resistance of advanced tumours to anti-GITR monotherapy [528]. Another phase Ib study using the GITR agonist (TRX518) alone and in combination with gemcitabine, pembrolizumab or nivolumab showed that TRX518 monotherapy + anti-PD1 combination was associated with intratumoral T_regs_ reductions and CD8^+^ increases and activation in patients with advanced solid tumours [529]. The safety, tolerability and potential clinical activity of a GITR agonist (BMS-986156) alone or in combination with nivolumab in patients with advanced solid tumours was also a focus of a phase I/IIa dose-escalation and cohort-extension clinical trial (NCT02598960). Based on this study, it was confirmed that BMS-986156 had a satisfactory safety profile, and BMS-986156 therapy plus nivolumab showed high safety and effectiveness [530]. The safety and preliminary efficacy of other agonistic IgG1 fusion protein targeting GITR, were evaluated in an open-label, first in-human, phase I, dose-escalation study in previously treated patients with advanced solid tumours. Unfortunately, despite prolonged stabilization of the disease in some patients, further progression was not planned due to the lack of demonstrated tumour response [531]. In conclusion, due to the increased prevalence of GITR^+^ NK cells, attempts to use a GITR agonist in combination with PD-1 and/or TIGIT blocking antibodies may be a good future therapeutic strategy in HNSCC patients [513]. Increased proportions of CXCR6 were found in intratumoral NK and T cells, probably caused by ligand-receptor chemotaxis of CXCL16/CXCR6. Therefore, these molecules may become an interesting target in adoptive anti-cancer therapy for higher natural killer cell frequency and increased ADCC activity [513]. Furthermore, ongoing phase I/II clinical trials in a variety of human cancers are also being performed using IL-2-stimulated donor allogeneic NK cells. Future research to improve NK cell immunotherapy against various malignancies also includes research to enhance NK cell cytotoxicity by activation with a combination of cytokines, such as IL-2/IL-15, or by cross-contact with dendritic cells [284,532].

Intracellular signalling and functional effects of NK cell activity, as well as therapeutic strategies related to NK cells in HNSCC is shown in Figure 4B.

#### 3.2.3. Adoptive Cell Therapy (ADP) and CAR-T Cell Therapy in Patients with HNSCC

Advances in cell engineering and culture approaches to enable efficient gene transfer and ex vivo cell expansion have facilitated the use of novel strategies of cancer immunotherapy in clinical application or in clinical trials. One such approach is adoptive cell therapy (ADP), including TCR/CAR-T transfer and CAR-NK cell adoptive transfer. This specific immune cell therapy involves inducing autologous immune cells with enhanced anti-cancer properties through their isolation from a patient, transduced ex vivo to express or target a specific antigen, and then infused back to the same patient. These immunocompetent cells can be derived from naturally occurring tumour-infiltrating T/NK cells, or genetically synthetic-biology-based engineered T cells recognizing tumour antigens, such as T cell receptor (TCR), or chimeric antigen receptor (CAR), which is composed of antibody binding domains fused to T cell signalling domains [533,534,535,536,537]. Unlike CAR-T cells, the TCR-T cells arise from peripheral blood T cells rather than from tumour-infiltrating T cells, and can recognize the intracellular antigens.

Strategies for use of anti-cancer T cell-based therapy can be performed by ex vivo manipulation of T cells through ACT of unmodified (TILs) or genetically modified T cells (TCRs, CARs) and by in vivo manipulation of T cells using antibodies (bispecific and checkpoint inhibitors). These approaches may induce monoclonal (TCRs, CARs, bispecific antibodies) or polyclonal (TILs, checkpoint inhibitors) anti-tumour T cells. Targeted by exogenously transduced TCRs or CARs, active T cells can destroy tumour cells expressing specific antigens in an MHC-dependent (TCR) or independent (CAR) manner. In addition, the host can be manipulated before cell transfer to eliminate suppressor cells, such as T_regs_ and MDSCs, and promote in vivo expansion of transferred lymphocytes by eliminating endogenous lymphocytes that compete for the same survival and stimulatory factors, i.e., IL-7 and IL-15 cytokines [538,539]. Unfortunately, many TCR-T trials in tumours of various origins have been associated with high toxicity. In contrast to TCR-T, the advantage of CAR-T technology is obvious. Moreover, the benefits of CAR-T cell therapy are related to inducing a secondary immune response against the tumour, even if the tumour does not uniformly express the primary target antigen. Indeed, the lysis of some cells by CARs can release tumour-specific neoantigens or epitopes that can be processed and presented by APCs to TIL cells. Unfortunately, CAR-T has limitations. In particular, its use on patients was frequently related to the cytokine release syndrome (CRS), a life-threatening complication, and its effectiveness against solid tumours is limited due to a lack of specific and reliable antigen targets on the tumour cells and the physical barrier represented by the TME [540,541,542,543,544].

In HPV*^+ve^* HNSCCs, MHC class I molecules process and present epitopes E6 and E7 and may constitute specific targets for T cell receptor (TCR) gene engineered T cells [537,545]. For example, a two-arm open-labelled phase I trial evaluating the use of HPV-E6-specific TCR-T cells, with or without the anti-PD1 auto-secreted element, in the treatment of HPV*^+ve^* HNSCCs is currently underway in HPV-positive HNSCC patients (NCT03578406). Engineered TCR-containing T cells (TCR-T) that can specifically recognize the presented HPV antigen have become a viable treatment for this type of cancer. Although modified T therapies have been well recognized in hematologic malignancies, solid cancer treatment has been a major obstacle due to the immunosuppressive tumour microenvironment. One of the key mechanisms of tumour-induced suppression is the PD-L1/PD-1 interaction, which induces T cell depletion. Therefore, TCR-T cells armed with a PD1 antagonist can further enhance the efficacy of TCR-T in solid tumours. Furthermore, the use of TCR-modified E7 T cells is currently being investigated in a phase I/II clinical trial in metastatic or refractory/recurrent human papillomavirus (HPV16^+^) cancers, including OPC (NCT02858310). T cells genetically engineered with a TCR targeting HPV-16 E7 (E7 TCR) display specific reactivity against HLA-A2^+^, HPV-16^+^ target cells (HLA-A*02:01-positive). The goal of this study was to evaluate the response to the administration of T cells engineered with the T cell receptor (TCR) gene and to determine the objective induction rates of tumour responses in certain malignancies, including HPV-16^+^ cancers [459].

Recent studies have attempted to use NK cell adoptive transfer immunotherapy against tumour cells. The benefits of adoptive NK transfer include the improvement of anti-cancer responses of autologous NK cells through cytokine stimulation, e.g., IL-2, IL-12, IL-15, IL-18 and IFN-γ type I, which enhance the function and proliferation of natural killer cells. This therapeutic intervention results in a population of highly activated NK cells with increased expression of adhesion molecules, cytokine-induced activating receptors (NKp44), perforin, granzymes, FasL, TRAIL and increased cytokine proliferation and production [546]. Importantly, the use of CAR-NK cells can provide a universal cell therapy without the need for human leukocyte antigen (HLA) matching or prior exposure to tumour-associated antigens (TAA). Indeed, CAR-NK cells represent a useful strategy for recognizing various antigens, maintaining greater proliferative capacity and persistence in vivo, exhibiting better infiltration into tumours, and the ability to overcome a resistant tumour microenvironment, leading to persistent tumour cytotoxicity. Moreover, CAR-NK can be used in the development of the “off-the-shelf” anti-cancer immunotherapeutic products [547].

In addition to these strategies to modulate CAR-NK cell intrinsic properties, there are increasing efforts to use other newer next generation approaches for CAR construct in the solid tumours. The second and third generation CAR constructs have one or more intracellular costimulatory domains to enhance signalling. The fourth generation constructs, called armoured CARs, contain molecular payloads that endow CAR-designed cells with additional functions. Multispecific targeting is also used in biological assays, which refers to a genetic engineering strategy to simultaneously target multiple tumour antigens. Another set of logic-gated CAR molecules are designed to express both the activating (aCAR) and the inhibitory (iCAR) CAR receptor on their surface. The aCAR can recognize tumour-specific epitopes and is able to initiate immune cell activation and target engagement, whereas iCAR detects epitopes present only on healthy tissues and inhibits the activation of immune cells. This strategy aims to integrate both signals so that CAR-modified logic-gated NK cells can kill cancer cells differently while sparing healthy tissues with common antigens. The role of bispecific killer engager (BiKe) is also interesting. Bispecific antibodies with binding sites for both FcRγIII (CD16A) and a tumour-specific epitope act as a specific molecular binder that joins and transports NK cells to the vicinity of tumour cells for target elimination in a manner similar to CARs [548].

#### 3.2.4. Therapeutic Vaccines in Patients with HNSCC

The use of cancer vaccines as a new anti-cancer strategy aims to induce or enhance a tumour-specific T cell immune response through active immunization. Research indicates that therapeutic vaccines are effective in activating anti-cancer immunity in humans. Moreover, vaccination combined with an immune checkpoint blockade is also a novel therapeutic strategy to achieve maximum benefit from cancer vaccines [549].

HPV-positive HNSCC contains non self-antigens, including viral neoantigens E6 and E7, which are specific to HPV-infected cells. Enhancing anti-HPV T cell activity is the goal of several HPV-targeted therapeutic vaccines currently under investigation in HPV-positive HNSCC. An example is the development of a DNA vaccine targeting the E6/E7 proteins of HPV-16 and HPV-18 for immunotherapy in combination with a recombinant vaccine boost and PD-1 antibody. The use of the pBI-11 DNA vaccine constructed via the addition of codon-optimized human papillomavirus 18 (HPV18), E7 and HPV16, and 18 E6 genes to the HPV16 E7-targeted DNA vaccine pNGVL4a-SigE7(detox) HSP70 (DNA vaccine pBI-1) has generated stronger therapeutic responses. Furthermore, a combination of the pBI-11 DNA and tissue-antigen HPV vaccine (TA-HPV) boosts vaccination with the PD-1 antibody blockade significantly improved the control of TC-1 tumours and extended the survival in the animal model [550].

Another phase Ib/II pilot study has analysed the safety, tolerability and immunogenicity of immunotherapy with MEDI0457 DNA immunotherapy targeting HPV16/18 E6/E7 with IL-12 encoding plasmids, dosed perioperatively or following completion of concurrent chemoradiotherapy. The vast majority of patients showed elevated antigen specific T cell activity and persistent cellular responses with induction of HPV-16-specific CD8^+^ T cells. MEDI0457 shifted the CD8^+^/Foxp3^+^ ratio and increased the number of perforin immune infiltrates in all patients [551]. MEDI0457 was also studied in combination with anti-PDL1 durvalumab in a phase Ib/IIa trial in patients with HPV-positive R/M HNSCC who had progressed on to at least one prior regimen. An increase in both tumoral CD8^+^ T cells and peripheral HPV-specific T cells was shown. Therefore, this approach may constitute a complementary strategy to PD-1/PD-L1 inhibition in HPV-associated HNSCCs to improve therapeutic outcomes [552].

The combination of the immune checkpoint blockade and tumour-specific vaccine was also investigated for patients with incurable human papillomavirus 16-related cancers in a phase 2 clinical trial (NCT02426892). The study supported that the efficacy of nivolumab, the anti-PD-1 immune checkpoint mAb, was improved by the administration of ISA 101, a synthetic HPV-16 long-peptide vaccine that induced HPV-specific T cells. The ORR of 33% and median OS of 17.5 months was promising compared with PD-1 inhibition alone in similar patients [553]. ISA-101 has also been studied in combination with utomilumab, a 4-1BB (CD-137, an inducible costimulatory receptor expressed on activated T and NK cells) agonist, in patients with incurable HPV-16-positive oropharyngeal cancer (NCT03258008) [554]. The use of CD137 agonist mAb urelumab also enhanced cetuximab (anti-EGFR)-activated NK-cell survival, DC maturation, and tumour antigen cross-presentation in patients with head and neck cancer. In neoadjuvant cetuximab-treated patients with HNSCC, upregulation of CD137 by intratumoral, cetuximab-activated NK cells correlated with FcγRIIIa V/F polymorphism and predicted clinical response. Moreover, immune biomarker modulation was observed in an open label, phase Ib clinical trial, of patients with HNSCC treated with cetuximab plus urelumab [555]. ISA-101 is also being analysed in a phase II trial (NCT04369937) in combination with pembrolizumab and cisplatin-based chemoradiation for subjects with newly diagnosed, locoregionally advanced, intermediate risk HPV^+ve^ HNSCC. The purpose of this interventional study is to evaluate patient survival following this combination of treatments [459].

The experimental vaccine, ADXS11-001, is currently being evaluated as an effective strategy in stimulating the immune defence system against HPV-positive oropharyngeal squamous cell carcinoma before transoral robotic surgery (NCT02002182). ADXS11-001 uses a live-attenuated Listeria monocytogenes (LM) vaccine encoding a HPV16 E7 oncoprotein interconnected to LM listeriolysin O. Unfortunately, preliminary results indicate a high rate (56%) of serious adverse events in the studied population [459].

An interesting strategy is also the phase Ib/II trial of TG4001 (NCT03260023), a therapeutic HPV vaccine, and its combination with avelumab in patients with recurrent/metastatic (R/M) HPV-16^+^ cancer. The majority of the subjects have displayed detectable peripheral responses against E6 and/or E7 oncoprotein after treatment, and an overall trend towards decreased circulating regulatory T cells (T_regs_). The percentage of PD-L1^+^ tumour cells doubled in all low or moderate expressors. The phenotypic features disclosed an increased expression of genes associated with both adaptive and innate immunity. TG4001 with avelumab was also shown to induce a shift to an immune-hot tumour gene signature with an increase of CD8^+^ infiltration, a decrease of infiltrated T_regs_/CD8^+^ ratio as well as an increase in the proportion of PD-L1^+^ cells in low to moderate PD-L1 expressing tumours [556].

The safety and immunogenicity of AMV002, a therapeutic HPV DNA vaccine, has also been investigated in a phase I dose escalation study (ACTRN12618000140257) for patients with prior treatment for HPV^+*ve*^ oropharyngeal carcinoma. AMV002 was well tolerated at all dose levels and offered enhanced specific immunity to virus-derived tumour-associated antigens in subjects [557]. In addition, the potential of DPX-E7, a vaccine based on the synthetic HPV16 peptide E711-19, is currently being investigated in an open-label phase Ib/II study (NCT02865135) in HLA-A*02-01 patients with HPV16-associated head and neck cancer. DPX-E7 therapy is a form of immunotherapy that induces an anti-tumour response by generating CD8^+^ T cells [459].

#### 3.2.5. Immunotherapy Based on Nanotechnology in Patients with HNSCC

The effectiveness of anti-cancer immunotherapies can often be reduced due to inter alia the induction of autoimmunity, nonspecific interaction among immune cells and immunostimulating agents, and impaired targeting of tumour cell when administered directly into the body; however, studies suggest that these limitations may be overcome by therapy based on nanotechnology [558]. Research shows that organic/inorganic nanoparticles, spherical forms with sizes ranging from 1 to 100 nm, are able to increase the efficiency of anti-tumour drugs, decrease their side effects, and control the distribution profile of drug molecules. They can also passively accumulate in high concentrations in solid tumours through the effect of increased permeability and retention (EPR) in tumour cells [559,560].

One of the directions for nanoparticle-based immunotherapy in HNSCCs is the use of these molecules in combination with checkpoint inhibitors (ICIs), which seems to be a promising way to improve the effects of immune checkpoint inhibitors, to achieve less toxic cancer treatment and improve the delivery of tumour antigens and adjuvants [561]. For example, the use of nanocomplex-based on gold nanoparticles (AuNPs) conjugated with anti-EGFR cetuximab by the immobilization of cetuximab on the AuNPs surfaces, demonstrated long-term stability, EGFR affinity and cancer cell death due to apoptosis of several cancer cells, including HNSCCs. Association of these drugs to nanocarriers have shown to be an effective way to reduce drug-toxicity and lower the occurrence of drug-related adverse effects [562]. The newly developed multifunctional nanocomposites could also be a promising candidate for the treatment of postoperative HNSCC. The gold nanocages (AuNCs) containing anti-programmed death 1 (PD-1) antibodies (αPD-1@AuNCs) can activate a strong anti-tumour immune response by inducing tumour pyroptosis and increasing T cell infiltration and decreasing the number of immunosuppressive cells. This effect was observed when applied in combination with ICB therapy in a postoperative HNSCC mouse model [563]. In another study, nanobody-albumin nanoparticles (NANAPs) were used for the delivery of 17864-Lx-a, a platinum-bound sunitinib analogue multikinase inhibitor, to EGFR overexpressing tumour cells. 17864-NANAPs showed a 40-fold higher binding to EGFR-positive squamous head and neck cancer cells leading to the successful release in the tumour cell and inhibition of proliferation; in contrast, the non-targeted formulations had no antiproliferative effects on HNSCC cells [564]. Sunitinib has been found to exert immunomodulatory properties by increasing the invasion of lymphocytes and DCs into the tumour milieu, while reducing the number and activity of T_regs_ and bone marrow-derived suppressor cells [565]. Nanoparticles can also be used to deliver immunostimulant agents for effective tumour immunotherapy, providing a smart new vehicle for the efficient presentation of tumour antigens and adjuvants on antigen-presenting cells [566,567,568]. An example is the use of the tandem peptide nanocomplex (TPNC) carrying CpG DNA ligand of TLR9s (iTPNC), which inhibits tumour growth in animal models of various cancers. The use of TPNC enhanced the efficiency of CTLA-4 antigen activation and mediated TLR involvement in T cells, and improved the responsiveness to anti-CTLA4 treatment with combinatorial effects after intratumoral administration. In addition, thanks to the activity of the nanocomplex formula, it was possible to dramatically reduce the treatment dose required to obtain a therapeutic effect, which was associated with a reduced risk of side effects associated with the systemic “off-target” reaction [569].

An interesting strategy in the treatment of HNSCC involves the use of IL-1α-loaded polyanhydride nanoparticles in combination with cetuximab. Cetuximab stimulated the secretion of pro-inflammatory cytokines from HNSCC cells in vitro via the IL-1α/IL-1R1/MyD88-dependent signalling pathway. Importantly, the IL-1 blockade had no effect on the anti-cancer efficacy of anti-EGFR mAb, but increased IL-1α expression and successfully induced a T cell-dependent anti-cancer immune response and reduced the immune suppressive cell action [570]. The cytotoxic CTLA-4 molecule can also be delivered by the cytotoxic T lymphocyte-associated molecule-4 (CTLA-4)-siRNA (NPsiCTLA-4). This siRNA delivery system has been found to enter both CD4^+^ and CD8^+^ T cell subsets at tumour sites in vitro and in vivo. This strategy led to a significant increase in the percentage of anti-tumour CTLs and decrease of the inhibitory T regulatory cells (T_regs_) among tumour infiltrating lymphocytes (TILs) [571]. Furthermore, tumour-associated immunosuppressive T_regs_ can be depleted using iron-oxide nanoparticles, combining anti-CTLA-4 immunotherapy with photodynamic therapy [572]. These molecules can be used to change the activation state of tumour-associated macrophages (TAMs) to a tumour-suppressor phenotype, thus serving as the basis of an anti-tumour strategy. For instance, iron-oxide nanoparticles contributed to the reprogramming of tumour-associated macrophages to a pro-inflammatory M1-like phenotype, suppressing tumour growth, and efficiently delivered the ovalbumin antigen (OVA) to dendritic cells, thus activating CD4^+^ and CD8^+^ effector T cells in animal models of solid tumours [573]. In another study of an iron-oxide core with cGAMP and E6/E7 peptides, HPV16 was conjugated to prevent HNSCC immune escape. The E6/E7-targeted nanosatellite vaccine has been shown to increase the number of tumour-specific CD8^+^ T cells more than 12-fold in the TME and reduce tumour burden. Furthermore, production of this nanosatellite vaccine with an anti-PD-L1 significantly increased the number of tumour-specific CTLs and reduced populations expressing depletion markers, resulting in more effective tumour control [574]. Another strategy using hafnium-oxide nanoparticles (NBTXR3) activated by radiotherapy (RT) resulted in increased radiation dose deposits within cancer cells compared to RT alone. Furthermore, NBTXR3+RT supported an immunogenic cell death-mediated effect in preclinical trials (e.g., NCT03589339), and immunocompetent cell infiltration in some advanced cancers treated with an anti-PD-1 therapy. NBTXR3 is currently being studied in seven clinical settings, including a phase I/II study in elderly frail patients with locally advanced HNSCC. To date, no early dose limiting toxicities (DLTs) have been observed [575].

Nanotechnology was also used for chemotherapy and photothermal treatment. Nanoparticles of gelatine, a natural biopolymer, also show antineoplastic potential. These molecules are degraded by matrix metalloproteinases (MMPs) called gelatinases. MMP-degradable gelatine nanoparticles (GNPs) were simultaneously loaded with photosensitizer indocyanine green (ICG) along with signal transducer activator of transcription 3 (STAT3) inhibitor NSC74859 (NSC, N) for efficient photothermal therapy (PTT) and immunotherapy of HNSCCs. The use of near-infrared radiation released ICG nanoparticles, which resulted in the effective photothermal destruction of tumours. In addition, inhibition of the STAT3 molecule induced strong anti-cancer immunity leading to enhanced anti-cancer therapy. Based on two HNSCC mouse models, it was demonstrated that the use of Gel-N-ICG nanoparticles with laser irradiation led to the inhibition of tumour microenvironment immunosuppression (TME) and reduction in the population of PD-1 cells, and increased the anti-cancer efficacy of the therapy [576]. Researchers also developed a nanosatellite vaccine targeting a tumour antigen to enhance the efficacy of the STING (interferon gene stimulator) agonist and sensitize SOX2-expressing HNSCCs to checkpoint blockade. Combining the nanosatellite vaccine with anti-PD-L1 led to higher CD8^+^CTL activity, but also reduced CTL exhaustion [574,577].

Nanoparticles are also used to create nanovaccines, which have the potential to deliver tumour antigens and adjuvants to lymphoid tissues and increase therapeutic efficacy by loading with immunosuppressive inhibitors or immunostimulant compounds in various solid tumours, including HNSCCs. This promising strategy may allow for maximal anti-cancer effect in cancer immunotherapy and transforming non-responders into responders [578,579].

Another vaccine based on a CpG-loaded aluminium phosphate nanoparticle delivery system demonstrated great antineoplastic potential in an animal model of HNSCC cancer. Following immunization, strong cellular immunity was observed in spleen and lymph node cells, including high activity of IFN-γ^+^CD4^+^ T cells, IFN-γ^+^CD8^+^ T cells, cytotoxic T lymphocytes (CTLs) and cytokine excretion, leading to significant tumour growth suppression and prolonged survival [580].

Another strategy may be the use of a liposomal HPV16 mRNA formulation (HPV16 E7 RNA-LPX vaccine). Intravenous administration of the vaccine to HR-HPV16-positive mice led to a strong E7 antigen-specific CD8^+^ T cell response and a persistent memory phenotype. The tumours of the immunized mice were characterized by a strong infiltration of activated immune cells and HPV16-specific T cells and a pro-inflammatory, cytotoxic and less immunosuppressive polarity. Moreover, the combination of PD-L1 with the HPV16 E7 RNA-LPX vaccine resulted in a synergistic inhibition of tumour growth and a significant improvement in survival [581].

Overexpression of mucin 1 (MUC1), a transmembrane glycoprotein, is also associated with the radio resistance and radiosensitivity observed in HNSCC cells. Therefore, attempts are being made using mannose-modified lipid/calcium/phosphate (LCP) nanoparticles to deliver MUC1-encoding mRNA to DCs in lymph nodes. Moreover, in vitro studies indicate that combined immunotherapy with a vaccine and an anti-CTLA-4 monoclonal antibody can significantly enhance the anti-tumour immune response compared to the vaccine or monoclonal antibody alone, and MUC1 can be a target for CAR-T therapy in HNSCC [582]. Recently, a nanovaccine consisting of a simple physical mixture of an antigen and a synthetic polymeric nanoparticle (PC7A NP) has also been developed. Following the application of this nanovaccine, inhibition of the tumour growth was observed in various tumour models, including human papillomavirus E6/E7 tumours. PC7A NP vaccine enhanced antigen delivery and cross-presentation in APCs and stimulated CD8^+^ T cell responses. Moreover, the combination of the PC7A nanovaccine and the anti-PD-1 antibody showed great synergy and was associated with the complete inhibition of tumour growth and improved survival in animal tumour models, suggesting anti-tumour memory generation [583].

In order to study the consequence of the use of the vaccine-induced immunologic answer for the progression of viral-associated HNSCC, single (gp100) and multiple (B16-tumor lysate containing gp100) immunogenic viral antigens were closed within lactic-co-glycolic acid (PLGA) nanoparticles. The use of this 80 kDa polymer was associated with the enhanced production of anti-tumour inflammatory/Th1 cytokines, as well as a higher expression of immune-stimulating cytokines, such as IFN-γ and reduced excretion of immunosuppressive cytokines, such as IL-10 in HNSCCs cell lines [579,584,585]. Vaccines using mesoporous silica rods (MSRs) also affected the anti-tumour response and immune editing in a preclinical HNSCC MOC2-E6E7^+^ model. Vaccination with the E7 peptide-loaded MSR vaccine resulted in strong infiltration of antigen-specific CD8^+^ T cells, tumour growth inhibition, and the moderately prolonged survival in HPV^+^ oral tumour mice [586].

### 3.3. Summary of Key Recommendations for HNSCC Immunotherapy According to the Society for Immunotherapy of Cancer (SITC), The National Comprehensive Cancer Network (NCCN) and the American Society of Clinical Oncology (ASCO)

The last years of the 20th century were a period of intensive pre- and clinical research into the use of new and modern immunotherapies in the treatment of patients with HNSCC. Therefore, The Society for Immunotherapy of Cancer (SITC) (available from: https://www.sitcancer.org/research/cancer-immunotherapy-guidelines, accessed on 31 January 2023) has established a committee of recognized experts to develop a consensus of recommendations for cancer immunotherapy, including HNSCC, published as the Cancer Immunotherapy Guideline—Head and Neck Cancer by SITC HNC Subcommittee. This communication contains therapeutic guidelines, i.e., inclusion/exclusion clinical criteria, immunological, immunohistochemical (IHC) and clinical parameters for patient selection, immunotherapy sequence, monitoring of the response to immunotherapy, management of adverse events and analysis of specific prognostic biomarkers that have been evaluated in HNSCC; these include the programmed death ligand 1 (PD-L1) expression, tumour mutational burden (TMB), and immune gene signatures within both the tumour and the surrounding tissue [42,587]. Due to differences in criteria, availability and regulations for immunotherapeutic agents in various countries, SITS primarily refers to new therapies approved by the U.S. Food and Drug Administration (FDA), but also comments on immunotherapeutic approvals by the European Commission. In the Cancer Immunotherapy Guideline—Head and Neck Cancer, the recommendations with the highest clinical value are those with high level consensus (i.e., with level 1 and level 2A evidence) with more than 50% of subcommittee votes [42,587]. Further detail into management strategies have been also referred to the National Comprehensive Cancer Network^®^ (NCCN^®^) Clinical Practice Guidelines in Oncology (NCCN Guidelines^®^). Head and Neck Cancers; NCCN Evidence BlocksTM (Version 1.2023, 01/30/23) [Available online: www.nccn.org/patents for current list of applicable patents. Head and Neck Cancers Version 1.2023—30 January 2023] and the American Society of Clinical Oncology^®^ (ASCO^®^) Head and Neck Cancer Guideline; Knowledge Conquers Cancer [Available online: https://www.cancer.net/cancer-types/head-and-neck-cancer/types-treatment#immunotherapy (accessed on 30 January 2023)] [43,587,588,589].

Key clinical immunotherapy recommendations for the treatment of HNSCC patients include [43,587,588,589]:Integration of immunotherapy with PD-1 inhibitors in the treatment of relapsed/metastatic head and neck squamous cell cancer (R/M HNSCC):
Frontline therapy: The FDA approved pembrolizumab (Keytruda, Merck), the anti-programmed cell death protein (PD-1) monoclonal antibody, for the first-line treatment of patients with unresectable R/M HNSCC;
Pembrolizumab has been approved for mono immunotherapy in HNSCC patients with naïve R/M HNSCC, whose tumours express a PD-L1 biomarker with a combined positive score, CPS ≥ 1 confirmed by IHC staining (or positivity for PD-L1 is ≥1% tumour proportion score, TPS), as defined by the FDA-approved test;Pembrolizumab has been approved for immunotherapy in combination with platinum and fluorouracil (FU) in HNSCC patients with R/M HNSCC, whose tumours express a PD-L1 biomarker with a combined positive score, CPS < 1 confirmed by IHC staining), as defined by the FDA-approved test;Pembrolizumab has been approved for immunotherapy in combination with platinum and fluorouracil (FU) in all naïve R/M HNSCC patients, regardless of PD-L1 biomarker specifications;The FDA also expanded the use of the PD-L1 IHC 22C3 pharmDx kit for selecting patients with HNSCC for pembrolizumab treatment as a single immunotherapeutic agent.Second-line therapy: The FDA approved two immunotherapeutic agents, the anti-programmed cell death protein (PD-1) monoclonal antibodies, nivolumab (Opdivo, Bristol-Myers Squibb) and pembrolizumab (Keytruda, Merck), for the treatment of patients with R/M HNSCC unresponsive to platinum-based treatment:
Pembrolizumab or nivolumab monotherapy may be proposed for the treatment of patients with R/M HNSCC without a clinical response to six months of platinum-based chemotherapy (if TPS ≥ 50);Option: best supportive care if not eligible for the treatment of patients with R/M HNSCC without a clinical response to six months of platinum-based chemotherapy and prior immunotherapy;Pembrolizumab or nivolumab monotherapy may be proposed for the treatment of patients with any PD-L1 CPS status and platinum-refractory R/M HNSCC.
HPV status and the use of immunotherapy in HNSCC: HPV status (based on the p16 biomarker IHC overexpression) should be taken into account in the selection of therapy, but it has no influence on the decisions for patients with R/M HNSCC treatment with the use of immunotherapy.
Evaluation of immune treatment response and further treatment recommendations for patients with advanced HNSCC:
Initial clinical follow-up after one month of treatment with assessment of immune-related symptoms and adverse events (AEs);Each subsequent assessment of immune-related symptoms and AEs should be evaluated at least monthly;For initial assessment, a baseline clinical exam of the patient with imaging via CT or PET-CT scan following, should be performed. In monitoring patients for signs of response, patient evaluation (via radiographic imaging) should occur every three months. If radiographic progression is observed early in treatment, and the patient is clinically stable, treatment continuation until progression is confirmed on a second scan is recommended;If clinical response after treatment and six months of maintenance immunotherapy is observed, continue treatment for at least two years or until disease progression or toxicity;If disease progression on or after treatment with a PD-1 inhibitor is observed, enrolment in a clinical trial, treat with palliative radiotherapy and/or chemotherapy is recommended;Combination therapy (notably chemotherapy + immunotherapy IO) for rapidly growing disease due to the need for an enhanced response rate is recommended;
In the first line setting for patients with R/M HNSCC with rapid/symptomatic progression, whose tumours express the PD-L1 biomarker, extreme chemotherapy regimen is recommended/Option: TPeX regimen or pembrolizumab + chemotherapy (FDA);In the first line setting for patients with R/M HNSCC with rapid/symptomatic progression, but without PD-L1 assessment, pembrolizumab + chemotherapy (FDA) or extreme chemotherapy regimen are recommended/Option: TPeX regimen;TMB (tumour mutational burden) testing if CPS is not available or in patients with rare tumours; TMB ≥ 10 interpreted as high; correlating with a clinical benefit to PD1 inhibitors


### 3.4. Latest Immunotherapies for HNSCC

Recent studies indicate the feasibility of various new ICIs targeting co-stimulatory and co-suppressive checkpoints expressed on immune cells as novel targets [590,591]. An example is the dose-escalation/-expansion study phase I/IIa clinical trial, which is evaluating the rationale for the use of the anti-LAG-3 antibody, BMS-986016 and nivolumab in metastatic solid tumours, including the HNSCC cohort (NCT01968109). Eftilagimod alfa (efti), a soluble LAG-3 protein that binds to a subgroup of MHC class II molecules and mediates APC activation and CD8 T cell activation, is expected to stimulate dendritic cells and increase lymphocyte recruitment, which may lead to stronger anti-cancer responses compared to the effects of pembrolizumab alone [592]. Preliminary results of the TACTI-002 phase II study on the use of efti plus pembrolizumab in the second-line treatment of patients with unselected HNSCC PD-L1 metastases are promising and indicate that the ORR and DCR are 39% and 50%, respectively. Efti in combination with pembrolizumab is safe and shows encouraging anti-tumour activity, also in first line NSCLC across all PD-L1 expression levels (NCT03625323) [590,593].

Targeting the EGFR pathway with cetuximab may also increase the effectiveness of immunotherapy by activating NK ADCC and inducing the expression of PD-L1 and other immunosuppressive factors. An example may be the results of a phase II study evaluating the use of pembrolizumab in combination with cetuximab in platinum-resistant or ineligible R/M HNSCC. Studies in the analysed cohort showed promising activity with a 45% ORR.

Another example of a drug molecule is lenvatinib, which is a multikinase inhibitor directed against vascular endothelial growth factor receptor (VEGFR) 1-3, fibroblast growth factor receptor (FGFR) 1-4, platelet-derived growth factor alpha receptor (PDGFR-α), rearranged while transfection (RET) and KIT. The combination of pembrolizumab and lenvatinib has been evaluated in LEAP-010, a phase III randomised, double-blind, placebo-controlled study in patients with R/M HNSCC (NCT04199104). Now, as the study continues, randomization will be stratified by PD-L1 tumour proportion score (<50% vs ≥50%), HPV OPC status, and the Eastern Cooperative Oncology Group (ECOG) performance status (PS). Combination therapy will continue for 2 years until disease progression or unacceptable toxicity [594].

It is also worth mentioning another new therapeutic target, i.e., inducible T cell co-stimulator (ICOS) or CD278 and its ligand (ICOSL). The INDUCE-1 study (NCT02723955), investigated the role of the ICOS agonist, GSK3359609, alone and in combination with pembrolizumab, or chemotherapy (5FU plus platinum) with or without pembrolizumab. In the anti-ICOS-only cohort, the ORR and DCR were 6% and 31%, respectively, while in the pembrolizumab cohort, the ORR and DCR were 24% and 65%, respectively. The combination of GSK609 with pembrolizumab shows promising anti-tumour activity and a manageable safety profile in pts with previously treated, PD-1/L1 naive HNSCC [595]. New data from additional safety cohorts included in the CT/pembrolizumab arms (GSK3359609 plus CT and GSK3359609 plus CT and cohort) confirmed that the safety profile of the triple combination is achievable, mainly in cases of adverse events related to CT and pembrolizumab [596].

Recently, a new trial has started: the INDUCE-3 trial is a randomized, double-blind study of GSK3359609 plus pembrolizumab vs placebo plus pembrolizumab for first-line treatment of PD-L1-positive R/M HNSCC. Furthermore, the role of other co-stimulator/co-inhibitor, such as TIM-3 and KIR combined with anti-PD-1, is under investigation in various advanced solid tumours (NCT02817633).

Epigenetic modification is another key regulator of immune cell function. An example is the clinical trial In a phase I/II, which analyses the use of vorinostat, a histone deacetylase inhibitor (HDACis), in combination with pembrolizumab in R/M HNSCC and salivary gland cancer (SGC), achieving in the HNSCC cohort an ORR of 36*%* [597].

## 4. Conclusions and Future Directions

The immunopathology of squamous cell carcinoma of the head and neck represents an increasingly promising and rapidly developing field of oncology, one that requires a precise understanding of the mechanisms of anti-cancer defence of the human cells, as well as the phenomenon of escape from immune surveillance of malignant cells. Knowledge of the activation/inhibition of the complex and various pro-and anti-cancer immunomodulatory pathways, and the interactions between tumour cells and immunocompetent cells, is an essential element in the development of modern cytotoxic or combined immunotherapy. Such discoveries could allow for further improvement of the clinical effectiveness of therapeutic proposals and greater use of effective and personalized immunotherapy. The possibility of a more detailed identification of the population of patients with HNSCC who could benefit most from the applied immunotherapy is also important. Currently, many extensive studies into the use of various forms of anti-cancer immunotherapy in human neoplasms remain as preclinical and early phase I/II studies and phase-III clinical trials. However, in HNSCC therapy, the most promising direction remains the application of monoclonal tumour- or immuno-targeted antibodies (mAb) directed against growth receptors and elements of the intracellular checkpoints of the immune system cells; however, many strategies have been proposed, such as vaccines and the transfer of modified T cells cultured in vitro or T cells modified for better expression of chimeric antigen receptors. Researchers are also intensively looking for therapeutics that inhibit the escape mechanisms of tumour cells from immune surveillance and adoptive cell therapy, which would be characterized by the maximum clinical effectiveness and the lowest possible risk of adverse effects of immunotherapy, in both HPV*^+ve^* and HPV*^−ve^* tumours, although the detailed mechanisms require further research.

Ongoing studies have highlighted the benefits of preventive vaccines. Various individual immunotherapy strategies are also currently being actively investigated in HNSCC, including therapeutic vaccination, T cell adoption therapy, and immune checkpoint blockade. The main goals of existing immunotherapy strategies include increasing the recognition of cancer cells by the immune system, releasing TAAs/TASs and enhancing their presentation; they also aim to increase recruitment and better efficiency of T cell effectors (increased activity of the innate immune response, i.e., activity of NK and CAF cells, overcoming immune suppression, i.e., inhibition of T_regs_ and MDSCs), to improve the infiltration of immune cells into tumour and TME via T cells activation and expansion in vivo; in addition, these strategies aim to improve TIL transfer and the modulation of immunosuppression, support the supply of activating/inhibitory cytokines and better regulate inhibitory and co-stimulatory receptors, i.e., by the use of immune checkpoint inhibitors (ICIs)TME. Promising future HNSCC immunotherapy strategies include the determination and identification of biomarkers that will predict the benefit of immunotherapy. In addition, it has been proposed to use combined strategies exploiting potential synergies between immunotherapies and radiotherapy or chemotherapy, to implement immunotherapy based on nanotechnology and to generally expand the settings and malignant tumours in which immunotherapy will give the best clinical effects.

This review summarizes and analyses recent papers describing the involvement of immunocompetent cells in the development and progression of HNSCC and provides a better understanding of the role of the immune system and the potential of the intracellular checkpoints and tumour antigens as immunotherapeutic targets.

Despite the use of state-of-the-art comprehensive treatment modalities, including surgery, radiotherapy and chemotherapy, the recurrence and overall survival rates in HNSCC patients have not improved satisfactorily over the past few decades. Therefore, it is necessary to introduce new treatment strategies, including immunotherapy, which has proven to be a promising alternative to standard therapies. Unfortunately, a significant proportion of patients do not respond to monocomponent immunotherapy. As such, it is vital to understand and identify the immunological mechanisms that determine the course of cancer.

For vast topics, such as cancer immunology, there is a constant need for regular collections of up-to-date knowledge. These can serve as a base for oncologists who, with a thorough understanding of signalling pathways, are better able to refer immunological parameters to clinical parameters in further studies. Such close cooperation between immunologists/molecular oncologists and oncology practitioners allow for a more comprehensive insight into understanding and identifying the causes of treatment failure, and will enable the development of strategies to overcome them in the future. In conclusion, a deeper understanding of immune evasion mechanisms and the role of immunocompetent cells is warranted to identify the most effective combination and combination therapies for HNSCC. As such, in this regard, personalized therapy for head and neck cancer may become increasingly likely in the future.

In conclusion, a more in-depth understanding of immune evasion mechanisms and the role of immunocompetent cells is warranted to identify the most effective combination and combination therapies for HNSCC. Moreover, in this aspect, the future use of personalized therapy for head and neck cancer is also not theoretical. Then, specific disorders of the immune system occurring in the course of a specific neoplastic disease will be able to be identified on the basis of biopsies or peripheral blood samples, constituting the basis for the selection of optimal combinations of targeted therapy, including immunotherapy. A greater understanding of this area may also contribute to the further development of new immunotherapeutic strategies and the identification of new potential immunological biomarkers necessary for greater clinical and prognostic benefit in HNSCC patients.

Undoubtedly, further multicentre studies are also needed to find more effective and low-toxic immunotherapeutic strategies to inhibit the neoplastic escape and immune tolerance mechanisms, and these would be a promising approach to promote new methods of treating HNSCC. The new studies will be necessary to test novel combinations and build up the biomarkers useful for patient selection, as well. Importantly, further detailed analysis of the immunological parameters of the tumour and its microenvironment, and a full understanding of the importance of the immune response to neoplastic antigens is also necessary.

A summary of recent and new discoveries in the field of HNSCC immunotherapy and their relation to previous knowledge is shown in Figure 5.

## Figures and Tables

**Figure 1 cancers-15-01642-f001:**
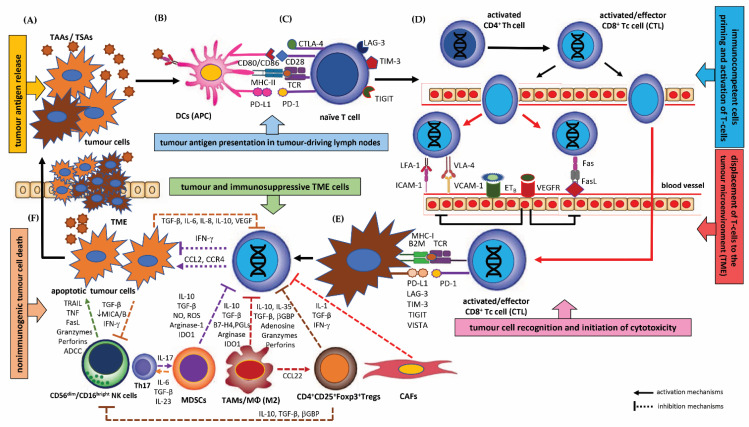
The cancer immunity cycle and the steps in cancer immunoediting. (**A**) Immunogenic cell death and tumour antigen release from the neoplastic cells, (**B**) antigen processing and presentation by antigen presenting cells (APCs), i.e., dendritic cells (DCs) to naïve immune cells in tumour-draining lymph nodes, (**C**) immunocompetent cells priming and activating T cells in the local lymph nodes, (**D**) displacement of T cells to the tumour microenvironment (TME) via the blood stream and infiltration to the tumour, (**E**) recognition of tumour cells by immune cells, (**F**) T cell mediated immune response and killing of cancer cells; abbreviations: TAAs/TSAs: tumour-associated antigens/tumour-specific antigens, CTLA: cytotoxic T cell antigen 4, also known as CD152 (cluster of differentiation 152), CD80/CD86: costimulatory molecules CD80 (B7-1)/CD86 (B7-2), CD28: cluster of differentiation 28, the receptor for CD80 (B7-1) and CD86 (B7-2) proteins, TCR: T cell receptor, PD-L1/PD-1: programmed death-ligand 1 (PD-L1), also known as cluster of differentiation 274 (CD274) or B7 homolog 1 (B7-H1)/programmed cell death protein 1, also known as PD-1 and CD279, Fas/Fas-L: Fas receptor, also known as FasR, apoptosis antigen 1 (APO-1 or APT), cluster of differentiation 95 (CD95) or tumour necrosis factor receptor superfamily member 6 (TNFRSF6)/Fas ligand, also known as CD95L or CD178), VEGFR: vascular endothelial growth factor receptor, ET_B_: endothelin B receptor, VLA-4: adhesion molecules, also known as integrin α4β1 (very late antigen-4), VCAM-1: vascular cell adhesion protein 1 also known as vascular cell adhesion molecule 1 (VCAM-1) or cluster of differentiation 106 (CD106), LFA-1: lymphocyte function-associated antigen 1, ICAM-1: intercellular adhesion molecule 1, also known as CD54 (cluster of differentiation 54), B2M: β2 microglobulin, a component of MHC class I molecules, LAG-3: lymphocyte-activation gene 3, TIM-3: T cell immunoglobulin and mucin-domain containing-3, TIGIT: T cell immunoreceptor with Ig and ITIM domains, VISTA: immune checkpoint molecule for human T cells, TGF-β: transforming growth factor-β, MICA/B: MHC class I polypeptide–related sequence A/B, βGBP: interferon-induced guanylate-binding protein 1, TRAIL: TNF-related apoptosis-inducing ligand, CAFs: cancer-associated fibroblasts, CD4^+^CD25^+^Foxp3^+^T_reg_: regulatory T cells, known as suppressor T cells, TAMs/MФ: tumour-associated macrophages, MDSCs: myeloid-derived suppressor cells, CD56^dim^/CD16^bright^ NK cells: activated natural killer cells.

**Figure 2 cancers-15-01642-f002:**
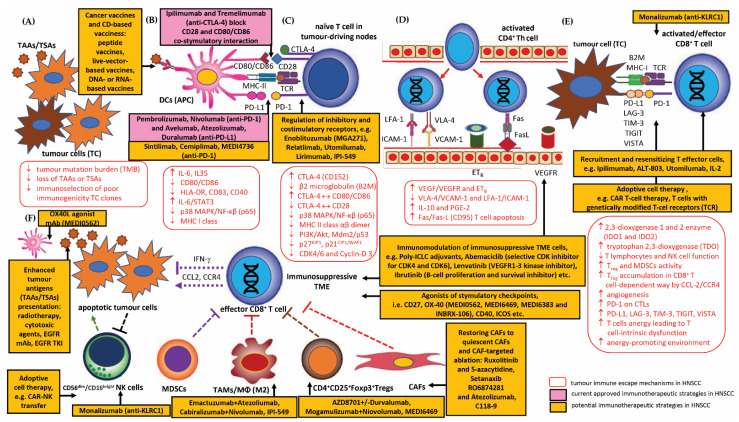
The crucial mechanisms of tumour immune escape and summary of current approved/potential strategies to overcome immunosuppressive TME in HNSCC. (**A**) Deficiency in the tumour antigen release from the tumour cells (TC), (**B**) disfunction of dendritic cells (DCs) and APC presentation of tumour antigens to naïve immune cells in tumour-draining lymph nodes, (**C**) defect in immunocompetent cell priming and activation of effector T cells, (**D**) disturbed trafficking of T cells to the tumour microenvironment (TME) and enhanced formation of new blood vessels (angiogenesis), (**E**) defect in the recognition of tumour cells by immunocompetent cells, (**F**) inhibition of T cell mediated immune response and killing of cancer cells; Abbreviations: TAAs/TSAs: tumour-associated antigens/tumour-specific antigens, TMB: tumour mutation burden, CTLA: cytotoxic T cell antigen 4 also known as CD152 (cluster of differentiation 152), CD80/CD86: costimulatory molecules CD80 (B7-1)/CD86 (B7-2), CD28: cluster of differentiation 28, the receptor for CD80 (B7-1) and CD86 (B7-2) proteins, TCR: T cell receptor, PD-L1/PD-1: programmed death-ligand 1 (PD-L1), also known as cluster of differentiation 274 (CD274) or B7 homolog 1 (B7-H1)/programmed cell death protein 1, also known as PD-1 and CD279, Fas/Fas-L: Fas receptor, also known as FasR, apoptosis antigen 1 (APO-1 or APT), cluster of differentiation 95 (CD95) or tumour necrosis factor receptor superfamily member 6 (TNFRSF6)/Fas ligand, also known as CD95L or CD178), VEGFR: vascular endothelial growth factor receptor, ET_B_: endothelin B receptor, VLA-4: adhesion molecules, also known as integrin α4β1 (very late antigen-4), VCAM-1: vascular cell adhesion protein 1, also known as vascular cell adhesion molecule 1 (VCAM-1) or cluster of differentiation 106 (CD106), LFA-1: lymphocyte function-associated antigen 1, ICAM-1: intercellular adhesion molecule 1, also known as CD54 (cluster of differentiation 54), IDO1, IDO2: 2,3-dioxygenase 1 and 2 enzyme, TDO: tryptophan 2,3-dioxygenase, B2M: β2 microglobulin, a component of MHC class I molecules, LAG-3: lymphocyte-activation gene 3, TIM-3: T cell immunoglobulin and mucin-domain containing-3, TIGIT: T cell immunoreceptor with Ig and ITIM domains, VISTA: immune checkpoint molecule for human T cells, TGF-β: transforming growth factor-β, MICA/B: MHC class I polypeptide–related sequence A/B, βGBP: interferon-induced guanylate-binding protein 1, TRAIL: TNF-related apoptosis-inducing ligand, CAFs: cancer-associated fibroblasts, CD4^+^CD25^+^Foxp3^+^T_reg_: regulatory T cells, known as suppressor T cells, TAMs/MФ: tumour-associated macrophages, MDSCs: myeloid-derived suppressor cells, CD56^dim^/CD16^bright^ NK cells: activated natural killer cells.

**Figure 3 cancers-15-01642-f003:**
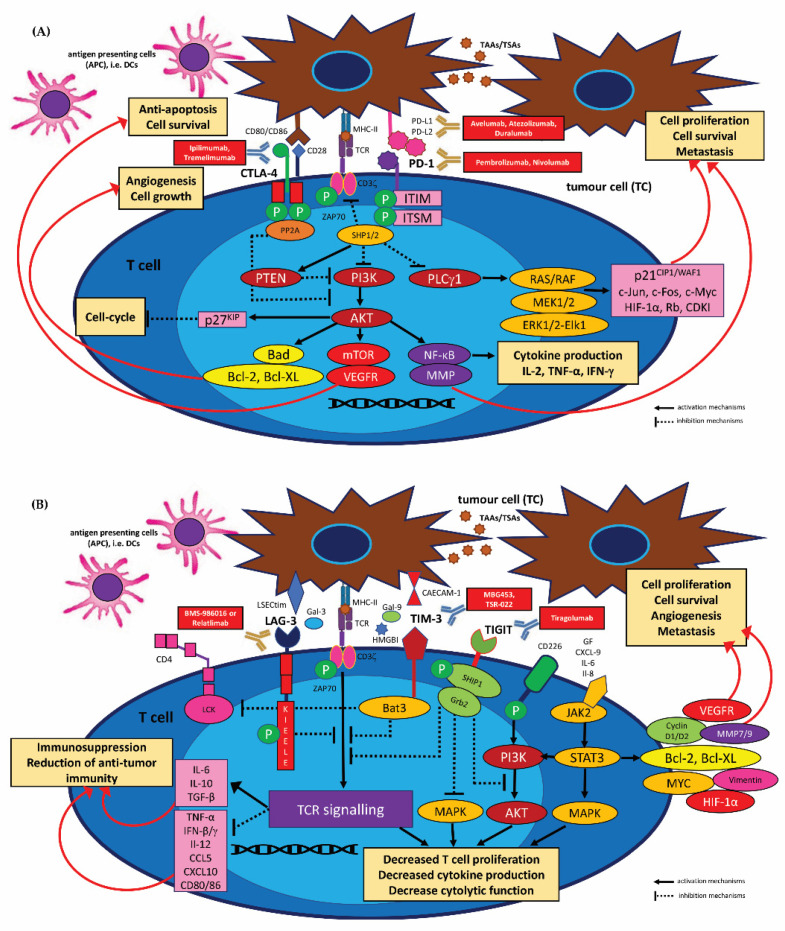
Intracellular immune checkpoint receptor signalling and functional effects of their activity in HNSCC. (**A**) PD-1 and CTLA-4 target different molecules to inhibit T cell activity by inhibiting PI3K/AKT signalling. PD-1 ligation by PD-L1 or PD-L2 on tumour cells (TCs) leads to phosphorylation of ITSM/ITIM motifs in T cells, which recruits the tyrosine phosphatases SHP1 and SHP2 to inhibit TCRζ-induced activation of the PI3K/AKT and to activate *PTEN*, a protein coding gene. This lowers Bad, Bcl-2 and Bcl-xL expression, downregulates mTOR/VEGFR signalling and suppresses transcription, resulting in the accumulation of p27^KIP1^ and inhibition of cyclin-dependent kinases (CDKI); this blocks cell cycling and proliferation. PD-1 ligation also inhibits PLCγ1 and the downstream Ras/Raf/MEK1/2/ERK1/2 pathway leading to the upregulation of the pro-apoptotic mechanisms. CTLA-4 engagement by its ligands CD80/CD86 on TCs activates the serine/threonine phosphatase PP2A, which directly inhibits the TCRζ/CD28-mediated AKT phosphorylation, but preserves PI3K activity. In contrast to PD-1, CTLA-4 stimulation does not inhibit Ras/Raf/MEK/ERK or PLCγ1 signalling. Ligation of CTLA-4 phosphorylates the pro-apoptotic factor Bad and enhances Bcl-XL activity to prevent T cell apoptosis, (**B**) LAG-3 is believed to signal through its unique KIEELE motive to transduce antiproliferative signals from the TCR. Upon Caecam-1 and Gal-9/TIM-3 triggering, Bat3 is released form the cytoplasmatic tail of TIM-3 and allows for binding of SH2 domain containing Src kinases, such as (LCK) and ZAP-70 which subsequently block TCR signalling. Upon ligand interaction, TIGIT becomes phosphorylated, and recruitment of SHIP1 and Grb2 blocks the PI3K and MAPK pathways, resulting in reduced T cell activation, proliferation, and effector functions; JAK2/STAT3 activation increases the production of immunosuppressive molecules, including IL-6, IL-10, VEGFR, HIF-1-α, cyclin D1/D2, Bcl-2, Bcl-XL and vimentin; as STAT3 levels rise in the TME, immunological molecules and inhibitory cytokines are released. Abbreviations: TAAs/TSAs: tumour-associated antigens/tumour-specific antigens, CTLA: cytotoxic T cell antigen 4, CD80/CD86: costimulatory molecules, TCR: T cell receptor, ZAP-70: zeta(ζ)-chain-associated protein kinase-70, PD-L1/PD-1: programmed death-ligand 1/programmed cell death protein 1, ITIM: immunoreceptor tyrosine-based inhibitory motif, ITSM: immunoreceptor tyrosine-based switch motif, SHP1/SHP2: protein-tyrosine phosphatases, PI3K: phosphoinositide 3-kinase, AKT: protein kinase B also known as PKB, PLCγ1: phospholipase C, gamma 1, mTOR: mammalian target of rapamycin, VEGFR: receptors for vascular endothelial growth factor, MMP: matrix metalloproteinase enzymes, Raf: serine/threonine-specific protein kinase, Ras: protein superfamily of small GTPases, MEK: mitogen-activated protein kinase, ERK: extracellular signal-regulated kinase, Bad: Bcl-2 associated agonist of cell death, Bcl-XL: B-cell lymphoma extra-large; c-Jus, c-Fos, c-Myc: transcriptions factors, Rb: retinoblastoma protein, PI3K: phosphatase 3-kinase, MAPK: mitogen-activated protein kinase, HIF-1α: hypoxia-inducible factor 1-alpha, CDKI: cyclin-dependent kinase inhibitor protein, LAG-3: lymphocyte activation gene 3, TIGIT: T cell immunoglobulin and ITIM domain, GITR: glucocorticoid-induced TNFR-related protein, TIM-3: T cell immunoglobulin and mucin-domain containing-3, LCK: lymphocyte-specific protein tyrosine kinase, ZAP-70: zeta-chain-associated protein kinase-70, SHIP1: SH2 domain containing inositol-5-phosphatase, Grb2: growth factor receptor bound protein 2, MICA/B: MHC class I chain-related proteins A and B, NKG2D: natural killer group 2D.

**Figure 4 cancers-15-01642-f004:**
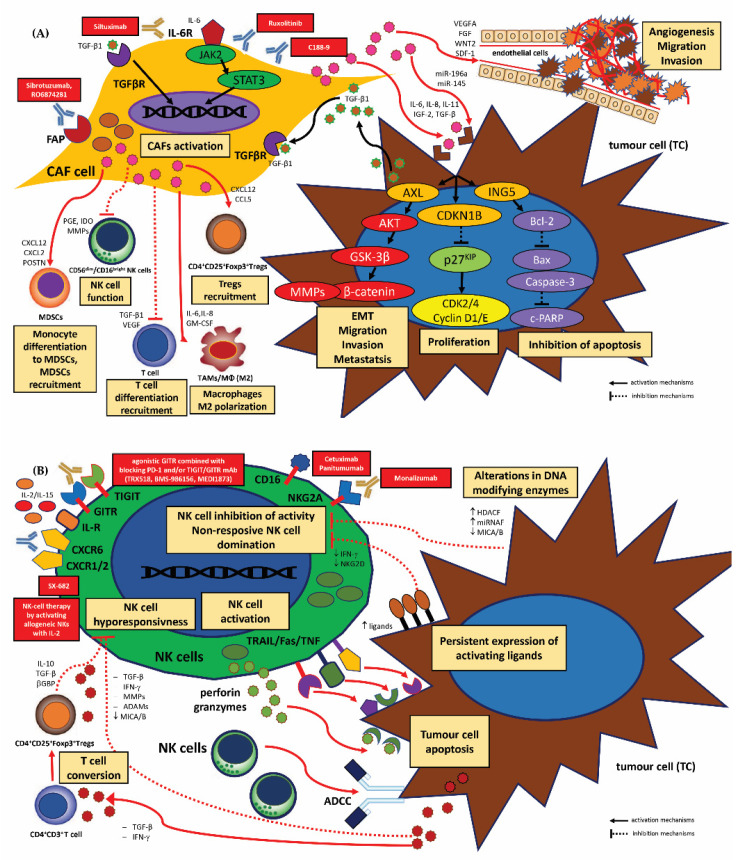
The functional effects of CAFs and NK cell activity and therapeutic strategies for targeting CAFs in HNSCC. (**A**) CAF-derived cytokines (TGF-β1, IL-6, IL-8, IL-11, IGF-2) and microRNAs (miR-196a, miR-145) are transported via exosomes to HNSCC cancer cells. This results in the activation of AXL, CDKN1B and Bcl-2 and their downstream signalling. These pathways lead to increased β-catenin and MMP expression, as well as upregulation of cyclin D1/E and downregulation of c-PARP molecule. The effects are enhanced proliferation, migration, invasion and tumour cell metastasis, and inhibition of apoptosis. Secretion of cytokines, growth factors and other molecules (IL-6, IL-8, GM-CSF, TGF-β1, VEGF, PGE, IDO, MMPs) by CAFs target various immune cells, inhibiting T cell and NK cell activity and enhancing CD4^+^CD25^+^Foxp3^+^T_regs_, MDSC and M2 TAM recruitment and function, (**B**) NK cell recognition is mediated by a balance of activating and inhibitory signals. NK cells kill target tumour cells by the release of cytotoxic granules containing perforin and granzymes, by the expression or release of TRAIL, TNF and FasL and by antibody-dependent cellular cytotoxicity (ADCC), resulting in apoptosis of the target cell. Tumour cells may escape NK cell dependent immunosurveillance by alterations in DNA modifying enzymes, such as those involved in epigenetic gene regulation and repression of MICA/B. Expression of activating ligands and sustained triggering of NKG2D leads to hypo-responsiveness and decreased cytotoxicity. Cytokines released by the tumour, such as TGF-β and IFN-γ, repress MICA/B, downmodulate NKG2D expression and IFN production in NK cells and promote the conversion of CD4^+^ T cells into regulatory CD4^+^CD25^+^Foxp3^+^T_regs_ cells, which suppress immunosurveillance. Upregulation of MMPs and ADAMS promotes the shedding of activating ligands, such as MICA, which binds to NKG2D on CD4^+^ T cells and NK cells, resulting in hypo responsiveness in NK cells; Abbreviations: TGFβR: transforming growth factor, beta receptor, FAP: fibroblast activation protein, JAK2: Janus kinase 2, STAT3: signal transducer and activator of transcription 3, AXL: tyrosine-protein kinase receptor, AKT: protein kinase B also known as PKB, GSK3β: glycogen synthase kinase 3, MMP: matrix metalloproteinase enzymes, CDK2: cyclin-dependent kinase 2 protein, Bax: apoptosis regulator, Bcl-XL: B-cell lymphoma extra-large, c-PARP: poly (ADP-ribose) polymerase, TIGIT: T cell immunoglobulin and ITIM domain, GITR: glucocorticoid-induced TNFR-related protein, NKG2D: natural killer group 2D, TRAIL: TNF-related apoptosis-inducing ligand, FAS: apoptosis antigen 1.

**Figure 5 cancers-15-01642-f005:**
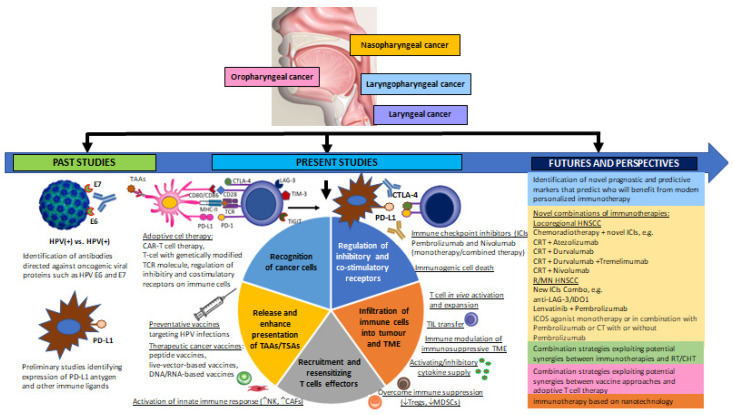
Summary of recent and new discoveries in the field of HNSCC immunotherapy and their relation to previous knowledge. Previous studies have indicated the role of oncogenic viruses HPV 16, 18, 31, 33, 35, etc., in the development of mainly the oropharynx and oral cavity, Additionally, immune markers on HNSCC tumour cells and circulating antibodies targeting viral antigens were identified. It has also been indicated that immunological markers may have a significant prognostic and predictive role in the response to traditional HNSCC therapies. Current ongoing studies have identified the benefits of preventive vaccines. Various individual immunotherapy strategies in HNSCC are also currently being actively investigated, including therapeutic vaccination, T cell adoption therapy, and immune checkpoint blockade. The main goal of currently used immunotherapy strategies is, among others: to increase the recognition of cancer cells by the immune system; release and enhance the presentation of TAAs/TASs; recruitment and desensitising T cell effectors (increased activity of innate immune response, i.e., activity of NK and CAF cells; overcoming immune suppression, i.e., inhibition of T_regs_ and MDSCs); infiltration of immune cells into the tumour and TME via T cell activation and expansion in vivo; TIL transfer; modulation of the immunosuppressive, activating/inhibitory cytokine supply; regulation of the inhibitory and co-stimulatory receptors, i.e., use of immune checkpoint inhibitors (ICIs)TME. Promising future HNSCC immunotherapy strategies include the determination and identification of biomarkers that will predict the benefits for immunotherapy; the use of combined strategies exploiting potential synergies between immunotherapies and radiotherapy or chemotherapy; immunotherapy based on nanotechnology; expanding settings and malignant tumours in which immunotherapy will give the best clinical effects; abbreviations: HPV: human papilloma virus, TAAs/TSAs: tumour-associated antigens/tumour-specific antigens, PD-L1: programmed cell death protein ligand, CTLA-4: cytotoxic T cell antigen 4, ICIs: immune checkpoint inhibitors, CRT: chemoradiotherapy, CHT: chemotherapy, RT: radiotherapy, LAG-3: lymphocyte activation gene-3, IDO1: indoleamine 2,3-dioxygenase 1, T_regs_: regulatory T cells, MDSCs: myeloid-derived suppressor cells, CAFs: cancer-associated fibroblast, NK: natural killer cells.

**Table 1 cancers-15-01642-t001:** Immune cell populations and their anti-tumour or pro-tumour functions within the TME.

Immune Cells	Markers of Immune Cells	Mechanisms	
		Cytokine and Other Factors	Functions
Tumour-Antagonizing Immune Cell Activity			
APCs (DCs)	CD11c^+^/CD11b^+^/CD8^−^/CD11a/CD15s/CD18/CD29/CD44/CD49d/CD50/CD54	IRF7, NF-κB activation, IFN-α, IL-6, IL-8, TNF-α	- Presenting tumour-associated epitopes (TAAs and TSAs) and providing costimulatory signals to the naïve antigen-specific T cells for T cell activation; - Activation of NK cells and B cells; - Upregulation of co-stimulatory molecules.
Effector T cells	CD3^+^/CD4^+^ T cells Th1 phenotype lymphocytes	IFN-γ	- Playing an important role in modulating immune responses to tumour cells; - Recognition of tumour antigens in association with MHC class II molecules (MHC II-restricted antigen recognition); - Indirect activation of tumour-suppressing CTLs by activating the antigen-presenting cells; - Secretion of IFN-γ and TNF-α to activate effector CD3^+^CD8^+^ cells (CTLs) and induce cytotoxicity in tumour cells; - IFN-γ produced by Th1 cells activates macrophages (TAMs), increasing phagocytosis of tumour cells; - Contribution to the killing of tumour cells via the TNF-related apoptosis-inducing ligand (TRAIL) pathway.
	CD3^+^/CD4^+^ T cells Th_2_ phenotype lymphocytes	IL-4, IL-5, IL-13	- Recruitment of eosinophils to the tumour environment via IL-4 and IL-13; anti-tumour eosinophil activity includes attraction of tumour-specific CTLs, activation of macrophages (TAMs), and vascularization of the stroma; however, IL-5 may enhance tumour proliferation.
	CD3^+^/CD4^+^ T cells Th_17_ phenotype lymphocytes	IL-17A, IL-17F	- Coordination of chronic inflammatory responses, which tend to promote tumour growth and cell survival; - Expression of high levels of IL-6 and TGF-β may induce Th17 cell polarization, creating a tumour-promoting niche.
	CD3^+^/CD8^+^ T cytotoxic cells CTLs	perforin/granzyme B and granulysin release, IFN-γ, TNF-α	- Recognition of tumour antigens in association with MHC class I molecules (MHC I-restricted tumour antigen recognition); - Killing target tumour cells by granule exocytosis and Fas/FasL-mediated induction of apoptosis.
NK cells	CD3^−^/CD16^+^/CD56^+^	IFN-γ, TNF-α, GM-SCF, IL-5, IL-6, IL-8, IL-10, IL-13, CCL2, CCL3, CCL4, CCL5, CXCL10	- Secreting pro-inflammatory cytokines and chemokines to promote the anti-tumour activity; - Cytotoxicity without prior tumour antigen (TAAs/TSAs) presentation; - Mediating the tumour killing response mainly through releasing perforin and granzymes to induce the apoptosis of the target tumour cells; - Modulation of adoptive immune anti-tumour response.
NKT cells	CD3^+^/CD16^+^/CD56^+^/CD161^+^/CD1a^+^	IFN-γ, TNF-α, GM-SCF, TGF-β, IL-2, IL-4, IL-5, IL-6, IL-10, IL-13, IL-17A	- Induction of cytotoxicity in tumour cells; - Maintenance of antigen-specific immunological memory.
M1 TAMs/MФ	CD68^+^	IL-12, IL-23, TNF-α, CCL2, CCL5, CXCL5, CXCL9, CXCL10, CXCL16 PD-L1/PD-L2 expression MMP-9 release	- Contribution to the CD3^+^ CD4^+^ Th1 immune response, inhibition of tumour cell proliferation, cytotoxic activity; - Inhibition of T lymphocyte function; - Remodelling of ECM.
N1 TANs	CD66b^+^/CD11b^+^/CD14^−^/ HLA-DR^+^/CD177^+^/CD15^hig^	ROS secretion ICAM-1, TNF-α cytokine secretion Fas signalling MMP-8 release	- TRPM2 activation leads to tumour growth inhibition; - Leukocyte recruitment, enhanced proliferation of T cells leads to immune anti-tumour response; - Activation of caspase cascade and apoptosis of cancer cells; - Remodelling of ECM, downregulation of β1-integrin activity, cleavage of cytokines, decreased activation TGF-β → ↓ miR-21 expression → ↓ PDCD4.
Tumour-Promoting Immune Cell Activity			
M2 TAMs/MФ	CD68^+^	IL-1ra, IL-4, IL-13 IL-10, TGF-β, PPARγ, VEGF, arginase-1 (Arg-1), IDO	- Promotion of tumour progression, suppression of CD4^+^ and CD8+ Th lymphocytes anti-tumour immune response, encourage CD8^+^ tolerance, angiogenesis.
N2 TANs	CD15^+^/HLA-DR^+^/CD11b^+^/CD14^−^/CD33^+^/Lox-1^+^	ROS secretion chemokine/cytokine (CXCR4, MMP-9, VEGF, Lox1, FATP2) neutrophil elastase (NE) production NET formation cathepsin G, MMP-8/9 and VEGF secretion Arginase-1 (Arg-1) secretion	- DNA mutations lead to tumour promotion and progression; - Enhanced CCL17 expression secretion → T_regs_ recruitment to the TME, cytotoxic activity; - Activation of EGFR and TLR4 → ERK-dependent gene transcription, PI3K-AKT activation; - Inactivation of thrombospondin-1 lead to tumour proliferation; - TME remodelling leads to cancer invasion and metastasis; - Remodelling of ECM → TGF-β activation, enhanced vascular permeability and endothelial cell growth lead to angiogenesis; - Downregulation of cytotoxic CD8^+^T cell effects leads to immunosuppression.
MDSCs	CD11b^+^/CD33^+^/CD14^+^/ CD15^+^/CD16^+^/HLA-DR^−^/ CD3^−^/CD19^−^/CD56^−^	NO, ROS, HIF-1α, iNOS, Arginase-1 (Arg-1) secretion IL-10, TGF-β1 secretion MIF, COX2, PGF_2_ (PGLs), SCF, M-CSF, IDO1, IL-6, IL-1β, IL-4, IL-13, TNF-α, GM-CSF, VEGF, PD-L1 expression MMP-9 release	- Suppression of immune responses; - Inhibition of T cell activation and proliferation; - Differentiation into TAMs and mature DCs and M2 macrophages; - Promotion of angiogenesis; - Remodelling and degradation of ECM.
T_regs_	CD4^+^/CD25^+^/Foxp3^+^	IL-10, TGF-β1, IL-35 secretion perforin/granzyme A/B release VEGF expression	- Suppression or downregulation of induction and proliferation of effector T cells such as CD3^+^CD4^+^ and CD3^+^CD8^+^ T cells, B cells, CD56^dim^/CD16^bright^ NK cells, macrophages (M1/M2 phenotype TAMs) and DCs; - Promotion of angiogenesis.
CAFs	αSMA^+^/FAP^+^/FSP-1^+^/CD33^−^CAV-1/FSP-1/PDGFR-α/ PDGFR-β/Thy-1 absent cytokeratin (CK)	EGF, HGF, VEGF, MCP1, Ly6c, COX2, PGE2, BDNF, MFAP5, CXCL1, CXCL12, CXCL14, CCL2, CCL5, CCL7 IL-6, IL-17A, TGF-β1 and 2 secretion MMP2/3, vimentin release	- Stimulation of tumour growth and spread; - Promotion of angiogenesis; - Promotion of invasion and metastasis, bone resorption; - Chemo- and radio-resistance; - Remodelling and degradation of ECM.

APCs (DCs): antigen-presenting cells (dendritic cells); NK: natural killer cells; NKTs: NK-like cells; TAMs/MФ; tumour-associated macrophages; TANs: tumour-associated neutrophils; MDSCs: myeloid-derived suppressor cells; Treg: regulatory T cells; CAFs: carcinoma-associated fibroblasts.

**Table 2 cancers-15-01642-t002:** Key differences in the immune system modulation between HPV(−)^neg^ and HPV(+)^pos^ HNSCC.

Immune Settings	Immune Parameters	HPV(-)^neg^ HNSCC	HPV(+)^pos^ HNSCC
Immune cells	CD3^+^ zeta chain T cells (CD3ζ)	Lower (↓)	Higher (↑)
	CD4^+^ T cells/ T cell proliferation and Th1,Th17 differentiation	Lower (↓)/Inhibition	Higher (↑)/Stimulation
	CD8^+^ IFN-γ T cells (CTLs)	Lower (↓)	Higher (↑)
	CD4^+^/CD8^+^ ratio	Increased (↑)	Decreased (↓)
	CD45RO^+^ cells	Lower (↓)	Higher (↑)
	CD3^+^ IL-17 T cells	Lower (↓)	Higher (↑)
	CD19^+^/CD20^+^ B cells	Lower (↓)	Higher (↑)
	CD4^+^CD25^+^Foxp3^+^T_regs_	Higher (↑)	Lower (↓)
	CD56^dim^ NK cells	Lower (↓)	Higher (↑)
	TI APCs/DCs	Lower (↓)	Higher (↑)
	TAMs CD68+ M1/M2 phenotype	M2 domination	M1 domination
	Myeloid dendritic cells (MDCs)	Lower (↓)	Higher (↑)
Cytokines/ Chemokines	Secretion of chemokines via an IL-1/IL-1R-mediated axis	Higher (↑)	Lower (↓)
	Neutrophil-specific chemokines (CXCL1, −5, −6, −8)	Higher (↑)	Lower (↓)
	Monocyte-specific chemokines (CCL7, −8)	Higher (↑)	Lower (↓)
	T-lymphocyte chemokines (CCL3, −4, −5)	Higher (↑)	Lower (↓)
	Expression of co-stimulatory CD80 and CD83 molecules on immature DCs	Lower (↓)	Higher (↑)
	Inhibitory cytokine IL-10, TGF-β, IL-6 production	Higher (↑)	Lower (↓)
	Cytokine IL-2 and/or IFN-γ production	Absence	Presence
	Response to mitogens or IL-2	Decreased (↓)	Increased (↑)
	Level of cytotoxic mediators (granzyme A, granzyme B, perforins)	Lower (↓)	Higher (↑)
Antigens/Transcription factors	Aberrant activation of the STAT3 and NF-ĸB related to TGF-β, IL-6 signalling	Increased (↑)	Decreased (↓)
	HLA-DR expression	Decreased (↓)	Increased (↑)
Immune inhibitory checkpoint ligand and receptors	T cell exhaustion markers: PD-1, CTLA-4, TIM-3, LAG-3, IDO1, KIR, TIGIT	Increased (↑)	Decreased (↓)

HPV: human papilloma virus; HPV(−)/HPV(+) HNSCC: HPV-negative/HPV-positive HNSCC; CD45: an essential regulator of T- and B-cell antigen receptor signalling, leukocyte common antigen, LCA; CTLs: CD8^+^ cytotoxic cells; CD4^+^CD25^+^Foxp3^+^T_regs:_ regulatory T cells; CD56^dim^ NK: activated natural killer cells; TI APCs/DCs: tumour infiltrating antigen presenting cells/dendritic cells; TAMs: tumour associated macrophages; MDSCs: myeloid-derived suppressor cells; STAT3: signal transducer and activator of transcription 3; NF-ĸB: nuclear factor NF-kappa-B; PD-1: programmed death ligand; CTLA-4: cytotoxic T cell antigen 4; TIM-3: T cell immunoglobulin and mucin-domain containing-3; LAG-3: lymphocyte-activation gene 3; IDO1: indoleamine 2,3-dioxygenase 1 enzyme; KIR: killer immunoglobulin-like receptor; TIGIT: T cell immunoreceptor with Ig and ITIM domains.

## Data Availability

Data available in a publicly accessible repository.

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
