# Peer review of "The Role of Different Immunocompetent Cell Populations in the Pathogenesis of Head and Neck Cancer—Regulatory Mechanisms of Pro- and Anti-Cancer Activity and Their Impact on Immunotherapy"

_cancers, 2023, doi:10.3390/cancers15061642_

Round 1

Reviewer 1 Report

This study aims to better understand the role of different immunocompetent cell populations in the pathogenesis of head and neck cancer. According to the latest GLOBCANdata, head and neck squamous cell carcinoma (HINSCC) represents the sixth most prevalent human malignancy. This review was intended to provide a comprehensive overview of the role of immune response in the genesis of HINSCC tumours, molecular signatures, and the mechanisms that regulate pro- and anticancer activity. The study corpus encompasses a wide range of recent molecular, observational, and interventional studies on the role of immune signaling pathways and the interaction between neoplastic cells and immune cells in human HNSCC. Much evidence suggests that the tumour microenvironment, tumour infiltrating lymphocytes and various circulating subpopulations of immunocompetent cells, such as regulatory T cells, cytotoxic CD3+CD8+ T cells (CTL) and CD3+CD4+ T helper type 1/2/9/17 lymphocytes, T follicular helper cells and CD56dim/CD16bright activated natural killer cells, carcinoma-associated fibroblasts, myeloid-derived suppressor cells, tumour-associated neutrophils, as well as tumour-associated macrophages can affect the initiation, progression, and spread of HNSCC and determine the response to immunotherapy. This review presents the latest reports and molecular studies on the antitumour role of selected subpopulations of immunocompetent cells in the pathogenesis of HNSCC, also including HPV+ and HPV- tumours. The present review also provides an overview of the early preclinical and clinical trials and phase II clinical trials published in this area, which highlight the unprecedented effectiveness and limitations of immunotherapy in HNSCC, and the emerging issues facing the field of HNSCC immune oncology. The content of this manuscript is suitable for publication in Cancers.

Author Response

Response to Reviewer 1 Comments

Point: The content of this manuscript is suitable for publication in Cancers.

Response: I would like to thank you for your considered and substantive review of my work and for confirming that the manuscript is suitable for publication in Cancers.

Reviewer 2 Report

The paper reports the role of the immune response in HNSCC tumor-genesis, molecular signatures and the mechanisms regulating pro- and anti-cancer activity. This is an interesting attempt for cancer biology and immuno-oncology. However, data are limited and insufficient evidence made in the manuscript. I have several comments and suggestions which I believe need to be addressed before the paper should be accepted. Some comments and suggestions are given below:

1.     The author should provide more information about the key clinical immunotherapy recommendations for the treatment of HNSCC.

2.     The authors need to add a mechanism summary figure to highlight the new findings and their connection with previous knowledge.

3.     The quality of the figures 2, 3 needs to be improved.

4.     The conclusion should be rewritten.

Author Response

Response to Reviewer 2 Comments

I would like to thank you for your considered, substantive and helpful review of my work. All your comments and suggestions have been taken into consideration when improving the work, and that they have made an invaluable contribution to the redrafting and editing of the revised text. A point-by-point answer to the Reviewer’s comments is given below.

Point 1: The author should provide more information about the key clinical immunotherapy recommendations for the treatment of HNSCC.

Response 1: I entirely agree. In accordance with the Reviewer’s suggestion, the manuscript has been supplemented with the key clinical recommendations for the treatment of HNSCC based on the latest NCCN Guidelines (Version 1.2023, 01/30/23) and the latest therapeutic data. In the revised and supplemented text on key therapeutic recommendations, the following publications were referred to:

  1. The National Comprehensive Cancer Network® (NCCN®) Clinical Practice Guidelines in Oncology (NCCN Guidelines®). Head and Neck Cancers; NCCN Evidence BlocksTM (Version 1.2023, 01/30/23) [Available from: www.nccn.org/patents for current list of applicable patents. Head and Neck Cancers Version 1.2023 - January 30, 2023].
  2. The Society for Immunotherapy of Cancer consensus statement on immunotherapy for the treatment of squamous cell carcinoma of the head and neck (HNSCC). Cohen, E.E.W.; Mell, L.K.; Bell, R.B.; Bifulco, C.B.; Leidner, R.; Burtness, B.; Gillison, M.L.; Harrington, K.J.; Le, Q.-T.; Lee, N.Y.; Lewis, R.L.; Zandberg, D.P.; Ferris, R.L.; Licitra, L.; Mehanna, H.; Raben, A.; Sikora, A.G.; Uppaluri, R.; Whitworth, F. Immunother. Cancer. 2019, 7 (1). DOI: 10.1186/s40425-019-0662-5.
  3. Borcoman, E.; Marret, G.; Le Tourneau, C. Paradigm Change in First-Line Treatment of Recurrent and/or Metastatic Head and Neck Squamous Cell Carcinoma. Cancers, 2021, 13(11), 2573. DOI: 10.3390/cancers13112573.
  4. Yilmaz, E.; Ismaila, N.; Bauman, J. E.; Dabney, R.; Gan, G.; Jordan, R.; Kaufman, M.; Kirtane, K.; McBride, S. M.; Old, M. O.; et al. Immunotherapy and Biomarker Testing in Recurrent and Metastatic Head and Neck Cancers: ASCO Guideline.  J. Clin. Oncol. 2023, 41(5), 1132–1146. DOI: 10.1200/JCO.22.02328.

Point 2: The authors need to add a mechanism summary figure to highlight the new findings and their connection with previous knowledge.

Response 2: In accordance with the Reviewer’s suggestion, the manuscript has been supplemented by an additional figure (Figure 5) summarizing current and new discoveries in HNSCC immunotherapy and their relation to previous knowledge.

Point 3: The quality of the figures 2, 3 needs to be improved.

Response 3: In accordance with the Reviewer’s suggestions, the Figures have been edited and improved by a professional graphic designer for better drawing resolution; they have also been split into Figures 3A and 3B and Figures 4A and 4B to show intracellular immune signalling and the functional effects of their activity in HNSCC, as well as current therapeutic strategies for targeting immune cells and specific receptors in HNSCC.

Point 4: The conclusion should be rewritten.

Response 4: In accordance with the Reviewer’s suggestions the Conclusions and Future Directions section has been rewritten and supplemented with a justification of the value of the review paper in relation to translational medicine or clinical trials. In addition, section 3.4. Latest Immunotherapies for HNSCC, has also been supplemented.

I thank the Reviewer for reviewing my work. I hope that the new version complies with the Reviewer’s suggestions, and may be again considered for publication in Cancers.

Reviewer 3 Report

Katarzyna et al. had provided a review related to the role of immune response in HNSCC tumorigenesis, molecular signatures and the mechanism regulating pro andante cancer activity and also provide the impact of current immunotherapeutic strategies for overcoming immune escape of HNSCC and the future direction. Overall, it is well written and the content is very lengthy. I have few comments would like to provide.

1.     What is the current gap of review paper(s) that related immune response in HNSCC tumorigenesis, molecular signatures and the mechanism regulating pro and anti-cancer activity and also provide the impact of current immunotherapeutic strategies for overcoming immune escape of HNSCC. What is the gap in knowledge and what make this review paper of yours address the gap in knowledge. For example, I can name a few of review paper that exhibit similar kind of yours (PMID: 34341329, PMID: 30833002, PMID: 32987799; PMID: 32017286; PMID: 30814108).

2.     What is the take home of this paper that can improve or enlighten the clinician or scientist for translational medicine or clinical trial? This is because I cannot see this information in the future direction section. I found this review paper is repeat what we have known and fail to provide me a suggestion or enlighten to improve patient clinical outcome.

3.     Ref that use for GLOBOCAN was wrongly cite for HNSCC. As far as I am ware, it is referred to lip and oral cavity. Therefore, whatever that was cited between GLOBOCAN and HNSCC is misleading in this paper.

4.     Ref 6-10 did not conduct any epidemiology or clinical study to show the overall survival for HNSCC is less than 60%. Please change the ref by citing clinical study that show the overall survival of HNSCC patient.

5.     Ref 11, 14 and 19 is not enough to claim the result. You need to cite the TCGA HNSCC paper.

6.     Why CDKN2A was not identified as the tumor suppressive gene in line 114.

7.     The author claimed PD-L1 was fond to be over-expressed in HPV positive HNSCC compared to HPV negative HNSCC sample. As PDL1 is known as a prognosticator for HNSCC (PMID: 30666437, PMID: 26893364), wouldn’t HPV positive HNSCC will have poorer prognosis compare to HPV negative HNSCC. However, it is not the case in reality. Please provide plausible reasons to support the discrepancy.

8.     Why study for  PMID: 30666437 and PMID: 26893364 was different with ref no 318 outcome.

9.     Line 223 onward lack of the guideline from NCCN 2022 for the immunotherapy for R/M HNSCC patient.

10. Reference 318 was conducted in HPV negative OSCC. The author cannot use it to explain the role of due to over-expression of PDL1 in the tumor and therefore it correlates with high TIL in HPV positive HNSCC. The molecular and prognosis for HPV neg and positive is different.

11. How the author explain the fact that there is higher MDSC was observed in HPV positive HNSCC and yet provide a good prognosis and high TIL since MDSC is known to suppress the TIL or CD8 T cell activities.

12. Typo in line 1188

13. Line 1458, is the author is referring to anti-LAG3 instead of LAG3 alone?

14. Again ref 419 and 420 is wrongly cited. These two studies did not perform any experiment to determine the survival rates.

15.  Ref 27 and 30 is the same. Please make sure all other ref is appearing once instead of twice.

16. Is it a typo for 1.1.3 instead of 1.3?

17. Those keynote and NCT trial under section 1.13 (or 1.3) is written everywhere. Please provide a table to summarize it. It should contain the finding ie response rate and median OS and approach of treatment strategy.

18.  Why 3.2.1.1 is use for PD-1 inhibitor?

19. Figure 3 requires to break figure 3 A and B. Then Figure 3 C and D can be figure 4 A and B. This is because too pack and the word is hard to see.

20.  The author only refer to NCCN 2019 which is outdated. Please refer to the latest NCCN 2022 or 2023 for guidance.

21. Typo line 2645.

Author Response

Response to Reviewer 3 Comments

I would like to thank you for your considered, substantive and helpful review of my work. All your comments and suggestions have been taken into consideration when improving the work, and they have made an invaluable contribution to the redrafting and editing of the revised text. A point-by-point answer to the Reviewer’s comments is given below.

Point 1: What is the current gap of review paper(s) that related immune response in HNSCC tumorigenesis, molecular signatures and the mechanism regulating pro and anti-cancer activity and also provide the impact of current immunotherapeutic strategies for overcoming immune escape of HNSCC. What is the gap in knowledge and what make this review paper of yours address the gap in knowledge. For example, I can name a few of review paper that exhibit similar kind of yours (PMID: 34341329, PMID: 30833002, PMID: 32987799; PMID: 32017286; PMID: 30814108).

Response 1: The last decade has seen the considerable growth of new theories regarding the role of the immune system in tumorigenesis, particularly those associated with immune oncology and clinical immunology; this has resulted in incredible progress in immunotherapeutic strategies. However, despite the use of state-of-the-art comprehensive treatments, including surgery, radiotherapy and chemotherapy, the rates of recurrence and overall survival in HNSCC patients have not improved satisfactorily over the past few decades. Therefore, it is necessary to constantly review the current state of knowledge regarding new treatment strategies, including immunotherapy, which has turned out to be a promising alternative to standard therapies, for the benefit of clinicians and laryngologists working in Oncology. Hence, in my opinion, all works in the discussed field, especially review papers, present an important contribution to the dissemination of knowledge about the role of the immune system in carcinogenesis; they also facilitate a greater understanding of complex, multidirectional intracellular signaling and intercellular relations, which can often be obscure for clinicians, as well as the latest results from original research and clinical trials.  

As such, the example papers listed by the reviewer (PMID: 34341329, PMID: 30833002, PMID: 32987799; PMID: 32017286; PMID: 30814108), together with the present publication, are a valuable source of current knowledge on this subject. The author's discussion of the most important aspects of the immunobiology of head and neck cancers also provides a substantial contribution to the ongoing discussion on the role of the immune system in carcinogenesis. This article focuses on the crucial regulatory mechanisms of pro- and anti-tumour activity, the key genetic or epigenetic changes that favor tumour immune escape, and the strategies that tumour employs to avoid recognition by immunocompetent cells, as well as resistance mechanisms to T and NK cell-based immunotherapy in HNSCC. It describes the main mechanisms of immunotherapy, and of immunotherapy drugs that are approved or under clinical evaluation in head and neck squamous cell carcinoma (HNSCC); it also addresses drug-specific biomarkers and future directions for treatment (i.e., immunotherapy based on nanotechnology in patients with HNSCC and new immunotherapies for HNSCC). Furthermore, it also provides an overview of the pre- and clinical early trials (I/II phase) and phase-III clinical trials published in this arena, which highlight the unprecedented effectiveness and limitations of immunotherapy in HNSCC, and the emerging issues facing immuno-oncology in HNSCC.

Point 2: What is the take home of this paper that can improve or enlighten the clinician or scientist for translational medicine or clinical trial? This is because I cannot see this information in the future direction section. I found this review paper is repeat what we have known and fail to provide me a suggestion or enlighten to improve patient clinical outcome.

Response 2: As noted above, the article is intended to summarize current knowledge and recent discoveries regarding the role of the immune system in carcinogenesis in HNSCC. It also examines the clinical potential of this knowledge and future perspectives of immunotherapy. By collecting together such a wide range of theoretical, experimental and clinical findings, it is intended to provide this data in an easily accessible format to both clinicians and scientists. While review papers are intended to be a compendium of knowledge on issues selected by the author of the article, each author of a review paper also hopes that their work will become a source of inspiration; in this case, it is hoped that the article provokes further important questions about carcinogenesis in the readership and a desire to seek answers to key issues. Such inspiration may open doors for new therapeutic strategies to help prevent malignant disease.

Point 3: Ref that use for GLOBOCAN was wrongly cite for HNSCC. As far as I am ware, it is referred to lip and oral cavity. Therefore, whatever that was cited between GLOBOCAN and HNSCC is misleading in this paper.

Response 3: Unfortunately, I cannot fully agree with the reviewer's comment. Epidemiological data contained in the latest version of GLOBOCAN 2020 (Sung, H.; Ferlay, J.; Siegel, R. L.; Laversanne, M.; Soerjomataram, I.; Jemal, A.; Bray, F. Global Cancer Statistics 2020: GLOBOCAN Estimates of Incidence and Mortality Worldwide for 36 Cancers in 185 Countries. CA Cancer J. Clin. 2021, 71(3), 209–249. DOI: 10.3322/caac.21660.) concern not only lip and oral cavity cancer, but also other cancers of the head and neck region, including cancer of the oropharynx, hypopharynx, larynx, nasopharynx and other tumors of the region head and neck (please look at the attachments below, which are screen scrolls from the GFLOBOCAN 2020 website). In view of the above, I would prefer not to delete the text on HNSCC epidemiology, based on GLOBOCAN 2020 data (see the charts from the GLOBOCAN 2020 website attached below).

I would also like to add that the reference item regarding the GLOBOCAN statistical surveys has been changed to the latest version. Another recent paper discussing the epidemiology of HNSCC in recent years has also been added to the References (Gormley, M.; Creaney, G.; Schache, A; Ingarfield, K.; Conway D.I.. Reviewing the epidemiology of head and neck cancer: definitions, trends and risk factors. Br. Dent. J. 2022, 233, 780–786. DOI: 10.1038/s41415-022-5166-x). This paper provides an update on the global cancer burden using the GLOBOCAN 2020 estimates of head and neck cancer incidence and mortality produced by the International Agency for Research on Cancer.

Point 4: Ref 6-10 did not conduct any epidemiology or clinical study to show the overall survival for HNSCC is less than 60%. Please change the ref by citing clinical study that show the overall survival of HNSCC patient.

Response 4: I entirely agree. In accordance with the Reviewer’s suggestion, the indicated publications have been removed. The manuscript has been supplemented by new and relevant references on epidemiological and clinical studies which also included HPV status as a prognostic factor:

Fakhry, C.; Westra, W. H.; Wang, S. J.; van Zante, A.; Zhang, Y.; Rettig, E.; Yin, L. X.; Ryan, W. R., Ha, P. K.; Wentz, A.; et al. The prognostic role of sex, race, and human papillomavirus in oropharyngeal and non-oropharyngeal head and neck squamous cell cancer. Cancer 2017, 123(9), 1566–1575. DOI: 10.1002/cncr.30353.

Bravi, F.; Lee, Y. A.; Hashibe, M.; Boffetta, P.; Conway, D. I.; Ferraroni, M.; La Vecchia, C.; Edefonti, V. INHANCE Consortium investigators. Lessons learned from the INHANCE consortium: An overview of recent results on head and neck cancer. Oral Dis. 2021, 27(1), 73–93. DOI: 10.1111/odi.13502.

Ang, K. K.; Harris, J.; Wheeler, R.; Weber, R.; Rosenthal, D. I.; Nguyen-Tân, P. F.; Westra, W. H.; Chung, C. H.; Jordan, R. C.; Lu, C.; et al. Human papillomavirus and survival of patients with oropharyngeal cancer. N. Engl. J. Med. 2010, 363(1), 24–35. DOI: 10.1056/NEJMoa0912217.

Freitag, J.; Wald, T.; Kuhnt, T.; Gradistanac, T.; Kolb, M.; Dietz, A.; Wiegand, S.; Wichmann, G. Extracapsular extension of neck nodes and absence of human papillomavirus 16-DNA are predictors of impaired survival in p16-positive oropharyngeal squamous cell carcinoma. Cancer 2020, 126(9), 1856–1872. DOI: 10.1002/cncr.32667.

Point 5: Ref 11, 14 and 19 is not enough to claim the result. You need to cite the TCGA HNSCC paper.

Response 5: I entirely agree. In accordance with the Reviewer’s #3 suggestion, the TCGA HNSCC paper has been included – now reference number 19 (This is a report with what is currently the largest genomics data set of head and neck cancer).

Lawrence, M. S.; Sougnez, C.; Lichtenstein, L.; Cibulskis, K.; Lander, E.; Gabriel, S. B.; et al. Comprehensive genomic characterization of head and neck squamous cell carcinomas. The Cancer Genome Atlas Network. Nature 2015, 517, 576–582. DOI: 10.1038/nature14129.

The reference number 3 has also been added to this citation.

The Cancer Genome Atlas Network. Comprehensive genomic characterization of head and neck squamous cell carcinomas. Nature 2015, 517(7536), 576–582. DOI: 10.1038/nature14129.

Point 6: Why CDKN2A was not identified as the tumor suppressive gene in line 114.

Response 6: In accordance with the Reviewer’s #3 suggestion, the CDKN2 gene has been included as the tumour suppressive gene.

Point 7: The author claimed PD-L1 was fond to be over-expressed in HPV positive HNSCC compared to HPV negative HNSCC sample. As PD-L1 is known as a prognosticator for HNSCC (PMID: 30666437, PMID: 26893364), wouldn’t HPV positive HNSCC will have poorer prognosis compare to HPV negative HNSCC. However, it is not the case in reality. Please provide plausible reasons to support the discrepancy.

Response 7: The effect of PD-L1 expression on survival in HNSCC has not been clearly defined. Indeed, HPV+ve HNSCCs were observed to have increased expression of both PD-1 ligands (PD-L1 and PD-L2) in the stromal microenvironment, i.e. in both tumor cells and stroma in situ. Furthermore, it was also confirmed that PD-L1 expression has good prognostic value in these HNSCC types. It is postulated that such a relationship is strongly related to the fact that HPV+ve HNSCC patients demonstrated improved outcomes with PD-1/PD-L1 axis blockade as compared to those with HPV-ve tumours. These improved outcomes are likely driven to a greater extent by anti-PD-L1 inhibitors. Thus, the level of expression of PD-L1 may be a useful biomarker for selecting patients with better response to anti-PD1/PD-L1 therapy. This also explains that OS results obtained in papers analyzing survival rates based only on the expression of PD-L1 in treatment-naïve HSNCC patients are inconsistent with those observed in patients treated with ICIs, particularly patients with HPV positivity. There are also many discrepancies in the final conclusions, and even a lack of correlation between PD-L1 expression and prognostic indicators in HNSCC; these differences may result from bias due to small/statistically incomparable sample sizes, a failure to consider HPV status or the use of other methods of assessing this status. In addition, many groups may demonstrate different primary tumor locations, and thus different biologies.

It should be noted that the discrepancies in the results from the cited work (PMID: 30666437) may be due to the fact that the expression of the PD-L1 molecule was observed in as many as 80% of patients in a cohort study. However, it is difficult to relate this expression to statistical evaluations comparing HPV+ with HPV- if the p16 status was positive in only 12% of individuals. Moreover, PD-L1 expression was related to OS in anti-PD1/PD-L1 untreated HNSCC patients. The use of ICIs against PDL-1 would probably change the results, and thus the relationship between PD-L1 expression and survival factors.

In the second cited publication (PMID: 26893364), the authors did not take into account the HPV status at all, but indicated that PD-L1 positivity plays a role in EMT. If two categories of PD-L1 expression are included in the analyses, namely PD-L1+/EMT+ and PD-L1+/EMT-, then PD-L1 expression can be seen to be independently associated with EMT features in HNSCC. Survival analysis confirmed PD-L1+/EMT+ patients had a poorer prognosis than PD-L1+/EMT- patients in the TCGA cohort.

Point 8: Why study for PMID: 30666437 and PMID: 26893364 was different with ref no 318 outcome.

Response 8: The reference number 318 has been misquoted and removed. The manuscript has been supplemented by a new and relevant reference, the results of which are consistent with the conclusions of the cited publications.

Baruah, P.; Bullenkamp, J.; Wilson, P. O. G.; Lee, M.; Kaski, J. C.; Dumitriu, I. E. TLR9 Mediated Tumor-Stroma Interactions in Human Papilloma Virus (HPV)-Positive Head and Neck Squamous Cell Carcinoma Up-Regulate PD-L1 and PD-L2. Front. Immunol. 2019, 10, 1644. DOI: 10.3389/fimmu.2019.01644.

Point 9: Line 223 onward lack of the guideline from NCCN 2022 for the immunotherapy for R/M HNSCC patient.

Response 9: I entirely agree. In accordance with the Reviewer’s suggestions, a source of information on clinical recommendations for the treatment of HNSCC has been supplemented with the latest NCCN Guidelines version (Version 1.2023, 01/30/23) in this part of the manuscript.

Point 10: Reference 318 was conducted in HPV negative OSCC. The author cannot use it to explain the role of due to over-expression of PDL1 in the tumour and therefore it correlates with high TIL in HPV positive HNSCC. The molecular and prognosis for HPV neg and positive is different.

Response 10: I entirely agree. The reference number 318 has been misquoted and removed. The manuscript has been supplemented by a new and relevant reference, the results of which explain the role of over-expression of PD-L1 in the in HPV-positive HNSCC.

Baruah, P.; Bullenkamp, J.; Wilson, P. O. G.; Lee, M.; Kaski, J. C.; Dumitriu, I. E. TLR9 Mediated Tumor-Stroma Interactions in Human Papilloma Virus (HPV)-Positive Head and Neck Squamous Cell Carcinoma Up-Regulate PD-L1 and PD-L2. Front. Immunol. 2019, 10, 1644. DOI: 10.3389/fimmu.2019.01644.

Point 11: How the author explain the fact that there is higher MDSC was observed in HPV positive HNSCC and yet provide a good prognosis and high TIL since MDSC is known to suppress the TIL or CD8 T cell activities.

Response 11: There was an error in the text, i.e. the wrong term “MDSCs” (meaning Myeloid-Derived Suppressor Cells) was used instead of “MDCs” (meaning Myeloid Dendritic Cells). This has been corrected in the text and in Table 2.

Cyt.: In HPV-positive HNSCC, an increase in the infiltration of immunocompetent cells, i.e. T cells, including CD3+, CD4+, CD8+ IFN-γ cytotoxic T cells (CTLs), CD19+/CD20+ B cells, CD56dim/CD16bright activated NK cells, tumour infiltrating antigen presenting cells/dendritic cells (APCs/DCs), myeloid and plasmacytoid dendritic cells (MDCs) and a decrease of CD4+CD25+Foxp3+ regulatory T cells in TILs is observed, which enhances the stimulation of cellular immunity to tumour antigens. A higher MDCs was observed in HPV positive HNSCC and yet provide a good prognosis.

Point 12: Typo in line 1188.

Response 12: Typo line has been corrected.

Point 13: Line 1458, is the author is referring to anti-LAG3 instead of LAG3 alone?

Response 13: In accordance with the Reviewer's suggestions, the text has been revised and corrected (instead of the erroneous term LAG-3, the phrase "the antibodies targeting LAG-3" was used).

Point 14: Again ref 419 and 420 is wrongly cited. These two studies did not perform any experiment to determine the survival rates. 

Response 14: I entirely agree. In accordance with the Reviewer’s suggestions, references 419 and 420 have been changed and the correct references have been added.

Argiris A.; Harrington K.J.; Tahara M.; Schulten J.; Chomette P.; Ferreira Castro A.; Licitra L. Evidence-Based Treatment Options in Recurrent and/or Metastatic Squamous Cell Carcinoma of the Head and Neck. Front. Oncol. 2017, 7, 72. DOI: 10.3389/fonc.2017.00072.

Argiris A.; Harrington K.J.; Tahara M.; Schulten J.; Chomette P.; Ferreira Castro A.; Licitra L. Evidence-Based Treatment Options in Recurrent and/or Metastatic Squamous Cell Carcinoma of the Head and Neck. Front. Oncol. 2017, 7, 72. DOI: 10.3389/fonc.2017.00072.

Pontes, F.; Garcia, A. R.; Domingues, I.; João Sousa, M.; Felix, R.; Amorim, C.; Salgueiro, F.; Mariano, M.; Teixeira, M. Survival predictors and outcomes of patients with recurrent and/or metastatic head and neck cancer treated with chemotherapy plus cetuximab as first-line therapy: A real-world retrospective study. Cancer Treat. Res. Commun. 2021, 27, 100375. DOI:10.1016/j.ctarc.2021.100375.

Point 15: Ref 27 and 30 is the same. Please make sure all other ref is appearing once instead of twice.

Response 15: I entirely agree. In accordance with the Reviewer's suggestions, reference 30 has been changed and the correct reference has been added.

Schoenfeld, J. D.; Gjini, E.; Rodig, S. J.; Tishler, R. B.; Rawal, B.; Catalano, P. J.; Uppaluri, R.; Haddad, R. I.; Hanna, G. J.; Chau, N. G.; et al. Evaluating the PD-1 Axis and Immune Effector Cell Infiltration in Oropharyngeal Squamous Cell Carcinoma. Int. J. Radiat. Oncol. Biol. Phys. 2018, 102(1), 137–145. DOI: 10.1016/j.ijrobp.2018.05.002.

Point 16: Is it a typo for 1.1.3 instead of 1.3?

Response 16: Yes, it's a typo. Once again, the text has been divided into appropriate sections.

Point 17: Those keynote and NCT trial under section 1.13 (or 1.3) is written everywhere. Please provide a table to summarize it. It should contain the finding ie response rate and median OS and approach of treatment strategy.

Response 17: An additional Supplementary Table 1 (due to the limitation of the number of tables and figures by the Cancers Editor) with a summary of key immunotherapy trials incorporated ICIs in metastatic HNSCC setting has been included in the manuscript.

Point 18: Why 3.2.1.1 is use for PD-1 inhibitor?

Response 18: In accordance with the Reviewer’s suggestion, section 3.2.1.1 has been edited. Chapters 3.2.1.1. and 3.2.1.2. have been integrated.

Point 19: Figure 3 requires to break figure 3 A and B. Then Figure 3 C and D can be figure 4 A and B. This is because too pack and the word is hard to see.

Response 19: In accordance with the Reviewer’s suggestion Figure 3 has been edited and split into Figures 3A and 3B and Figures 4A and 4B.

Point 20: The author only refer to NCCN 2019 which is outdated. Please refer to the latest NCCN 2022 or 2023 for guidance.

Response 20: In accordance with the Reviewer’s suggestion, the key clinical recommendations for the treatment of HNSCC have been completed based on the latest NCCN Guidelines version (Version 1.2023, 01/30/23) and the latest therapeutic data. Further detail into management strategies have been also referred to the American Society of Clinical Oncology® (ASCO®) Head and Neck Cancer Guideline; Knowledge Conquers  Cancer [Available from: https://www.cancer.net/cancer-types/head-and-neck-cancer/types-treatment#immunotherapy].
Practical recommendations for treatment have been extended.

The National Comprehensive Cancer Network® (NCCN®) Clinical Practice Guidelines in Oncology (NCCN Guidelines®). Head and Neck Cancers; NCCN Evidence BlocksTM (Version 1.2023, 01/30/23) [Available from: www.nccn.org/patents for current list of applicable patents. Head and Neck Cancers Version 1.2023 - January 30, 2023].

The Society for Immunotherapy of Cancer consensus statement on immunotherapy for the treatment of squamous cell carcinoma of the head and neck (HNSCC). Cohen, E.E.W.; Mell, L.K.; Bell, R.B.; Bifulco, C.B.; Leidner, R.; Burtness, B.; Gillison, M.L.; Harrington, K.J.; Le, Q.-T.; Lee, N.Y.; Lewis, R.L.; Zandberg, D.P.; Ferris, R.L.; Licitra, L.; Mehanna, H.; Raben, A.; Sikora, A.G.; Uppaluri, R.; Whitworth, F. J. Immunother. Cancer. 2019, 7 (1). DOI: 10.1186/s40425-019-0662-5.

Borcoman, E.; Marret, G.; Le Tourneau, C. Paradigm Change in First-Line Treatment of Recurrent and/or Metastatic Head and Neck Squamous Cell Carcinoma. Cancers, 2021, 13(11), 2573. DOI: 10.3390/cancers13112573.

Yilmaz, E.; Ismaila, N.; Bauman, J. E.; Dabney, R.; Gan, G.; Jordan, R.; Kaufman, M.; Kirtane, K.; McBride, S. M.; Old, M. O.; et al. Immunotherapy and Biomarker Testing in Recurrent and Metastatic Head and Neck Cancers: ASCO Guideline. Am. J. Clin. Oncol. 2023, 41(5), 1132–1146. DOI: 10.1200/JCO.22.02328.

Point 21: Typo in line 2645.

Response 21: The typo has been corrected.

I thank the Reviewer for reviewing my work. I hope that the new version complies with the Reviewer’s suggestions, and may be again considered for publication in Cancers.

Reviewer 4 Report

The presented review is comprehensive and highly complex, Authors describe the role of immune cells and also cancer-associated fibroblasts in the biology of head and neck cancer. The authors describe in detail the function of all types of leukocytes infiltrating the tumor and their dialogue with cancer cells and cancer-associated fibroblasts with a focus on paracrine bioactive factors and exosomes. The review is based on almost 600 references, For a better understanding 2 tables and 3 schematic pictures are included. The pictures represent weak points because the letters are so small that they are not easily readable. They need revision.

Author Response

Response to Reviewer 4 Comments

I would like to thank you for your considered, substantive and helpful review of my work. Your suggestion has been taken into consideration when improving the work.

Point: The pictures represent weak points because the letters are so small that they are not easily readable. They need revision.

Response: In accordance with the Reviewer’s suggestions, the Figures have been edited and improved by a professional graphic designer for better drawing resolution. Moreover, Figure 3 has been edited, completed and split into Figures 3A and 3B and Figures 4A and 4B.

I thank the Reviewer for reviewing my work. I hope that the new version complies with the Reviewer’s suggestions, and may be again considered for publication in Cancers.

Round 2

Reviewer 3 Report

Hi Aiyarucht, I have gone through the responses based on the author reply.  My decision is accepting in current format. Again, I failed to access the website for my decision. Plus I am currently out of the office and I cannot access to my working email. Please kindly do the needful and if you insist me to provide my decision pls send to my personal email vincent_sean@hotmail.com

Thank you   

Vincent